# MCM complexes are barriers that restrict cohesin-mediated loop extrusion

Bart J. H. Dequeker[1,10], Matthias J. Scherr[2,10], Hugo B. Brandão[3,9], Johanna Gassler[1,4], Sean Powell[1], Imre Gaspar[4], Ilya M. Flyamer[5], Aleksandar Lalic[4], Wen Tang[6], Roman Stocsits[6], Iain F. Davidson[6], Jan-Michael Peters[6], Karl E. Duderstadt[2,7✉], Leonid A. Mirny[8✉] & Kikuë Tachibana[1,4✉]

Eukaryotic genomes are compacted into loops and topologically associating domains (TADs)[1–3], which contribute to transcription, recombination and genomic stability[4,5]. Cohesin extrudes DNA into loops that are thought to lengthen until CTCF boundaries are encountered[6–12]. Little is known about whether loop extrusion is impeded by DNA-bound machines. Here we show that the minichromosome maintenance (MCM) complex is a barrier that restricts loop extrusion in G1 phase. Single-nucleus Hi-C (high-resolution chromosome conformation capture) of mouse zygotes reveals that MCM loading reduces CTCF-anchored loops and decreases TAD boundary insulation, which suggests that loop extrusion is impeded before reaching CTCF. This effect extends to HCT116 cells, in which MCMs affect the number of CTCF-anchored loops and gene expression. Simulations suggest that MCMs are abundant, randomly positioned and partially permeable barriers. Single-molecule imaging shows that MCMs are physical barriers that frequently constrain cohesin translocation in vitro. Notably, chimeric yeast MCMs that contain a cohesin-interaction motif from human MCM3 induce cohesin pausing, indicating that MCMs are 'active' barriers with binding sites. These findings raise the possibility that cohesin can arrive by loop extrusion at MCMs, which determine the genomic sites at which sister chromatid cohesion is established. On the basis of in vivo, in silico and in vitro data, we conclude that distinct loop extrusion barriers shape the three-dimensional genome.

Eukaryotic genomes are folded into loops that are generated by structural maintenance of chromosomes (SMC) proteins, including cohesin and condensin complexes (reviewed previously[13]). Structures that emerge through loop extrusion are detected by Hi-C experiments. The extrusion process is hypothesized to form progressively larger loops until cohesin encounters a barrier and/or is released by Wapl (refs. [9–11]). The predominant barrier to loop extrusion in vertebrates is CTCF (ref. [12]), which has an instructive role in establishing extrusion-mediated structures that are visible in Hi-C[14]. However, the loop extrusion machinery encounters other obstacles on chromatin, such as nucleosomes and other protein complexes. Although RNA polymerases are moving barriers for condensin translocation in bacteria[15] and affect cohesin translocation in eukaryotes[16,17], it remains unknown how SMCs can extrude loops on 'busy' eukaryotic chromosomes that are bound by a myriad of proteins. Whether other DNA-bound proteins can influence three-dimensional genome architecture in eukaryotes is not known, and could be critical for understanding their function.

The minichromosome maintenance (MCM) complex is an abundant macromolecular machine that is essential for DNA replication in eukaryotes and archaea[18]. MCM2–MCM7 complexes (hereafter MCM) are loaded at replication origins by the origin recognition complex (ORC), Cdc6 and Cdt1 to form the pre-replication complex during mitosis and G1 phase[19]. The head-to-head double MCM hexamer topologically entraps double-stranded DNA and is catalytically inactive as a helicase until the initiation of DNA replication[20]. Notably, 10–100-fold more MCMs are loaded onto chromatin than are needed for S-phase progression. This is referred to as the 'MCM paradox'[21]. One hypothesis to explain this phenomenon is that surplus complexes mark dormant origins that fire under conditions such as DNA damage checkpoint activation[22]. Surplus MCMs have been shown to protect against DNA breaks by reducing replication fork speed[23]. Whether they have any functional consequences in G1 phase remains unclear. Given the abundance of MCMs, their long residence time on chromatin[24] (more than 6 h) and their comparable size[25] (13 nm) to the FtsK helicase (12.5 nm) (Extended Data Fig. 1) that can push cohesin on DNA in vitro[26], we asked whether MCMs are obstacles to cohesin-mediated loop extrusion and in this way influence genome architecture.

[1]Institute of Molecular Biotechnology of the Austrian Academy of Sciences (IMBA), Vienna BioCenter (VBC), Vienna, Austria. [2]Structure and Dynamics of Molecular Machines, Max Planck Institute of Biochemistry (MPIB), Martinsried, Germany. [3]Harvard Program in Biophysics, Harvard University, Cambridge, MA, USA. [4]Department of Totipotency, Max Planck Institute of Biochemistry (MPIB), Martinsried, Germany. [5]MRC Human Genetics Unit, Institute of Genetics and Molecular Medicine (IGMM), University of Edinburgh, Edinburgh, UK. [6]Research Institute of Molecular Pathology (IMP), Vienna BioCenter (VBC), Vienna, Austria. [7]Department of Physics, Technical University of Munich, Garching, Germany. [8]Department of Physics, Massachusetts Institute of Technology (MIT), Cambridge, MA, USA. [9]Present address: Illumina Inc., San Diego, CA, USA. [10]These authors contributed equally: Bart J. H. Dequeker, Matthias J. Scherr. ✉e-mail: duderstadt@biochem.mpg.de; leonid@mit.edu; tachibana@biochem.mpg.de

## MCMs impede CTCF-anchored loops

To test this hypothesis, we used the oocyte-to-zygote transition to investigate whether MCM loss affects loop extrusion. Oocytes are female germ cells that divide meiotically and, after fertilization, generate one-cell embryos (zygotes). These contain maternal and paternal pronuclei, the chromatin of which is organized into cohesin-dependent loops and TADs[27,28]. Although zygotes are limited by paucity of material, they offer advantages for: (1) studying MCM loading on newly assembled paternal chromatin; (2) deciphering haplotype-resolved chromatin organization; (3) manipulating the assembly of the pre-replication complex without interfering with cell-cycle progression, as there is no DNA replication between meiosis I and II; and (4) disentangling direct from indirect effects because of transcriptional inactivity[29].

To generate zygotes that are deficient in chromatin-bound MCMs, we interfered with the Cdt1-mediated loading pathway. Cdt1 deposits MCMs onto chromatin, and this reaction is inhibited by geminin, a target of the anaphase-promoting complex/cyclosome (APC/C)[30] (Extended Data Fig. 2a). Mutation of geminin's destruction box generates a non-degradable version (geminin(L26A)) that inhibits the Cdt1-mediated loading of MCMs in G1 phase[31] (Extended Data Fig. 2a). To achieve this, mouse oocytes were microinjected with mRNA encoding an injection marker GFP with or without geminin(L26A) (Fig. 1a). Metaphase II eggs were fertilized in vitro and zygotes were analysed in G1 phase (Extended Data Fig. 2c). Geminin(L26A) expression did not grossly affect the abundance of Scc1 and CTCF (Extended Data Fig. 2e, f). By contrast, few or no chromatin-bound MCMs were detected and EdU was not incorporated in zygotes expressing geminin(L26A) (referred to as 'MCM loss') (Fig. 1b, Extended Data Fig. 2b, d), demonstrating the efficient inhibition of MCM loading.

Using this approach, we generated MCM-loss and control zygotes, isolated maternal and paternal pronuclei in G1 phase and performed single-nucleus Hi-C (snHi-C) (Fig. 1a). The sparsity of snHi-C data precluded de novo calling of loops (referred to as 'peaks'), which represent contacts between CTCF-bound loci. Instead, we used 12,000 loop coordinates from mouse embryonic fibroblast Hi-C data that report on cohesin-dependent contacts in zygotes[32] (Fig. 1c). Notably, MCM loss resulted in an increase in aggregate peaks and aggregate TADs (referred to as 'peaks' and 'TADs') in maternal chromatin and an even stronger increase in paternal chromatin (Fig. 1d, Extended Data Fig. 3a–c), which will be focused on hereafter. The increase in peak strengths after MCM loss could reflect higher CTCF occupancy, but this could not explain the barrier effect seen in vitro (see Fig. 4d). Alternatively, it could reflect increased access of cohesin to CTCF sites, owing to changes in either loop extrusion (potentially caused by barrier loss) or transcription. There were few transcriptomic differences between control and MCM-loss zygotes (Extended Data Fig. 4a, b). We conclude that MCMs hinder the formation of CTCF-anchored loops and TADs largely independently of changes in gene expression.

To find out whether cohesin is responsible for the increase in snHi-C peak strength caused by MCM loss, we used a conditional genetic knockout approach based on Cre recombinase under control of the *Zp3* promoter to delete floxed alleles of the cohesin subunit *Scc1* in oocytes[27,33]. We expressed geminin(L26A) in *Scc1*[Δ/Δ] oocytes isolated from *Scc1*[fl/fl] (*Tg*)*Zp3*-Cre females and generated maternal *Scc1* knockout zygotes (*Scc1*[Δ(m)/+(p)]) (Extended Data Fig. 4c). Loops and TADs were undetectable in *Scc1*[Δ(m)/+(p)] zygotes, as reported previously[27], and remained undetectable if MCM loading was prevented (Extended Data Fig. 4d). We conclude that MCMs interfere with cohesin-dependent chromatin structures.

To determine how MCMs affect chromatin organization, we examined the contact probability $P_c(s)$ as a function of genomic distance (*s*) (Fig. 1e, Extended Data Fig. 3d, e). The position of the 'shoulder' on the $P_c(s)$ curve is informative of the mean size of extruded loops[27]

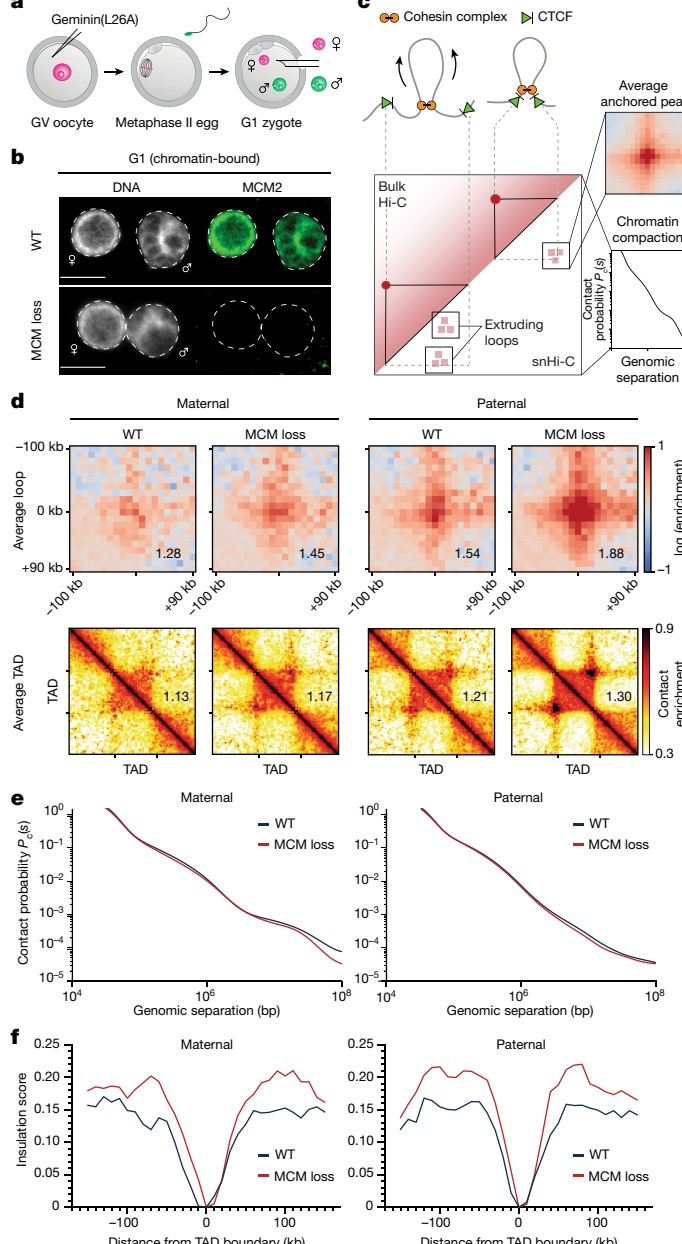

**Fig. 1 | Chromatin-bound MCMs impede loop and TAD formation in G1-phase zygotes. a**, Germinal vesicle (GV)-stage oocytes were injected with geminin(L26A) mRNA and metaphase II eggs were fertilized in vitro. Maternal and paternal pronuclei were extracted for snHi-C. **b**, Representative images of immunofluorescence staining of chromatin-bound MCM2 in wild-type (WT) and MCM-loss G1-phase zygotes. DNA is stained with DAPI. Scale bars, 10 μm. **c**, Comparison of contacts detected in snHi-C versus bulk Hi-C. Contact probability curves, $P_c(s)$, provide insights into chromatin compaction. **d**, Average loops and TADs for wild-type and MCM-loss chromatin in G1 phase. The data shown are based on *n* (WT, maternal) = 13, *n* (WT, paternal) = 16, *n* (MCM loss, maternal) = 16, *n* (MCM loss, paternal) = 15, from 4 independent experiments using 4–6 females for each experiment. Heat maps were normalized to an equal number of *cis* contacts. **e**, $P_c(s)$ curves for wild-type and MCM-loss conditions. **f**, Insulation scores at TAD borders.

(Extended Data Fig. 3f). Of note, MCM loss has little effect on the $P_c(s)$ curve below 1 Mb (Fig. 1e); this is reminiscent of CTCF loss[14,34], and suggests that the mean size of extruded loops is largely unaffected. However, the effect of CTCF loss on 'peaks' is opposite to that of MCM loss. We reasoned that if MCMs impede formation of CTCF-mediated

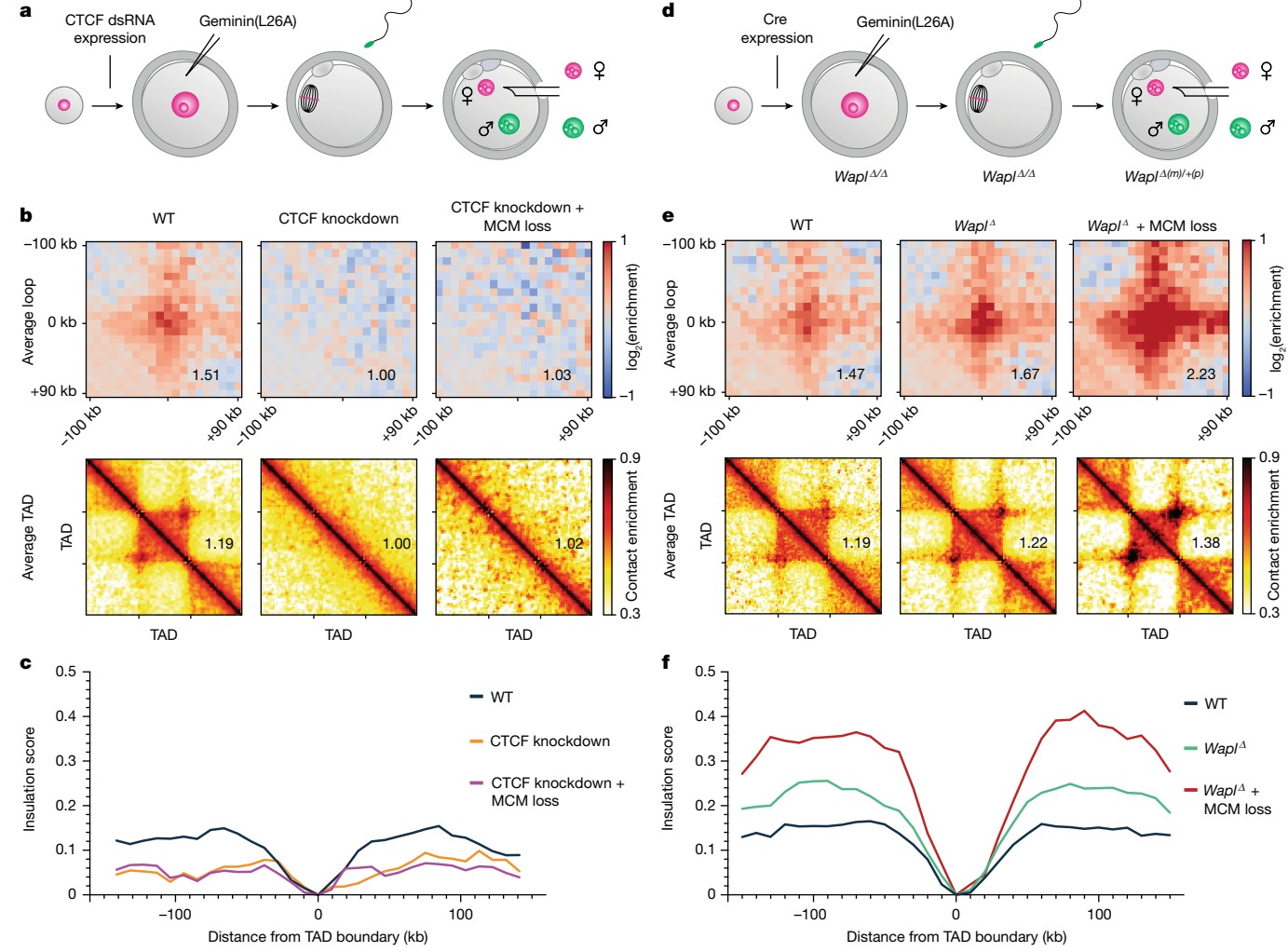

**Fig. 2 | MCMs impede CTCF-anchored loops and function independently of Wapl. a**, CTCF-knockdown oocytes from (*Tg*)*Zp3-dsCTCF* females were injected with geminin(L26A) mRNA and eggs were fertilized to generate zygotes for snHi-C analysis in G1 phase. **b**, Average loops and TADs for wild-type, CTCF-knockdown and CTCF-knockdown + MCM loss for paternal chromatin in G1. Data are based on *n* (WT, paternal) = 12, *n* (CTCF knockdown, paternal) = 8 and *n* (CTCF knockdown + MCM loss, paternal) = 8 nuclei, from 4 independent experiments using 4–6 females for each genotype. Heat maps were normalized to an equal number of *cis* contacts. **c**, Insulation scores at TAD borders for paternal chromatin. **d**, *Wapl*^Δ/Δ oocytes from *Wapl*^fl/fl (*Tg*)*Zp3*-Cre females were

injected with geminin(L26A) mRNA and eggs were fertilized to generate zygotes for snHi-C analysis in G1 phase. **e**, Average loops and TADs for control (wild type), *Wapl*^Δ and *Wapl*^Δ + MCM loss for paternal chromatin in G1. Data shown are based on *n* (WT, paternal) = 20, *n* (*Wapl*^Δ, paternal) = 11, *n* (*Wapl*^Δ + MCM loss, paternal) = 9 nuclei, from 4 independent experiments using 4–6 females for each genotype. Control samples are wild type (this study) pooled with *Wapl*^fl samples (published previously[27]). Heat maps were normalized to an equal number of *cis* contacts. **f**, Insulation scores at TAD borders for paternal chromatin.

structures, then MCM loss should lead to increased CTCF peaks and increased insulation of TAD boundaries, as observed (Fig. 1f). These effects on chromatin organization are consistent with a mechanism in which MCMs impede loop extrusion by altering loop positioning without considerably changing their sizes.

We tested whether CTCF and MCM together determine the strengths of peaks and TAD boundary insulation. We expressed geminin(L26A) in CTCF-knockdown oocytes isolated from *(Tg)Zp3-CTCFdsRNA* females and generated maternal CTCF-knockdown zygotes[35] (Fig. 2a, Extended Data Fig. 5a). CTCF knockdown without or with MCM perturbation resulted in a loss of loops and TADs (Fig. 2b, Extended Data Fig. 5b); this shows that CTCF is essential for these structures in zygotes. Knockdown of CTCF caused a weakening of TAD boundary insulation and did not grossly change $P_c(s)$ curves below 1 Mb (Fig. 2c, Extended Data Fig. 5c–e). The lack of TAD organization after knockdown of CTCF, irrespective of MCMs, suggests that MCMs have no instructive function for establishing position-specific boundaries. This is

consistent with MCMs being largely located in different positions in different cells[36].

We considered an alternative possibility that MCMs affect loops by functioning with Wapl in releasing cohesin from chromatin. This is based on the similar effects on Hi-C peak and TAD strengths after *Wapl* knockout and MCM loss[27,34,37] (Fig. 2e). If Wapl and MCMs function together, then their co-depletion would be expected to resemble individual depletions. If they function independently, then co-depletion could have synergistic effects. To distinguish between these, we expressed geminin(L26A) in *Wapl*^Δ/Δ oocytes isolated from *Wapl*^fl/fl (*Tg*)*Zp3*-Cre females and generated maternal *Wapl* knockout zygotes[32] (*Wapl*^Δ(m)/+(p)) (Fig. 2d). Combined MCM loss and *Wapl* knockout strongly increased peak and TAD strengths over the individual conditions (Fig. 2e, Extended Data Fig. 6a–d), suggesting that they function through separate mechanisms. The combined loss increased TAD boundary insulation, suggesting that MCMs restrict loop extrusion also when cohesin residence time is increased (Fig. 2f, Extended Data Fig. 6b).

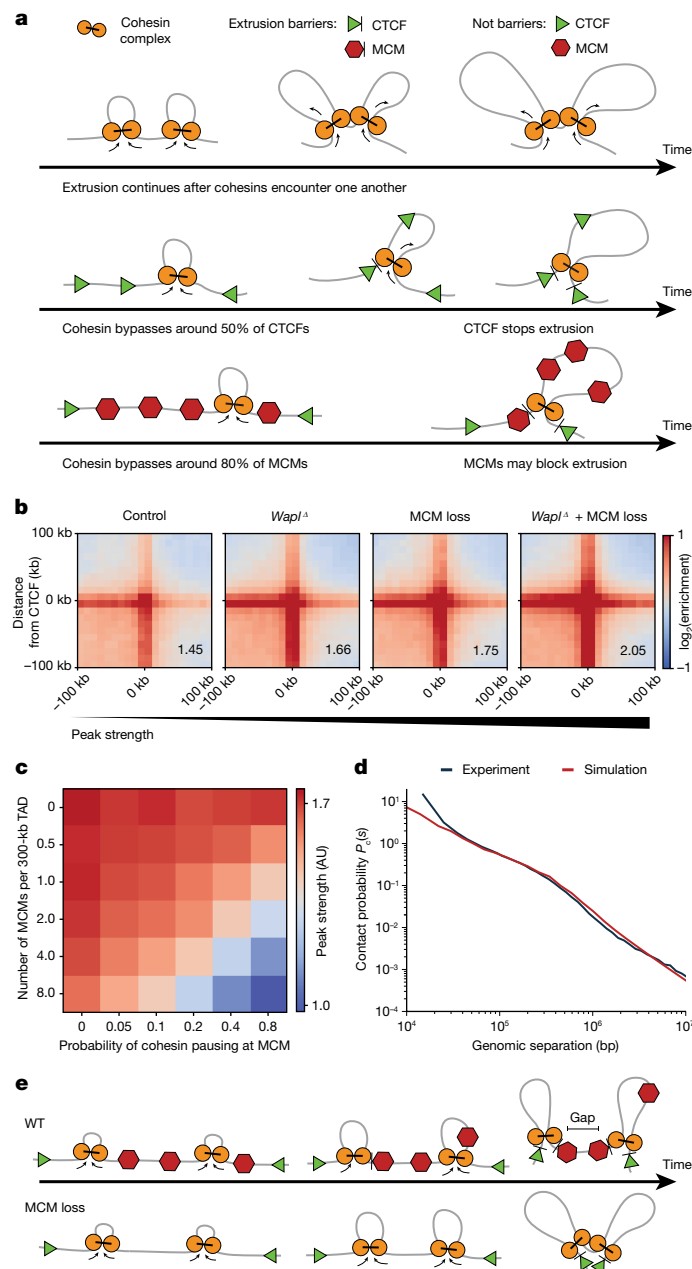

**Fig. 3 | Simulation model of MCMs as a random barrier to cohesin loop extrusion. a**, A quantitative model for loop extrusion by cohesin and interactions with CTCF and MCMs. Cohesin (yellow) extrudes loops. On encountering barriers such as CTCF sites (green triangles), MCMs (red hexagons) or other cohesins, extrusion can be blocked. Cohesins may bypass some CTCFs and MCMs, but not others; the choice to bypass an MCM or CTCF site is stochastic and varies in time. **b–d**, For paternal chromatin simulations, we assume that CTCF stalls cohesin around 45% of the time, as measured in mouse embryonic stem cells[12], and that MCMs stall cohesin 20% of the time. **b**, Peak strengths for simulated paternal chromatin under various perturbations for TADs of 300 kb average length. **c**, Matrix of peak strengths in the wild type, showing a linear trade-off between MCM density and its ability to pause cohesin. AU, arbitrary units. **d**, The simulated contact probability decay curve $P_c(s)$ for the MCM-loss condition is well matched with experimental data. **e**, Model summarizing the finding that chromatin-bound MCMs can function as barriers for loop extrusion in G1 phase.

## MCMs are semi-permeable barriers

We introduced MCMs as randomly located permeable extrusion barriers into polymer models of loop extrusion (Fig. 3a). Polymer

simulations identified parameters, including the permeability of MCMs, such that the peak strengths and $P_c(s)$ curves of paternal chromatin can be quantitatively reproduced (Fig. 3b, d, Extended Data Fig. 6e–h, Supplementary Figs. 2–5). In the model, cohesin extrudes loops and is stopped at CTCF sites (around 50% of encounters)[38] and MCMs (around 20% of encounters, for an estimated density of 1 MCM per 75 kb) (Fig. 3a). If MCM density is lower in zygotes, then blocking will occur more frequently (more than 20%) (Fig. 3c). The predicted semi-permeability of MCMs could explain how CTCF-anchored loops are generated in the presence of MCMs in G1 phase.

Our model provides a rationale for the seemingly contradictory ability of MCM to reduce CTCF–CTCF peak strength without strongly affecting the mean size of extruded loops. A peak emerges if cohesin extrudes all chromatin between a pair of CTCFs into loops. A random barrier prevents cohesin from extruding all chromatin between CTCFs, leaving an unextruded gap[39] (Fig. 3e). The effect of random barriers on the average loop size is, however, marginal (less than 15%) if barriers are sufficiently permeable or sparse (one per TAD of around 300 kb). This is an unexpected effect of random barriers on features of chromosome organization.

## MCMs affect transcription and loops

As these findings were obtained in zygotes, we tested whether MCMs also impede loop extrusion in somatic cells. To directly degrade MCMs, we treated G1-synchronized HCT116 cells carrying auxin-inducible degron MCM2-mAID alleles with dimethyl sulfoxide (DMSO) or auxin[40] (Extended Data Figs. 7a–g, 8a–f). Treatment with auxin reduced chromatin-bound MCM2 and MCM4 without grossly affecting the abundance of CTCF and cohesin (Extended Data Figs. 7c, 8c). Acute MCM degradation resulted in the differential expression of 229 genes (Extended Data Fig. 8l, m), which is comparable to the effects of acute CTCF degradation[14]. Hi-C data showed a moderate increase in aggregate peak strengths in MCM-depleted versus control cells (Extended Data Fig. 7d, e). To confirm this result using another method, we performed Micro-C and found that MCM depletion results in a moderate but genome-wide and significant ($P = 1.87 \times 10^{-70}$) increase in the peak strength (Extended Data Fig. 8d, g–i). Notably, de novo peak calling identified a greater number of loops in MCM-depleted cells, consistent with loop extrusion reaching CTCF sites more frequently (Extended Data Fig. 8j, k). The effects show the same directionality but are much more subtle than in zygotes and cannot be explained solely by RNA polymerase in somatic cells (Extended Data Fig. 8n, o). On the basis of the consistent increases in loop strengths and numbers after MCM degradation, we conclude that MCMs impede the formation of CTCF-anchored loops in somatic cells.

## MCMs block cohesin translocation

The most parsimonious interpretation of the effects of MCM loss on genome architecture is that MCMs interfere with loop extrusion by forming randomly located barriers. To directly test this, we established an MCM 'roadblock assay' for passive translocation of cohesin using total internal reflection fluorescence microscopy that detects real-time cohesin–MCM interactions at the single-molecule level. Origin licensing was reconstituted from purified components in a stepwise manner on origin-containing DNA molecules[43] (Fig. 4a). Loading of yeast MCM and double-hexamer formation—a hallmark of proper origin licensing—was observed in the presence of ORC, Cdc6 and Cdt1 (ref. [41]) (Extended Data Fig. 9a). Cohesin was introduced in low-salt conditions, followed by a high-salt wash to select for fully loaded MCMs[41–43]. To mimic intracellular conditions, experiments were imaged in physiological salt conditions, promoting cohesin translocation on fast timescales[38] (Extended Data Fig. 9b, Supplementary Video 1). The cohesin

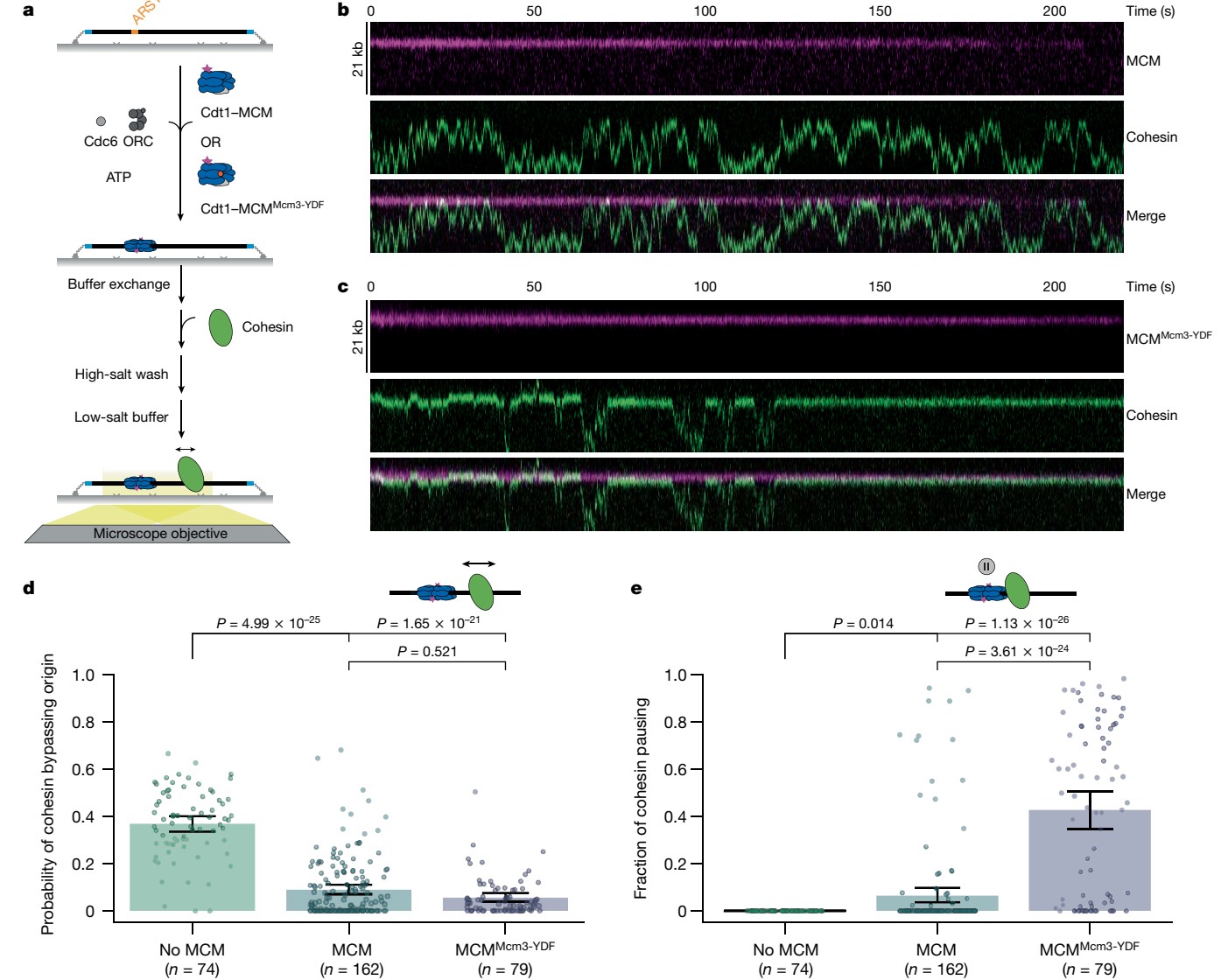

**Fig. 4 | MCMs are barriers for cohesin translocation in vitro. a**, Principle of a single-molecule cohesin translocation assay on licensed DNA. MCM is loaded onto DNA by the licensing factors ORC, Cdc6 and Cdt1, followed by cohesin. A high-salt wash removes licensing factors and intermediates from DNA. Cohesin translocation is visualized at physiological salt conditions (0.15 M NaCl) without free protein and buffer flow. **b**, **c**, Representative kymographs of translocating cohesin on licensed DNA. Origin-bound MCM (**b**) and MCM$^{Mcm3-YDF}$ (**c**) are efficient barriers for cohesin translocation. **d**, Probability of

translocating cohesin bypassing the origin in the absence or in the presence of MCM or MCM$^{Mcm3-YDF}$, calculated from 74, 162 or 79 molecules with 12,175, 15,348 or 9,455 visualized encounters, respectively. **e**, Cohesin translocation pauses at MCM in a YDF-disordered-region-dependent manner. Fraction of cohesin pausing of the total observation time in the absence or in the presence of MCM or MCM$^{Mcm3-YDF}$. Data in **d**, **e** are depicted as mean within a 95% confidence interval (generated by bootstrapping). *P* values were determined by Kruskal–Wallis test followed by Dunn's post-hoc test.

diffusion coefficient remained unchanged in the presence of MCMs (Extended Data Fig. 9c).

Direct visualization of cohesin encounters with MCMs revealed constrained cohesin translocation with a fourfold reduction in origin passage (Fig. 4b, d, Extended Data Fig. 9d–g, Supplementary Videos 2, 3). Similar results were obtained at higher salt concentrations (Extended Data Figs. 9c, 10a–f, Supplementary Videos 4–6). A subpopulation of cohesin molecules were unable to pass origins even once during the 220-s imaging window (67/162; Fig. 4b, Extended Data Fig. 9d). By contrast, origin passage was unimpeded in the absence of MCMs (72/74; Extended Data Fig. 9b). The observed permeability of MCMs is lower than that predicted by simulations, which could be a result of different conditions from loop extrusion in vivo. It will be important in future studies to test whether MCMs halt loop extrusion, which requires a combined assay that has thus far not been established owing to different

reaction conditions in vitro. We conclude that MCMs are physical barriers to cohesin translocation and may occasionally be bypassed.

Finally, we tested whether mammalian MCMs are stronger barriers than yeast MCMs to cohesin translocation. Unlike yeast Mcm3, human MCM3 contains a 19-amino-acid disordered region containing a YDF motif that is sufficient to bind STAG2–SCC1 cohesin in vitro[44]. The same motif mediates an interaction between CTCF and STAG2–SCC1 (ref. [44]). As there is no established human origin licensing assay, we modified the yeast assay to load a chimeric MCM complex containing a 'humanized' MCM3 subunit (MCM-YDF). We found that cohesin bypasses MCM-YDF slightly less frequently compared to MCM, suggesting that its barrier strength is comparable to that of yeast MCMs (Fig. 4d). Notably, we observed frequent pausing of cohesin upon encountering MCM-YDF (Fig. 4c, Extended Data Fig. 9h, i), with pauses accounting for 43% of the total observation time on average (Fig. 4e).

These pauses were much less frequent (6.4%) in the presence of MCMs lacking YDF, which suggests that pausing reflects a molecular docking of cohesin to the YDF region.

## Discussion

We have identified MCM complexes as barriers for loop extrusion on the basis of in vivo, in silico and in vitro data. MCMs are members of a new class of randomly positioned and cell-cycle-phase-specific barriers that impede the formation of CTCF-anchored loops and TADs. A key question is which features determine whether a protein impedes loop extrusion. Nanoparticles larger than the diameter of SMC complexes can be bypassed by extruding SMC complexes[45], suggesting that size is not the sole determinant. Unlike those permeable roadblocks, MCMs have two distinguishing features that can promote barrier function. MCMs bind DNA in a topological manner. Although the mechanism of bypassing obstacles is not known, it is conceivable that a topological engagement of proteins around DNA could interfere with cohesin binding to DNA to 'swing over' a protein[46]. Consistent with this, cohesin that topologically entraps sister chromatids restricts loop extrusion mediated by other cohesin complexes in oocytes[32,47]. In addition, the YDF disordered region alters the outcome of cohesin–MCM collisions from blocking to pausing, which suggests that MCMs are active chemical barriers with binding sites rather than passive physical barriers.

The finding that MCMs are barriers to loop extrusion provides a different perspective on the body of knowledge on MCMs and cohesin loading. MCMs recruit cohesin to pre-replication complexes in *Xenopus* extracts[48,49] and promote cohesin loading during DNA replication in human cells[50]. These studies proposed that MCMs have a role in loading cohesin; that is, capturing cohesin from nucleoplasm and converting it from a freely diffusive into a DNA-bound state. Our work raises the possibility that cohesin could arrive at an MCM site by loop extrusion, where it is either blocked, passes by or pauses. Extruding cohesin pausing at MCMs could potentially be converted into a topologically binding complex that establishes cohesion after passage of the DNA replication fork (Extended Data Fig. 10g). A similar conversion of DNA binding mode has recently been proposed at CTCF sites[51]. Our idea distinguishes cohesin loading by MCMs from the arrival of cohesin at MCM sites by loop extrusion.

Given the evolutionary conservation of MCMs, it is possible that replicative helicases might be ancestral barriers in species that lack CTCF-anchored loops, such as *Drosophila*, in which the establishment of TADs during embryonic development coincides with a switch in replication origin usage[52]. Finally, our data suggest that the 'MCM paradox' has consequences for chromatin organization and gene expression, which might have relevance for human pathologies such as Meier–Gorlin syndrome that are linked to mutations in the MCM loading pathway[53].

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

## Methods

### Animals

The mice used in this work were bred and maintained in agreement with the authorizing committee according to the Austrian Animal Welfare law and the guidelines of the International Guiding Principles for Biomedical Research Involving Animals (CIOMS, the Council for International Organizations of Medical Sciences). Mice were housed in individually ventilated cages under a 14-h light–10-h dark cycle at an ambient temperature of 22 °C ± 1 °C and humidity of 55% ± 5% with continuous access to food and water. Mice were housed in groups (maximum four males per cage and maximum five females per cage). All mice were bred in the IMBA animal facility. Wild-type, $Scc1^{fl/fl}$ and $Scc1^{Myc/+}$ mice were bred on a mixed background (B6, 129, Sv). $Wapl^{fl/fl}$ and $Zp3$-$dsCTCF$ mice were bred on a primarily C57BL/6J background. $Zp3$-$dsCTCF$ mice were maintained by breeding $Zp3$-$dsCTCF$ males with C57BL/6J females. Experimental $Scc1^{fl/fl}$ and $Wapl^{fl/fl}$ mice were obtained by mating of homozygous floxed females with homozygous floxed males carrying $Tg(Zp3Cre)$[54]. Experimental $Scc1^{Myc/+}$ mice were obtained by intercrossing heterozygous $Scc1^{Myc/+}$ mice. Experimental $Zp3$-$dsCTCF$ mice were maintained by breeding $Zp3$-$dsCTCF$ males with C57BL/6J females.

### Collection and in vitro culture of mouse oocytes

Ovaries were dissected from sexually mature female mice, which were euthanized by cervical dislocation. Fully grown germinal vesicle (GV) oocytes from 2–5-month-old females were isolated by physical disaggregation of ovaries with hypodermic needles. GV oocytes were cultured in M2 medium supplemented with 0.2 mM of the phosphodiesterase inhibitor 3-isobutyl-1-methylxanthine (IBMX, Sigma-Aldrich) at 37 °C. Mature oocytes were selected according to appearance (size, central nucleus, smooth zona pellucida) and cultured in M16 medium supplemented with IBMX in an incubator at 37 °C and 5% $CO_2$. Oocytes were cultivated in approximately 40-μl drops covered with paraffin oil (NidOil).

### Microinjection

GV oocytes were microinjected with in-vitro-transcribed mRNA dissolved in RNAse-free water (mMessage mMachine T3 kit, Ambion). The following mRNA concentrations were injected: 2.3 pmol hGeminin(L26A) and 0.2 pmol GFP. Microinjection was performed in approximately 20-μl drops of M2 (0.2 mM IBMX) covered with mineral oil (Sigma-Aldrich) using a Pneumatic PicoPump (World Precision Instruments) and hydraulic micromanipulator (Narishige) mounted onto a Zeiss Axiovert 200 microscope equipped with a 10×/0.3 EC plan-neofluar and 40×/0.6 LD Apochromat objective. Injected oocytes were cultured for 2 h and then released from IBMX inhibition by washing in M16 to resume meiosis.

### In vitro maturation and in vitro fertilization

Oocyte collection and culturing was performed as described above but M2 and M16 media were supplemented with 20% FBS (Gibco) and 6 mg ml$^{-1}$ fetuin (Sigma-Aldrich). After microinjection and IBMX release as described above, GV oocytes were subsequently incubated at 37 °C and in low-oxygen conditions (5% $CO_2$, 5% $O_2$, 90% $N_2$) to initiate in vitro maturation to metaphase II (MII) eggs. Next, MII eggs were in vitro fertilized 10.5–12 h after release of IBMX. Sperm was isolated from the cauda epididymis and vas deferens of stud males (2–5 months old) and capacitated in fertilization medium (Cook Austria GmbH) in a tilted cell culture dish for at least 30 min before incubation with MII eggs. For in vitro fertilization of wild-type, $Scc1^{fl/fl}$ $(Tg)Zp3$-Cre and $Zp3$-$dsCTCF$ oocytes, sperm was obtained from B6CBAF1 males, whereas sperm of C57BL/6J males was used for in vitro fertilization of $Wapl^{fl/fl}$ $(Tg)Zp3$-Cre oocytes. Zygotes were scored by the formation of visible pronuclei at 5 h after fertilization.

### In situ fixation, immunofluorescence staining and imaging

Zygotes were pulsed with 1 mM 5-ethynyl-2′-deoxyuridine (EdU) (Invitrogen) before in situ fixation to check the time frame of G1 phase. To check for DNA replication, zygotes were fixed in G2 after continuous incubation in the presence of EdU. Oocytes and zygotes were stripped from their zona pellucida by using acidic Tyrode's solution (Sigma-Aldrich) before in situ fixation in 4% paraformaldehyde (PFA) (in phosphate-buffered saline (PBS)) for 30 min at room temperature, followed by permeabilization in 0.2% Triton X-100 in PBS (PBSTX) for 30 min at room temperature. EdU-pulsed cells were processed according to the manual of the Click-iT EdU Alexa Fluor 647 imaging kit (Invitrogen). Blocking was performed using 10% goat serum (Dako) in PBSTX for 1 h at room temperature or at 4 °C overnight. Cells were incubated with primary antibodies for 2.5 h at room temperature or at 4 °C overnight. The following primary antibodies were used: anti-MCM2 (1:500; BD Transduction Laboratories, 610701), anti-CTCF (1:250, Peters Laboratory, A992), anti-MYC (1:500, Millipore, 05-724). After washing in blocking solution three times for at least 20 min, cells were incubated with goat anti-mouse Alexa Fluor 488 (1:500, Invitrogen, A11029), donkey anti-rabbit Alexa Fluor 568 (1:500, Invitrogen, A10042) or goat anti-mouse Alexa Fluor 647 (1:500, Invitrogen, A-21235) secondary antibodies for 1 h at room temperature. The excess of secondary antibody was removed by washing three times in 0.2% PBSTX for at least 20 min, which was followed by a short PBS wash and submerging for 20 min in Vectashield with DAPI (Vector Labs). Cells were mounted in Vectashield with DAPI using imaging spacers (Sigma-Aldrich) to preserve three-dimensional integrity. Detection of chromatin-bound MCM2 required pre-extraction before fixation and was performed as described previously[33]. In short, the zona pellucida was not removed and zygotes were incubated in ice-cold extraction buffer (50 mM NaCl, 3 mM $MgCl_2$, 300 mM sucrose, 25 mM HEPES, 0.5% Triton X-100) for 7 min on ice, followed by three short washes in ice-cold extraction buffer without Triton X-100. In situ fixation and immunofluorescence was performed as described above. To avoid zona pellucida collapse, cells were submerged in increasing Vectashield concentrations before final mounting. Image acquisition was performed on a Zeiss LSM780 or LSM880 confocal microscope using a plan-apochromat 63×/1.4 oil immersion objective. Image analysis was performed in Fiji/ImageJ. Mean intensity was measured within a defined nuclear area of each zygote. To measure nuclear signal over background, images were first deconvoluted by Huygens Professional (SVI) followed by segmentation into nuclei and surrounding cytoplasm using a custom ImageJ macro.

### Cell culture and synchronization

HCT116 cells were cultured as previously described[40]. In brief, cells were cultured in McCoy's 5A medium (Thermo Fisher Scientific) supplemented with 10% FBS (Gibco), 2 mM L-glutamine (Invitrogen) and 10% penicillin–streptomycin solution (Sigma-Aldrich). Cells were grown in an incubator at 37 °C with 5% $CO_2$. MCM2–mAID degradation was induced by addition of 500 μM 3-indoleacetic acid (Sigma-Aldrich) for 6 h. To synchronize cells in G1 for Hi-C analysis, a 2 mM thymidine arrest was followed by release into fresh medium for 6 h. Subsequently, nocodazole was added for 5 h, followed by shake-off of prometaphase cells and release in fresh medium for 4 h. Cells were fixed for Hi-C, microscopy and fluorescence-activated cell sorting (FACS). Cell-cycle profiling was performed using propidium iodide staining. For G1 FACS, cells were synchronized with a double-thymidine arrest–release followed by release into fresh medium for 12 h. Four hours before sorting, Hoechst 33342 (Sigma) was added to the medium at a concentration of 0.2 μg ml$^{-1}$. For the triptolide (Trp) experiment, 4 h before sorting, triptolide (Sigma) at 1 μM was added. Live-cell sorting was performed with the BD FACS Aria II flow cytometry instrument. The following gating strategy was used: gating for cells with SSC-A versus FSC-A, excluding doublets with FSC-H versus FSC-A, selecting Hoechst 33342-stained

cells with HOECHST-W versus HOECHST-A gating and Count versus HOECHST-A to select to select the G1 population. To avoid S-phase cell contamination, only cells in the left part of the G1 peak were collected (red dashed box in Extended Data Fig. 8) (see Supplementary Fig. 6 for the gating strategy).

## Chromatin fractionation and protein detection

Fractionation was performed as previous described[34]. In brief, cells were extracted in a buffer consisting of 20 mM Tris–HCl (pH 7.5), 100 mM NaCl, 5 mM MgCl, 2 mM NaF, 10% glycerol, 0.2% NP40, 20 mM β-glycerophosphate, 0.5 mM DTT and protease inhibitor cocktail (Complete EDTA-free, Roche). Chromatin pellets and supernatant were separated and collected by centrifugation at 1,700g for 5 min. The chromatin pellets were washed three times with the same buffer. Protein concentration was measured using a Bradford assay. Proteins were separated through SDS–PAGE on a Bolt 4–12% Bis-Tris Plus Gel (Invitrogen) and transferred to a nitrocellulose membrane. After overnight blocking with 5% skimmed milk in TBS-T at 4 °C, the membrane was incubated with primary antibodies for 2.5 h at room temperature. The following antibodies were used: anti-MCM2 (1:5,000; BD Transduction Laboratories, 610701), anti-MCM4 (1:5,000; Abcam, ab4459), anti-H3 (1:2,000; Cell Signaling, 97155), anti-GAPDH (1:2,500; Millipore, MAB374), anti-CTCF (1:1,000, Peters Laboratory, A992), anti-PCNA (1:500, Santa Cruz, PC10), anti-SCC1 (1:1,000, Millipore, 05-908) and anti-Pol II 8WG16 (1:500, Santa Cruz, sc-56767). Goat anti-mouse immunoglobulins–HRP (1:500, Dako, P0447) and goat anti-rabbit immunoglobulins–HRP (1:500, Dako, P0448) secondary antibodies were used to detect primary antibodies. Detection was performed using Immobilon Forte Western HRP Substrate (Merck) with a ChemiDoc imaging system (Bio-Rad).

## snHi-C

snHi-C was carried out as previously described[27,28,32,47]. Pronuclei of wild-type, *Scc1*[Δ/Δ], *Wapl*[Δ/Δ] and *Zp3-dsCTCF* zygotes were fixed around 1.5 h after visualization of pronuclei (corresponding to 6–6.5 h after fertilization) and therefore are expected to be in G1 phase of the cell cycle. No blinding or randomization was used for handling of the cells. In brief, isolated pronuclei were fixed in 2% PFA for 15 min, transferred to microwell plates (Sigma, M0815) and then lysed on ice in lysis buffer (10 mM Tris-HCl pH 8.0, 10 mM NaCl, 0.5% (v/v) NP-40 substitute (Sigma), 1% (v/v) Triton X-100 (Sigma), 1× Halt Protease Inhibitor Cocktail (Thermo Fisher Scientific)) for at least 30 min. After a brief PBS wash, the pronuclei were incubated in 1× NEB3 buffer (New England Biolabs) with 0.6% SDS at 37 °C for 2 h with shaking in a humidified atmosphere. The pronuclei were then washed once in 1× DpnII buffer (New England Biolabs) with 1× bovine serum albumin (BSA) (New England Biolabs) and further digested overnight with 5 U DpnII (New England Biolabs) at 37 °C in a humidified atmosphere. After a brief PBS wash and a wash through 1× ligation buffer (Thermo Fisher Scientific), the pronuclei were then ligated with 5 U T4 ligase (Thermo Fisher Scientific) at 16 °C for 4.5 h with rotation (50 rpm), followed by 30 min ligation at room temperature. Next, whole-genome amplification was performed using the illustra GenomiPhi V2 DNA amplification kit (GE Healthcare). In brief, the pronuclei were transferred to 0.2-ml PCR tubes in 3 µl sample buffer covered with mineral oil (Sigma-Aldrich) and were de-cross-linked at 65 °C overnight. Then, the pronuclei were lysed by adding 1.5 µl lysis solution (600 mM KOH, 10 mM EDTA, 100 mM DTT) and incubated for 10 min at 30 °C, followed by neutralization with the addition of 1.5 µl neutralization solution (4 vol 1 M Tris HCl, pH 8.0; 1 vol 3 M HCl). Whole-genome amplification was carried out by addition of 4 µl sample buffer, 9 µl reaction buffer and 1 µl enzyme mixture and incubation at 30 °C for 4 h followed by heat activation at 65 °C for 10 min. High-molecular-weight DNA was purified using AMPure XP beads (Beckman Coulter, 1.8:1.0 beads:DNA ratio) and 1 µg DNA was sonicated to approximately 300–1,300-bp fragments using the E220 Focused-Ultrasonicator (Covaris). The sonicated DNA

was purified with a PCR purification kit (Qiagen) and used to prepare Illumina libraries with the NEB Next Ultra Library Prep kit (Illumina). Libraries were sequenced on the HiSeq 2500 v4 with 125-bp paired-end reads (at the VBCF NGS unit) or on the NextSeq high-output lane with 75-bp paired-end reads (at the MPIB NGS core facility).

## snHi-C data analysis

snHi-C data were processed and analysed similarly to a previous report[28] and as previously described in[27,32,47]. In brief, the reads of each sample were mapped to the mm9 genome with bwa and processed by the pairtools framework (https://pairtools.readthedocs.io/en/latest/) into pairs files. These data were subsequently converted into COOL files by the cooler package and used a container for Hi-C contact maps.

Loops were analysed by summing up snHi-C contact frequencies for loop coordinates of over 12,000 loops identified using the Hi-C data from wild-type mouse embryonic fibroblasts published previously[32]. We removed the effect of distance dependence by averaging 20 × 20 matrices surrounding the loops and dividing the final result by similarly averaged control matrices. Control matrices were obtained by averaging 20 × 20 matrices centred on the locations of randomly shifted positions of known loops (shifts ranged from 100 to 1,100 kb with 100 shifts for each loop). For display and visual consistency with the loop strength quantification, we set the backgrounds levels of interaction to 1. The background is defined as the top left 6 × 6 and the bottom right 6 × 6 submatrices. To quantify the loop strength, the average signal in the middle 6 × 6 submatrix is divided by the average signal in the top left and bottom right (at the same distance from the main diagonal) 6 × 6 submatrices. Weighted statistics were calculated using the weights package in R (https://CRAN.R-project.org/package=weights).

For average TAD analysis, we used published TAD coordinates for the CH12-LX mouse cell line[3]. We averaged Hi-C maps of all TADs and their neighbouring regions, chosen to be of the same length as the TAD, after rescaling each TAD to a 90 × 90 matrix. For visualization, the contact probability of these matrices was rescaled to follow a shallow power law with distance (−0.25 scaling). TAD strength was quantified using contact probability normalized snHi-C data. In Python notation, if $M$ is the 90 × 90 TAD numpy array (where numpy is np) and $L = 90$ is the length of the matrix, then TAD_strength = box1/box2, where box1 = 0.5*np.sum(M[0:L//3, L//3:2*L//3]) + 0.5*np.sum(M[L//3:2*L//3, 2*L//3:L]); and box2 = np.sum(M[L//3:2*L//3, L//3:2*L//3]).

To calculate the insulation score, we computed the sum of read counts within a sliding 40-kb-by-40-kb diamond. The diamond was positioned such that the 'tip' touched the main axis of the snHi-C map corresponding to a 'self-interaction'. As snHi-C maps are not iteratively corrected, we normalized all insulation profiles by the score of the minimum insulation and then subtracted 1. This way, the insulation/domain boundary is at 0 and has a minimum of 0.

Contact probability $P_c(s)$ curves were computed from 10-kb binned snHi-C data. We divided the linear genomic separations into logarithmic bins with a factor of 1.3. Data within these log-spaced bins (at distance, $s$) were averaged to produce the value of $P_c(s)$. Both $P_c(s)$ curves and their log-space slopes are shown following a Gaussian smoothing (using the scipy.ndimage.filters.gaussian_smoothing1d function with radius 0.8). Both the $y$ axis (that is, log($P_c(s)$) and the $x$ axis (that is, log[$s$]) were smoothed. The average loop size was determined by studying the derivative of the $P_c(s)$ curve in log–log space; that is, the slope of log($P_c(s)$). The location of the maximum of the derivative curve (that is, the position of the smallest slope) closely matches the average length of extruded loops.

## Hi-C library preparation and sequencing

Hi-C was performed largely as described previously[3] with minor modifications. In brief, around 5 × 10[6] HCT116 cells were cross-linked in 1% formaldehyde for 10 min at room temperature, snap-frozen and stored at −80 °C. After permeabilization in lysis buffer (0.2% Igepal, 10 mM

Tris-HCl pH 8.0, 10 mM NaCl, 1× Halt Protease inhibitor cocktail) nuclei were isolated in 0.3% SDS in NEBuffer 3 at 62 °C for 10 min. SDS was quenched with 1% Triton X-100 at 37 °C for 1 h, then the nuclei were pelleted and resuspended in 250 μl DpnII buffer with 600 U DpnII (New England Biolabs) at 37 °C. After overnight digestion, 200 U DpnII was added followed by 2 h more incubation. Then, nuclei were spun down and resuspended in fill-in mix (biotin-14-dATP (Thermo Fisher Scientific), dCTP, dGTP and dTTP (Thermo Fisher Scientific), Klenow Polymerase (NEB), 1× NEB 2 buffer) for 1.5 h at 37 °C with rotation. After ligation at room temperature for 4 h with T4 ligase (NEB), the nuclei were pelleted, resuspended in 200 μl $H_2O$ and digested with proteinase K for 30 min at 55 °C in the presence of 1% SDS. NaCl was added to a final concentration of 1.85 M before cross-links were reversed at 65 °C overnight. After ethanol precipitation and a 70%–80% ethanol wash, DNA was resuspended in 10 mM Tris EDTA, transferred to a Covaris microtube (Covaris) and sheared to approximately 300–1,300-bp fragments on the E220 Focused-Ultrasonicator (Covaris). DNA was then bound to Dynabeads MyOne Streptavidin C1 beads (Thermo Fisher Scientific) for biotin pull-down. Beads were resuspended in $H_2O$ used for library preparation with the NEBNext Ultra II Library Prep kit for Illumina (NEB). Beads were then washed four times using Tween wash buffer (5 mM Tris-HCl, 1 M NaCl, 0.5 mM EDTA, 0.05% Tween20) and DNA was eluted using 95% formamide, 10 mM EDTA at 65 °C for 2 min. After precipitation, DNA was washed with 70–80% ethanol and resuspended in $H_2O$. The finished libraries were sequenced on the NovaSeq 6000 system (Illumina) with 100-bp paired-end reads (at the VBCF NGS unit) or on the NextSeq high-output lane (Illumina) with 75-bp paired-end reads (at the MPIB NGS core facility).

## Micro-C library preparation and sequencing

The Micro-C libraries were prepared using the Dovetail Micro-C Kit following the manufacturer's protocol. In brief, the chromatin was fixed with disuccinimidyl glutarate (DSG) and formaldehyde in the nucleus. The cross-linked chromatin was then digested in situ using micrococcal nuclease (MNase). After digestion, the cells were lysed with SDS to extract the chromatin fragments and the chromatin fragments were bound to chromatin capture beads. Next, the chromatin ends were repaired and ligated to a biotinylated bridge adapter followed by proximity ligation of adapter-containing ends. After proximity ligation, the cross-links were reversed, the associated proteins were degraded and the purified DNA was converted into a sequencing library using Illumina-compatible adaptors. Biotinylated molecules were pulled down on streptavidin beads before PCR amplification. The library was sequenced on the NextSeq high-output lane (Illumina) with 75-bp paired-end reads (at the MPIB NGS core facility).

## Hi-C and Micro-C data analysis

Hi-C and Micro-C data processing was performed using distiller—a nextflow-based pipeline (https://github.com/open2c/distiller-nf)[55]. Reads were mapped to the hg38 reference genome with default settings except dedup/max_mismatch_bp=0. Multiresolution cooler files[56] generated by distiller were used for visualization in HiGlass[57] and in the downstream analyses.

For downstream analysis, we used quaich (https://github.com/open2c/quaich), a new snakemake pipeline for Hi-C postprocessing. It uses cooltools (https://github.com/open2c/cooltools)[58], chromosight[59] and coolpup.py[60] to perform compartment and insulation analysis, peak annotation and pileups, respectively. The config file we used is available here: https://gist.github.com/Phlya/5c2d0688610ebc5236d5aa7d0fd58adb.

We annotated peaks of enriched contact frequency in untreated HCT116 cells from a previous report[61] using chromosight at 5 kb resolution with default parameters. Then we used this annotation to quantify the strength of Hi-C peaks in our datasets using pileups at 5 kb resolution. Similarly, valleys of insulation score at 10 kb resolution with a window of 500 kb (and prominence over 0.1) were identified in the same published dataset and filtered to remove those that don't disappear after cohesin depletion (or don't become at least fivefold weaker) to identify cohesin-dependent domain boundaries. These were used to quantify changes in insulation in our datasets. Neighbouring insulation valleys were joined together to form TADs; regions longer than 1.5 Mb were ignored. TAD coordinates were used for rescaled pileup analysis[28] to quantify their strength in our datasets. De novo peaks were called using Mustache[62].

To investigate whether the increase in loop strength occurs genome wide, we split all loop calls into 1 Mb bins, using the coordinate of the centre of the loops. Then for each bin, we created pileups normalized to the global chromosome arm-wide expected level of interactions, using coolpuppy at 5 kb resolution with 100 kb flanks. In addition, each pileup (105 × 105 kb) was normalized to the mean value of the top left and bottom right 3 × 3 pixels, to remove variability in local background between different regions of the genome. Then the mean of the central 3 × 3 square of the pileup was used as the measure of normalized loop strength for this bin. Having done this for both MCM2-depleted and control cells, we plotted the result as a histogram of $\log_2$ ratio between the two, to investigate whether the overall distribution of scores is shifted between the two conditions.

## RNA sequencing (RNA-seq) of G1 zygotes

For each replicate, a pool of 10 G1 zygotes were lysed, total RNA was extracted and cDNA was synthesized using the SMART-Seq v4 Ultra Low Input RNA Kit (Takara Bio Europe). Sequencing libraries were prepared with the Nextera XT DNA Library Preparation Kit for Illumina. Libraries were sequenced on the HiSeq 2500 v4 (Illumina) with 50-bp single-end reads at the VBCF NGS unit.

## RNA-seq of tissue culture cells

Total RNA from HCT116 cells was isolated using a lysis step based on guanidine thiocyanate (adapted from a previous study[63] and using magnetic beads (GE Healthcare, 65152105050450). mRNA sequencing libraries were prepared from 1 μg total RNA using NEBNext Poly(A) mRNA Magnetic Isolation Module (E7490) and NEBNext Ultra II Directional RNA Library Prep Kit for Illumina (E7760). Paired-end sequencing was performed on Illumina NextSeq 500 (2 × 43-bp reads). A total of six samples were multiplexed and sequenced on a NextSeq 500/550 High Output Kit v2.5 (75 Cycles) at the MPIB NGS core facility. BCL raw data were converted to FASTQ data and demultiplexed by bcl2fastq.

## RNA-seq analysis

FASTQ files from sequencing mouse G1 zygotes or the human HCT116 cell line were pseudoaligned to the mm10 or hg38 releases of the *Mus musculus* or *Homo sapiens* genomes, respectively, using Kallisto with 100 bootstraps[64]. The resulting abundance measures were analysed in R to generate PCA plots[65] (factoextra) and a heat map of the correlation matrix (heatmap.2)[66]. To find differentially expressed transcripts we used the Wald test for Sleuth model (sleuth) in R. Gene ontology (GO) term enrichment of molecular functions of up- and downregulated genes were carried out using ShinyGO (http://bioinformatics.sdstate.edu/go/).

The changes in the chromatin contact frequencies that occurred upon MCM depletion around the TSS of differentially expressed (DE), non-differentially expressed (non-DE) and non-expressed genes were analysed by aggregating the number of contacts as determined in Micro-C experiments with 5 kb resolution. The number of contacts was normalized with LOESS using HICcompare in R, and ensemble analysis of the four expression categories (upregulated, $n = 164$; downregulated, $n = 65$; non-DE, $n = 916$; non-expressed, $n = 1,000$) was carried out in distance bins of 0–5 kb, 5–25 kb, 25–250 kb, 250–1,000 kb and over 1,000 kb up- and downstream of the TSS. The mean change of contact frequencies in each bin for every category was calculated by averaging the auxin versus DMSO treatment ratios of the normalized sum of

contacts. All of the mean contact frequency changes were tested against the non-DE TSS control using the non-parametric Kruskal–Wallis test followed by pairwise Wilcoxon (Mann–Whitney *U*) test.

All plots were compiled with ggplot2 in R.

## Protein expression and purification

**Cohesin.** Human recombinant cohesin[STAG1, SCC1-Halo] was purified and fluorescently labelled with Janelia Fluor 549 HaloTag (Promega) as previously described[6].

**ORC and Cdc6.** *Saccharomyces cerevisiae* recombinant ORC and Cdc6 were purified as previously described[67].

**SFP synthase.** SFP synthase was purified essentially as previously described[68].

**Cdt1–MCM and Cdt1–MCM[Mcm3-YDF].** To generate fluorescently labelled *S. cerevisiae* recombinant Cdt1–MCM, the *S. cerevisiae* strain ySA4 was generated. In brief, a ybbR and 3×Flag tag were fused to the N and C terminus of Mcm6, respectively, generating Cdt1–MCM[ybbR-Mcm6]. The chimeric MCM complex containing a humanized Mcm3 subunit (Cdt1–MCM[Mcm3-YDF, ybbR-Mcm6]) was expressed in strain yMS1, which was generated by further modification of ySA4. For this, the corresponding region in *S. cerevisiae* Mcm3 was replaced by the 19-amino-acid disordered region that contains a YDF motif present in human MCM3[44], using CRISPR–Cas9-based genome editing essentially as previously described[69]. To target *S. cerevisiae* Mcm3, the following guide sequence was used: 5′-TATAATGTCACCGCTTCCTG-3′. The homologous repair template (synthesized by Eurofins Genomics) encoding the 19-amino-acid disordered region containing the YDF motif (underlined) was: 5′-ACTCCAAGAAGGTCAACGGCATCTTCCGTTAATGCCACGCC ATCGTCAGCACGCAGAATATTACGTTTTCAAGATGACGAACAGAACGCT GGTGAAGAC<u>GATGGGGATTCATACGACCCCTATGACTTCAGTGACACA GAGGAGGAAATGCCCTCAAA</u>GGCTTCAACTGGGGTTGAGAGTGTCTCC AAGACGTAGAGAACATCTTCACGCACCTGAGGAAGGTTCGTCGGGACCT CTTACCGAGGTCGGTACTCCA-3′. Notably, this strategy allowed the modification of all Mcm3 alleles (confirmed by sequencing) and thus ensured the complete absence of wild-type Mcm3 in the subsequent preparation. Strain yMS1 grew comparably to the parental strain ySA4, confirming that the YDF motif did not alter the MCM function.

Cells were grown in 6 l YP medium supplemented with 2% (v/v) raffinose at 30 °C. At an optical density at 600 nm ($OD_{600\,nm}$) of 1.2, cells were arrested at G1 by adding α-factor to a final concentration of 150 ng ml$^{-1}$ for 3 h. Subsequently, protein expression was induced by the addition of 2 % (v/v) galactose. After 4 h, cells were collected and washed once with cold MilliQ water + 0.3 mM PMSF and once with buffer A (100 mM HEPES-KOH, pH 7.6, 0.8 M sorbitol, 10 mM Mg(OAc)$_2$, 0.75 M potassium glutamate (KGlu)). Finally, cells were resuspended in 1 packed cell volume of buffer A + 1 mM DTT supplemented with a protease inhibitor cocktail (2 μM pepstatin, 2 μM leupeptin, 1 mM PMSF, 1 mM benzamidine, 1 μg ml$^{-1}$ aprotinin) and frozen dropwise in liquid N$_2$. Frozen cells were lysed in a freezer mill (SPEX) and lysed cell powder was resuspended in 1 packed cell volume buffer B (45 mM HEPES-KOH, pH 7.6, 0.02 % (v/v) Nonidet P40 Substitute, 5 mM Mg(OAc)$_2$, 10 % (v/v) glycerol, 1 mM ATP, 1 mM DTT) + 300 mM KGlu. All subsequent purification steps were performed at 4 °C unless stated otherwise. The lysate was cleared by ultracentrifugation at 235,000*g* for 60 min. Soluble lysate was incubated with 0.5 ml bed volume (BV) Anti-Flag M2 affinity gel (Sigma) equilibrated with buffer B + 300 mM KGlu for 3 h. The resin was washed twice with 20 BV buffer B + 300 mM KGlu and twice with 20 BV buffer B + 100 mM KGlu. Protein was eluted with buffer B + 100 mM KGlu + 0.5 mg ml$^{-1}$ 3×Flag peptide.

For site-specific labelling, Cdt1-MCM[ybbR-Mcm6] or Cdt1-MCM[Mcm3-YDF, ybbR-Mcm6] was incubated with SFP-Synthase and LD655-CoA (Lumidyne Technologies) at a 1:3:6 molar ratio for 2 h at 30 °C in buffer B + 100 mM

KGlu, 10 mM MgCl$_2$. Labelled protein was further purified on a Superdex 200 increase 10/300 gel filtration column (GE Healthcare) equilibrated in buffer B + 100 mM potassium acetate (KOAc). Protein-containing fractions were pooled, concentrated with a MWCO 50000 Amicon Ultra Centrifugal Filter unit (Merck) and stored in aliquots at −80 °C. The labelling efficiency was estimated to be around 90% from the extinction coefficients of Cdt1-MCM and LD655.

## Single-molecule imaging

Single-molecule assays were performed using an RM21 micromirror TIRF microscope (Mad City Labs) built in a similar manner to that previously described[70] with an Apo N TIRF 60× oil-immersion TIRF objective (NA 1.49, Olympus). Janelia Fluor 532 and LD655 were excited with a 532 nm and 637 nm laser (OBIS 532 nm LS 120 mW and OBIS 637 nm LX 100 mW, Coherent), respectively at a frame rate of around 6 fps. Residual scattered light from excitation was removed with a ZET532/640m emission filter (Chroma). Emission light was split at 635 nm (T635lpxr, Chroma) and recorded as dual-view with an iXon Ultra 888 EMCCD camera (Andor). All microscope parts were controlled using Micromanager v1.4 (ref. [71]) and custom Beanshell scripts.

## Preparation of PEG–biotin microscope slides

Glass coverslips (22 × 22 mm, Marienfeld) were cleaned in a plasma cleaner (Zepto, Diener Electronic) and subsequently incubated in 2% (v/v) 3-aminopropyltriethoxysilane (Roth) in acetone for 5 min. Silanized coverslips were washed with ddH$_2$O, dried and incubated at 110 °C for 30 min. Slides were covered with a fresh solution of 0.1 M NaHCO$_3$ containing 0.4% (w/v) biotin–PEG-SC-5000 and 15% (w/v) mPEG-SC-5000 (Laysan Bio) and incubated overnight. Functionalized slides were washed with ddH$_2$O, dried and incubated again overnight in a fresh biotin–PEG/mPEG solution. Slides were finally washed, dried and stored under vacuum.

## DNA substrate for single-molecule imaging

To generate pMSuperCos-ARS1, first, a 21 kb genomic DNA fragment of bacteriophage lambda (NEB) was flanked by a unique XbaI (position 0) and NotI restriction site on either end and cloned into a pSuperCos1 backbone (Stratagene). Second, the yeast origin ARS1 was inserted at a BamHI site around position 5.3 kb within the 21 kb genomic DNA fragment.

To produce the DNA substrate for single-molecule imaging, pMSuperCos-ARS1 was isolated from DH5α using a Plasmid Maxi Kit (Qiagen). One hundred micrograms of plasmid was digested with 100 U NotI-HF and XbaI (NEB) for 7 h at 37 °C. The resulting 21,202 bp ARS1-DNA fragment was separated from the SuperCos1 backbone on a 10–40 % sucrose gradient. DNA handles were prepared by annealing oligonucleotides MS_200/201 MS202/203 (see Supplementary Table 2 for oligonucleotide sequences) in equimolar amounts in 30 mM HEPES, pH 7.5, 100 mM KOAc by heating to 95 °C for 5 min and cooling to 4 °C at −1 °C per min. Annealed handles were mixed with the purified 21 kb ARS1-DNA at a molar ratio of 15:1 and ligated with T4 DNA Ligase in 1× T4 ligase buffer (both NEB) at 16 °C overnight. Free handles were removed on a Sephacryl S-1000 SF Tricorn 10/300 gel filtration column (GE Healthcare) equilibrated in 10 mM Tris, pH 8, 300 mM NaCl, 1 mM EDTA. Peak fractions were pooled, ethanol precipitated and reconstituted in TE buffer. Final DNA was stored in aliquots at −80 °C. Note that the final linear DNA is functionalized with biotin at a NotI site and an 18-bp single-stranded DNA overhang at an XbaI site that is used for orientation specific doubly tethering.

## Flow cell preparation

A functionalized PEG–biotin slide was incubated with blocking buffer (20 mM Tris-HCl, pH 7.5, 50 mM NaCl, 2 mM EDTA, 0.2 mg ml$^{-1}$ BSA, 0.025 % (v/v) Tween20) + 0.2 mg ml$^{-1}$ streptavidin (Sigma) for 30 min. A flow cell was assembled by placing a polydimethylsiloxane block

on top to generate a 0.5 mm wide and 0.1 mm high flow channel and a polyethylene tube (inner diameter 0.58 mm) was inserted at either end.

DNA was introduced to the flow cell at 5 pM in blocking buffer and incubated for 15 min in the absence of buffer flow to allow binding to the slide surface. To doubly tether DNA, the flow lane was flushed with 100 μM oligonucleotide MS_204 (see Supplementary Table 2 for oligonucleotide sequences) in blocking buffer at 100 μl per min.

## Single-molecule sliding assay

Helicase loading was achieved by introducing 0.25 nM ORC, 4 nM Cdc6 and 10 nM Cdt1–MCM[ybbR-LD655-Mcm6] or Cdt1–MCM[Mcm3-YDF, ybbR-LD655-Mcm6] in licensing buffer (30 mM HEPES-KOH, pH 7.6, 8 mM Mg(OAc)$_2$, 0.1 mg ml$^{-1}$ BSA, 0.05% (v/v) Tween20) + 200 mM KOAc, 5 mM DTT, 3 mM ATP to a prepared flow cell and incubating for 25 min. Cohesin loading and sliding was essentially performed as previously described[72]. Cohesin[STAG1, SCC1-Halo-JF546] (0.7 nM) was incubated with licensed DNA in cohesin binding buffer (35 mM Tris, pH 7.5, 25 mM NaCl, 25 mM KCl, 1 mM MgCl$_2$, 10% (v/v) glycerol, 0.1 mg ml$^{-1}$ BSA, 0.003 (v/v) Tween20, 1 mM DTT, 0.2 mM ATP) for 10 min. To remove free protein, DNA-bound licensing factors and MCM loading intermediates, the flow cell was washed with licensing buffer + 500 mM NaCl, 1 mM DTT, 0.6 mM ATP supplemented with an oxygen scavenging system (1 mM Trolox, 2.5 mM PCA, 0.21 U ml$^{-1}$ PCD (all Sigma))[73]. Imaging was either started directly (high-salt condition) or after lowering the salt concentration to 150 mM NaCl (physiological salt condition) in an otherwise identical buffer to that described for the high-salt condition. DNA was post-stained with 50 nM SYTOX Orange (Thermo Fisher Scientific) in the same buffer that was used during imaging.

## Single-molecule data analysis

Single-molecule data were analysed in Fiji using the Molecule Archive Suite (Mars) plug-in (https://github.com/duderstadt-lab/)[74] and custom Python scripts. In brief, all doubly tethered DNA molecules containing cohesin were chosen for analysis. Cohesin and MCM were tracked individually and merged with DNA to determine their position on the same DNA molecule. Pauses during cohesin translocation were determined by fitting cohesin trajectories (position on DNA versus time) with the kinetic change point algorithm[75] with the following settings: confidence value 0.6; global sigma 300 base pairs (bps)/s. Subsequently, resulting segments with rates lower than 200 bps per s, standard deviations of less than 30 bps per s and length greater than 1 s were classified as pause segments. If two adjacent segments were classified as pauses and the end and start position on DNA of the first and second pause segment, respectively, were within 1 kb, these segments were merged to one pause segment. The fraction of cohesin pausing reported was determined by calculating the cumulative time of all pause segments divided by the total observation time. These pauses were excluded when calculating cohesin–MCM passing probabilities and diffusion coefficients (see below).

The probability of cohesin passing MCM was addressed as follows: Frames in which cohesin colocalized with MCM (median position) within less than thresh1 were classified as encounter. Upon an encounter, if cohesin passed MCM in the consecutive frame by at least thresh2, the encounter was determined as successful bypassing. All remaining frames (distance > thresh1 to MCM) were further evaluated for MCM passing as described above, and in addition counted as an encounter with successful bypassing. DNA molecules with cohesin only were analysed the same way using the theoretical ARS1 position on DNA. All frames within the cohesin trajectory that were part of a translocation pause were excluded from this analysis and instead classified as one encounter with failed bypassing. To account for different resolution at different extensions, two dynamic thresholds, thresh1 and thresh2, were set to 1.5 kb and 0.5 kb at the mean DNA extension of all DNA molecules and adjusted for the individual length of the DNA molecule (Extended Data Fig. 9g).

MCM photobleaching steps were defined as abrupt drops in fluorescence intensity and detected using the kinetic change point algorithm[75].

Diffusion coefficients ($D$) were calculated with:

$$D = \frac{<x>^2}{2t},$$

in which $<x>^2$ is the mean square displacement in kb$^2$ and $t$ is the time in s.

All kymographs were generated using Fiji. For this, individual DNA ends were fitted with subpixel localization and the kymograph was generated along the connecting line. Individual DNA molecules doubly tethered with different extension to the slide surface and as a consequence, kymographs differ in pixel heights. These length differences were accounted for throughout all of the analysis steps described above.

## Loop extrusion simulations and contact map generation

**Simulations overview.** We introduced MCMs into polymer models of loop extrusion[11] (Fig. 3a), as randomly located extrusion barriers. Both CTCF and MCM barriers stall cohesin with some probability (CTCF 50%; ref. [38]) but allow bypassing, consistent with single-molecule experiments (Fig. 4d). By sweep parameters (processivity and linear density of cohesin, and density and permeability of MCM; Supplementary Figs. 2–5), we found a narrow range of values for each condition such that the peak strengths and paternal $P_c(s)$ curves can be simultaneously reproduced (Fig. 3b, d, Extended Data Fig. 6e–h, Supplementary Figs. 2–5). The simulations suggest that in wild-type conditions, cohesins extrude 110–130-kb loops and have a density of around 1 per 300 kb. MCM permeability was essential to achieve the increase in peak strength without strongly affecting the average loop size after MCM loss; in this regime, there is a linear trade-off between the MCM density and permeability (Fig. 3c). Using MCM densities (one per 30–150 kb) experimentally measured in other cell types (see below), cohesins should bypass MCMs in around 60–90% of encounters.

**Time steps and lattice set-up.** We use a fixed-time-step Monte Carlo algorithm as in previous work[39]. We define the chromosome as a lattice of $L = 10,000$ sites, in which each lattice site corresponds to 2 kb of DNA. Loop extruding factors (LEFs) are represented as two motor subunits, which move bidirectionally away from one another one lattice site at a time. When LEFs encounter one another, we assume that they cannot bypass each other as is typical for cohesin simulations[76]. The ends of the chromosome (that is, the first and last lattice sites) are considered boundaries to LEF translocation; this way, LEFs cannot 'walk off' the chromosome.

**CTCF and MCM boundary elements.** To simulate TADs, we specify that every 150th lattice site is a CTCF site. In this way, our simulated 20 Mb chromosome segment is composed of 66 TADs each of size 300 kb. CTCF sites may stall the translocation of a LEF subunit with a probability of 0.45. This stalling probability is chosen within the experimental estimates of 15%–50% fractional occupancy of CTCF sites via ChIP–seq and microscopy[38]. For simulations mimicking the 'control' and 'Wapl' depletion conditions (that is, where MCM is present on the genome), we also add random extrusion barriers to our lattice to mimic the presence of MCMs. For our parameter sweep, we add 33, 66, 132, 264, 528 barriers (that is, representing MCMs) randomly dispersed in the 20 Mb chromosome segment; this corresponds to a density of 1 MCM complex per 600 kb, 300 kb, 150 kb, 75 kb, 37.5 kb, respectively. The MCM barriers are fixed in place for the duration of a simulation. Like the CTCFs, the MCM barriers can also stall LEF translocation. A randomly translocating LEF subunit will be stalled at an MCM site with a probability of 0.0001, 0.05, 0.2, 0.4 or 0.8 (meaning that LEFs can bypass between around 20–100% of MCM sites). For both CTCF and MCM lattice sites, 'stalling' a LEF subunit is a permanent event that prevents further movement of that subunit. Stalling events are only resolved after dissociation of the LEF from the lattice. For simulations in which there is 'MCM loss',

we set the total number of random MCM barriers to zero but keep the CTCF lattice sites the same. All results presented in this paper are from an average over 25 different random distributions of MCMs (that is, 25 simulation runs were performed for each condition).

**LEF separations and processivity.** For our simulations of 'control' and 'MCM-loss' conditions, the default LEF processivity was 90 kb, and the default LEF separation was 300 kb. For our simulations of the '$Wapl^\Delta$' and '$Wapl^\Delta$ + MCM loss' conditions, the LEF processivity was 130 kb, and the separations were 180 kb. The approximately 50% increase in density after Wapl depletion is supported by quantitative immunofluorescence data indicating there is a modest enrichment of cohesin after removal of Wapl[37].

**Association and dissociation rates.** All simulations are performed with fixed numbers of extruders. The dissociation rate is ultimately tied to the 'processivity' of the LEF, which is the average distance in kb (or lattice sites) that the LEF travels before dissociating. We allow LEFs to randomly associate to at any lattice position after a dissociation event.

**Loop extrusion equilibration steps.** We compute 10,000 initialization steps for each simulation before creating any contact maps. This ensures that the loop statistics have reached a steady-state. Subsequent loop configurations were sampled every 100 simulation steps to generate contact maps. We sampled from at least 2,500 different LEF configurations (that is, 100 configurations from 25 different simulations) to generate contact probability decay curves and perform aggregate peak analysis (see below).

**Contact maps.** We generated contact maps semi-analytically, which uses a Gaussian approximation to calculate contact probability maps directly from the positions of LEFs. This approach was developed previously[39] and used to simulate bacterial Hi-C maps. We note that as the density of cohesins is sufficiently low in the zygotes (that is, the processivity and separation ratio is close to or less than 1), and as the contact probability scaling exponent up to 10 Mb is close to −1.5 in the absence of cohesins[27], we are justified in using the Gaussian approximation to generate contact maps. To generate the $P_c(s)$ curves, we use at least 9,000,000 random samples of the contact probability; these samples were taken from varying genomic positions and relative separations within the simulated 20 Mb of chromosome and averaged using logarithmically spaced bins (factor of 1.3). To generate the equivalent of the aggregate peak analysis for contact enrichments at CTCF sites, we used at least 144,000,000 random samples of the contact probability from a 100 kb by 100 kb window centred on the CTCF sites. These 144,000,000 samples were distributed evenly between 64 TADs (there are 66 TADs, but we excluded the 2 TADs closest to the chromosome ends) and at least 2,500 LEF conformations. Control matrices for normalization were obtained as described above, but using a shifted window shifted by 150 kb from the TAD boundaries. Aggregate peak analysis plots are shown coarse-grained to 20 × 20 bins.

**Comparing simulated and experimental data.** The criteria for comparing the experimental data and the simulated data were two-fold. First, we computed from snHi-C the corner peak strength above background; this was usually a number between 1 and 3 depending on the condition. Second, we computed the $P(s)$ curves from experiments genome wide. However, we knew from previous studies[27,34], that the effect of cohesin on $P(s)$ typically only extends up to around 1 Mb under normal conditions. Moreover, above 1 Mb, the semi-analytical approach to generating contact maps becomes less reliable as non-equilibrium effects, chain topology, and chain swelling may start to have a role in the $P(s)$ curve, which are not accounted for in our model[39]. Below 30 kb, Hi-C data have been shown to contain artefacts and can vary significantly between different protocols. Thus, we restricted our comparisons to the range 30 kb–1 Mb.

The criteria then for evaluating the goodness of a simulation, were to (1) obtain quantitative values for the corner peak strengths as close as possible to the experiments, preserving the correct relative ordering between various conditions (for example, in paternal zygotes, the corner peak strength from weakest to highest was: wild type, Wapl depletion, MCM depletion, MCM + Wapl depletion). We directly scored the goodness of the simulation by minimizing the absolute error between the simulated and experimental corner peak strengths. (2) Simultaneously, we evaluated the absolute values and shapes of the $P(s)$ curves between 30 kb–1 Mb. The goodness of $P(s)$ fit was evaluated by visual agreement. Therefore, we used a combined approach to evaluate the match between experiments and simulations, in which the dot strength and $P(s)$ curves were evaluated together.

**Estimation of chromatin-bound MCM density in mammalian cells.** Using mass-spectrometry analysis, the copy number of each MCM subunit is estimated at around 670,000 in HeLa cells[77], and quantitative immunoblotting shows that in late G1 phase around 45% of MCM2 is bound to chromatin[78]. This leads to the estimate that around 301,500 MCMs are bound to the chromatin in late G1. Knowing that MCMs form double hexamers on chromatin and that the average genome size of HeLa cells is around $7.9 \times 10^9$ (ref. [79]), we estimate a density of 1 MCM double hexamer every approximately 52 kb ($7.9 \times 10^9/(301,500/2)$) (assuming a random distribution of MCMs).

**Reporting summary**

Further information on research design is available in the Nature Research Reporting Summary linked to this paper.

## Data availability

All sequencing data in support of the findings of this study have been deposited in the Gene Expression Omnibus (GEO) under the series accession numbers GSE196497 (snHi-C and RNA-seq) and GSE155971 (Hi-C and Micro-C). The single-molecule video datasets supporting the findings in this study have been deposited at Zenodo with the following: https://doi.org/10.5281/zenodo.5911106 (high-salt experiments), https://doi.org/10.5281/zenodo.5911210 (physiological salt experiments) and https://doi.org/10.5281/zenodo.5911284 (YDF experiments). All data are also available from the authors upon request.

## Code availability

The snHi-C and single-molecule processing scripts are deposited at Zenodo under https://doi.org/10.5281/zenodo.5906351 and https://doi.org/10.5281/zenodo.5911644, respectively. The simulation codes are available at https://github.com/mirnylab/MCMs-as-random-barriers-to-loop-extrusion-paper.

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

**Acknowledgements** We thank M. S. Bartolomei for the *(Tg)Zp3-dsCTCF* mouse strain; M. T. Kanemaki for the *MCM2-mAID* HCT116 cell line; the D. Remus laboratory for generating the ySA4 MCM strain that was used in vitro single-molecule assays; Z. Lygerou for sharing the geminin(L26A)-GFP plasmid; U. Cramer and R. Hornberger for technical assistance in generating the MCM3–YDF yeast strains; and H. C. Theußl and J. R. Wojciechowski for reviving *(Tg)Zp3-dsCTCF* embryos. Illumina sequencing was performed at the VBCF NGS unit and the MPIB NGS core facility (http://www.vbcf.ac.at and https://www.biochem.mpg.de/7076201/NGS). B.J.H.D. is funded by the PhD program (DK) Chromosome Dynamics (W1238-B20), supported by the Austrian Science Fund (FWF). L.A.M. and H.B.B. acknowledge support from the National Institutes of Health Common Fund 4D Nucleome Program (DK107980) and the Human Frontier Science Program (HFSP RGP0057/2018). I.M.F. is supported by a Medical Research Council UK University Unit grant (MC_UU_00007/2). Research in the K.E.D. laboratory is supported by the European Research Council (ERC; ERC-StG-804098 ReplisomeBypass), the German Research Foundation (DFG, SFB863-111166240) and the Max Planck Society. Research in the J.-M.P. laboratory is supported by Boehringer Ingelheim, the Austrian Research Promotion Agency (Headquarter grant FFG-878286), the ERC (grant agreement no. 693949 and 101020558), HFSP (RGP0057/2018) and the Vienna Science and Technology Fund (grant LS19-029). Research in the K.T. laboratory is supported by the ERC (ERC-CoG-818556 TotipotentZygotChrom), HFSP (RGP0057/2018), the Herzfelder Foundation and FWF (P30613-B21), the Austrian Academy of Sciences and the Max Planck Society.

**Author contributions** K.T. conceived the project. B.J.H.D. supervised by K.T. performed snHi-C, Hi-C, Micro-C, RNA-seq and imaging experiments. H.B.B. developed and performed snHi-C data analysis and polymer simulations with L.A.M. M.J.S. supervised by K.E.D. performed single-molecule imaging and analysis. J.G. performed snHi-C and imaging experiments. S.P. performed snHi-C data analysis and RNA-seq analysis. I.G. performed RNA-seq analysis and contributed to imaging analysis. A.L. performed Micro-C experiments. J.G., S.P., I.G. and A.L. were supervised by K.T. I.M.F. performed Hi-C and Micro-C analysis. I.F.D. purified recombinant cohesin^STAG1, SCC1-Halo. W.T. contributed to development of cell lines and R.S. to bioinformatic analyses. I.F.D., W.T. and R.S. were supervised by J.-M.P. B.J.H.D., H.B.B., M.J.S., I.M.F. and I.G. prepared the figures. B.J.H.D., H.B.B., M.J.S., I.G., K.E.D., L.A.M. and K.T. wrote the manuscript with input from all authors.

**Funding Open** access funding provided by Max Planck Society.

**Competing interests** The authors declare no competing interests.

**Additional information**
**Correspondence and requests for materials** should be addressed to Karl E. Duderstadt, Leonid A. Mirny or Kikuë Tachibana.

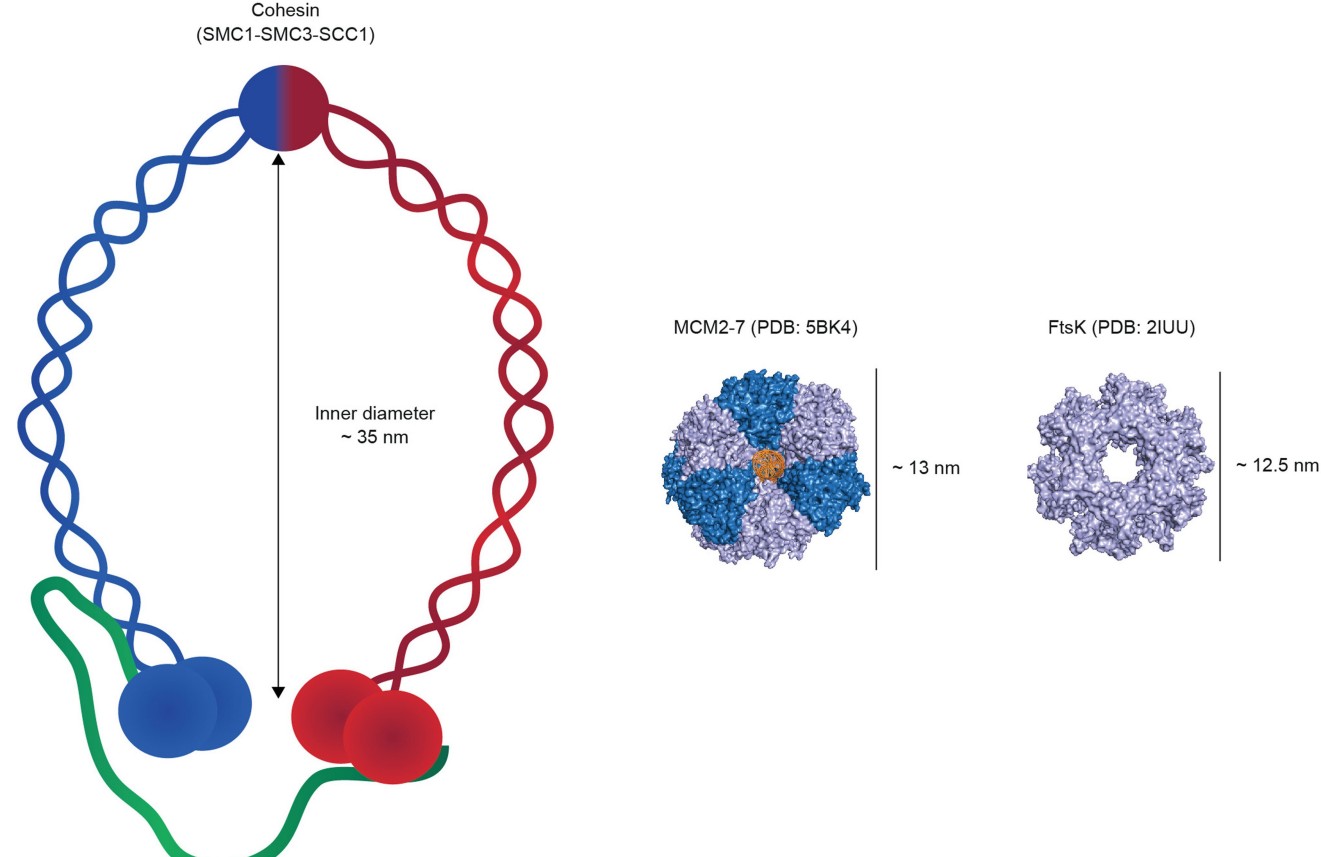

**Extended Data Fig. 1 | Dimensions of cohesin compared to the estimated size of the MCM2–7 complex and the FtsK roadblock.** Size comparison between a schematic representation of the ring-shaped heterotrimeric cohesin (SMC1-SMC3-SCC1) with the MCM2–7 double hexamer and the FtsK monohexamer. The size estimation of MCM2–7 and FtsK was based on their crystal structure. PDB accessions codes for each protein are shown in parentheses.

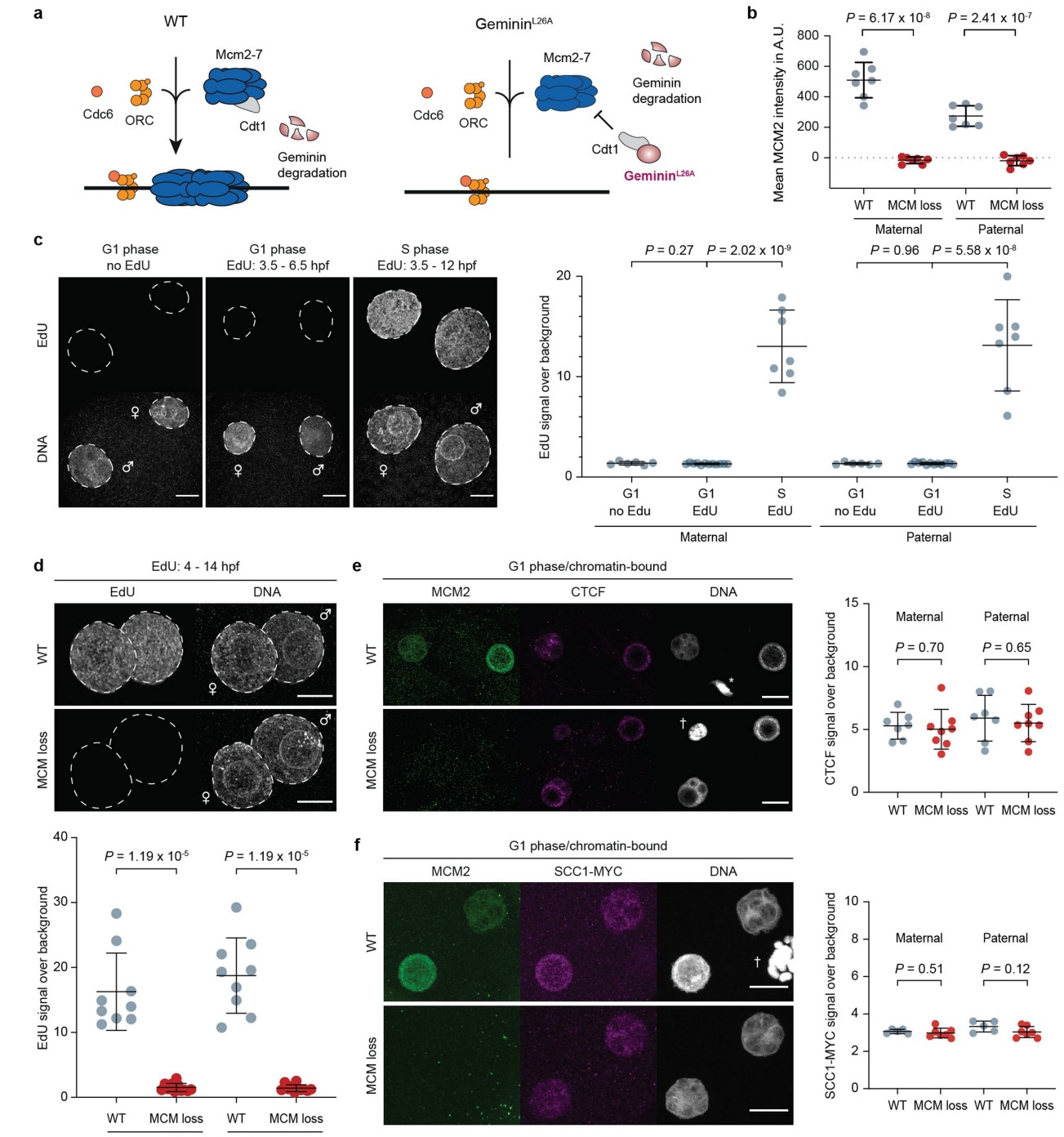

**Extended Data Fig. 2 | Prevention of MCM loading in G1-synchronized zygotes. a**, Expression of geminin(L26A), a non-degradable variant of geminin, prevents the recruitment of MCMs to chromatin in G1 phase by inhibiting the Cdt1-mediated loading pathway. **b**, Quantification of mean chromatin-bound MCM2 intensity in maternal and paternal pronuclei from wild type (WT, n = 7) and MCM-loading inhibited (MCM loss, n = 7) G1-phase zygotes from 1 experiment using 4 females. Representative image is shown in Fig. 1b. **c**, 5-ethynyl-2′-deoxyuridine (EdU)-cultured zygotes collected at 6.5h post-fertilization do not incorporate EdU (0/12) and therefore have not yet entered S phase. Zygotes without EdU in the medium (no EdU) also don´t incorporate EdU (0/6), while zygotes fixed at 12h post-fertilization show EdU-incorporation (7/7). Quantification of EdU intensity as signal over background shown for both maternal and paternal pronuclei from G1/no Edu (n = 6), G1/EdU (n = 12) and S/EdU (n = 7) zygotes examined in 1 experiment

using 6 females. **d**, Immunofluorescence analysis of WT (n = 9) and MCM loss (n = 11) zygotes from 1 experiment using 6 females that were cultured in continuous presence of EdU and collected at 14h post-fertilization (G2 phase). Top, representative image. Bottom, quantification of EdU signal over background in maternal and paternal pronuclei. **e**, **f**, Immunofluorescence analysis of chromatin-bound CTCF (**e**) and SCC1-MYC (**f**) in WT (n = 7 for CTCF, n = 5 for SCC1-MYC) and MCM loss (n = 8 for CTCF, n = 7 for SCC1-MYC) G1-phase zygotes from 1 experiment for each staining using 5 females each. Left, representative image. Right, Quantification of CTCF and SCC1-MYC intensity as signal over background in maternal and paternal pronuclei. *P* values were determined by two-sided unpaired *t*-test (**b**, **c**, **e**, **f**) or by two-sided Mann-Whitney *U* test (**d**). Quantifications in panels b-f are depicted as mean ± s.d.; *, sperm head. †, degraded polar body. DNA is stained with DAPI. Scale bars, 10 μm.

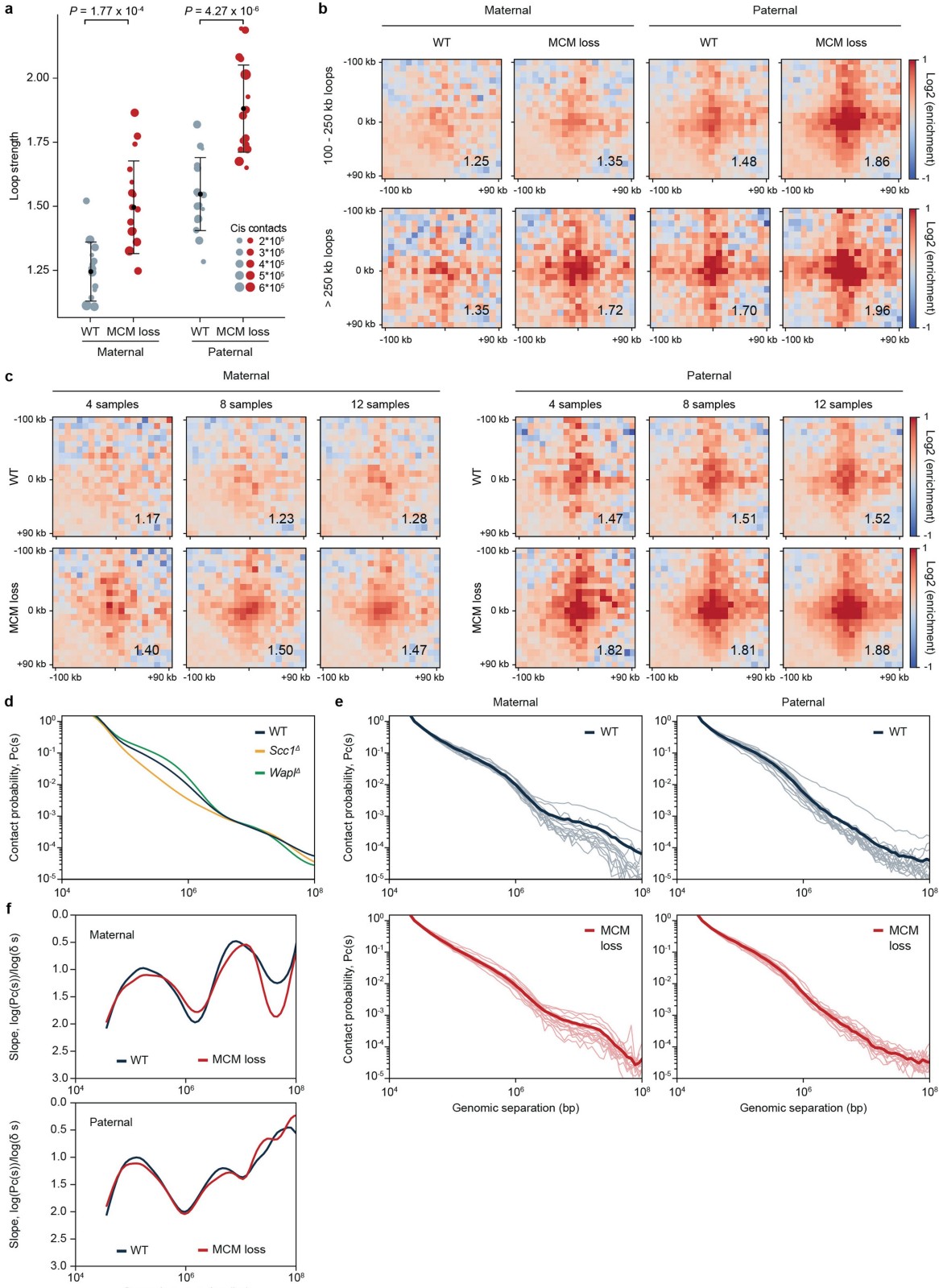

**Extended Data Fig. 3** | See next page for caption.

**Extended Data Fig. 3 | MCMs impede loops in G1 zygotes while having little effect on the $P_c(s)$ curve in the range of cohesin-dependent contacts up to 1 Mb. a**, Peak strength of individual samples for wild type (WT) and MCM-loading inhibited (MCM loss) conditions, shown for maternal and paternal pronuclei. Size of the bullets corresponds to the number of cis 1kb+ contacts per sample (weights). Data are based on n(WT, maternal) = 13, n(WT, paternal) = 16, n(MCM loss, maternal) = 16, n(MCM loss, paternal) = 15, from 4 independent experiments using 4-6 females for each experiment (same samples as in Fig. 1d). Data presented as weighted mean ± s.d.; *P* values were calculated using weighted statistics (Methods). **b**, Aggregate peak analysis for intermediate (100-250kb) and long loops (>250kb) in WT and MCM loss conditions for maternal and paternal pronuclei. Data are based on the same samples as in (**a**) and Fig. 1d. **c**, Aggregate peak analysis for WT and MCM loss conditions from a subset of 4, 8 and 12 samples, shown for maternal and paternal pronuclei. **d**, Contact probability $P_c(s)$ curve as a function of genomic distance (s). Cohesin is directly involved in shaping the $P_c(s)$ in the range up to 1 Mb. The contact frequency in this region is decreased after cohesin depletion (*Scc1*[4]) and is increased after enrichment of chromatin-bound cohesin (*Wapl*[4]). **e**, Contact probability $P_c(s)$ curves from individual maternal and paternal pronuclei with average $P_c(s)$ (same as in Fig. 1e) in bold overlaid. **f**, Slopes of the $P_c(s)$ curves (depicted in Fig. 1e) as an indication for the average size of cohesin-extruded loops in WT and MCM loss conditions.

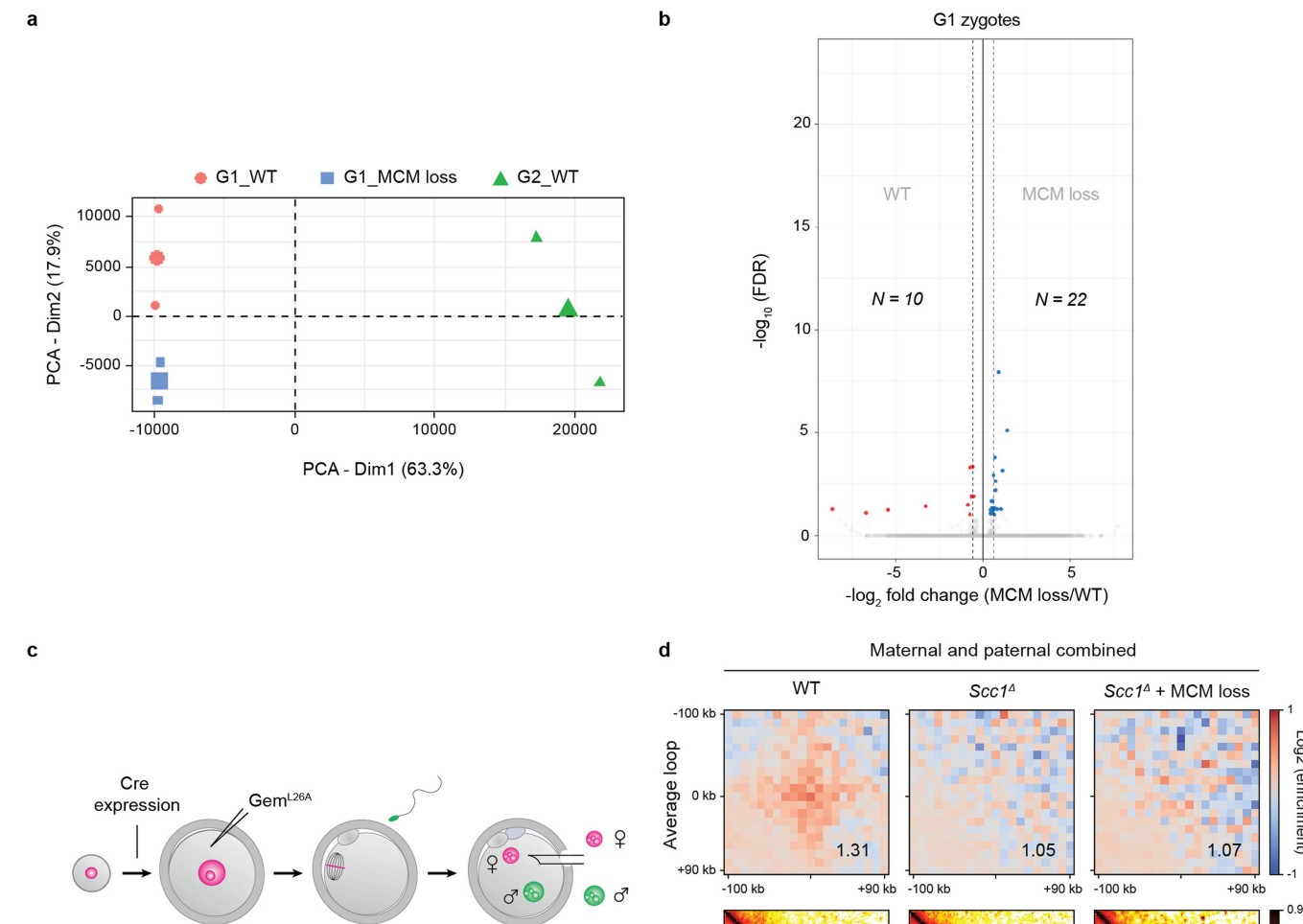

**Extended Data Fig. 4 | MCMs impede cohesin-dependent loops and TADs largely independently of transcriptional changes. a**, Principal component analysis (PCA) of the transcriptomes of G1 wild type (G1_WT), G1 MCM-loading inhibited (G1_MCM loss) and G2 wild type (G2_WT) zygotes. **b**, Volcano plot showing statistical significance $-\log_{10}$ (FDR) versus fold change ($\log_2$ fold change) for RNA-Seq data between WT and MCM loss conditions in G1 zygotes. Numbers indicate the number of transcripts significantly up- (right) or downregulated (left) after MCM loss at FDR = 0.1. Dashed vertical lines indicate $-0.585$ and $+0.585$ $\log_2$ fold change in expression (1.5-fold decrease and increase in expression), respectively. **c**, $Scc1^{\Delta/\Delta}$ oocytes from $Scc1^{fl/fl}$ *(Tg)Zp3*-Cre

females were injected with geminin(L26A) mRNA and eggs were fertilized to generate maternal knockout $Scc1^{\Delta(m)/+(p)}$ zygotes for snHi-C analysis in G1 phase. Because most proteins are provided by the oocyte and the G1-phase zygote is practically transcriptionally inactive, these maternal knockout zygotes are depleted for Scc1[33]. **d**, Aggregate peak and TAD analysis for $Scc1^{fl}$ (WT), $Scc1^{\Delta}$ and $Scc1^{\Delta}$ + MCM loss. Maternal and paternal data are shown pooled together. Data are based on n($Scc1^{fl}$) = 26, n($Scc1^{\Delta}$) = 42, and n($Scc1^{\Delta}$ + MCM loss) = 10 nuclei. Heat maps were normalized to an equal number of *cis* contacts. Control ($Scc1^{fl}$) samples and 38 Scc1-depleted samples ($Scc1^{\Delta}$) were previously published[27].

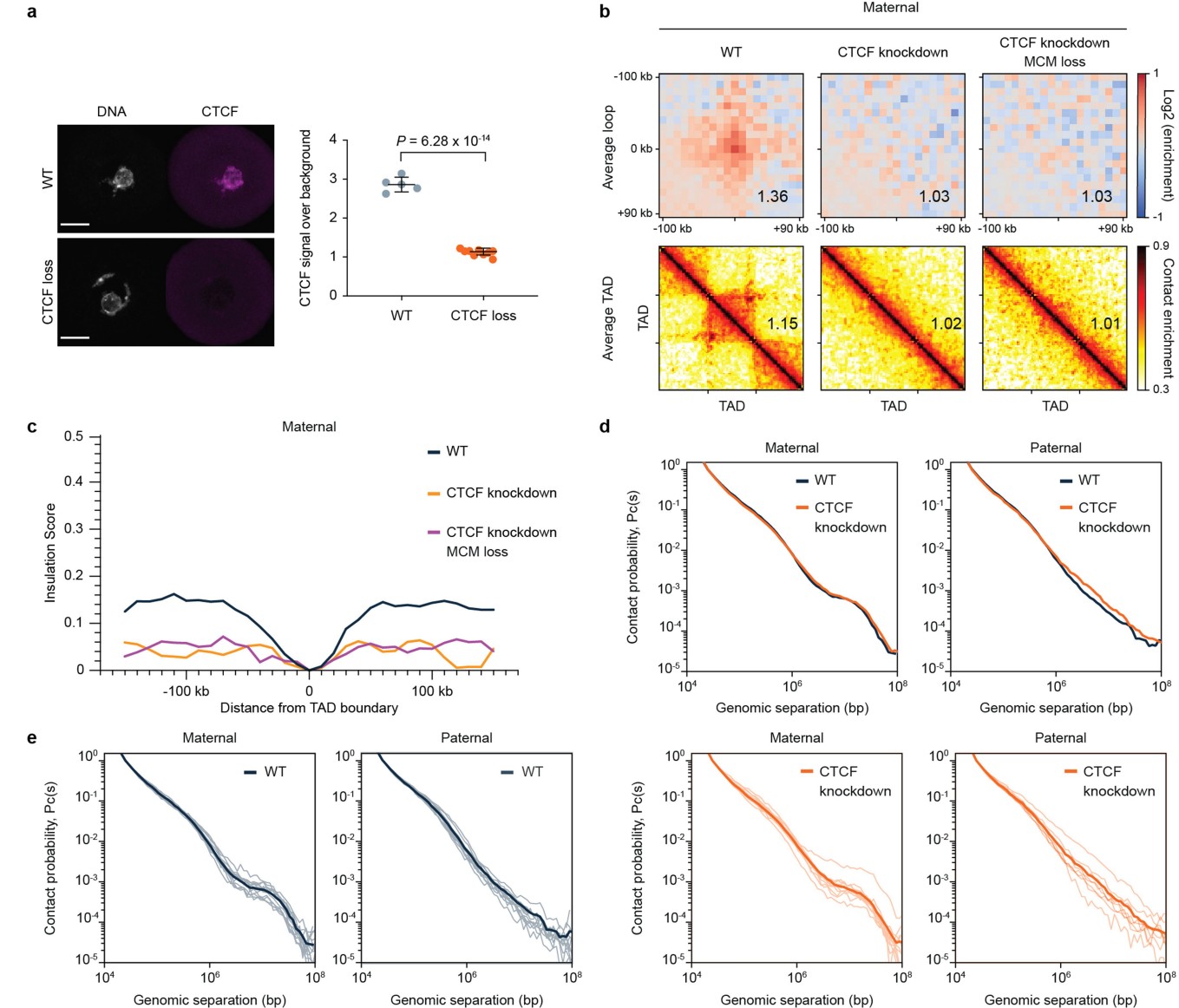

**Extended Data Fig. 5 | CTCF is required for loop and TAD formation in G1 zygotes. a**, Immunofluorescence analysis of CTCF in wild type (n = 5) and CTCF-depleted (n = 12) surrounded nucleolus (SN) oocytes from 1 experiment using 2 females for each genotype. Left, representative images, DNA is stained with DAPI. Scale bars: 20 μm. Right, Quantification of CTCF intensity as signal over background. Data are presented as mean ± s.d.; *P* value was determined by two-sided unpaired *t*-test. **b**, Aggregate peak and TAD analysis for wild type (WT), CTCF-depleted (CTCF knockdown) and CTCF-depleted combined with prevention of MCM loading (CTCF knockdown + MCM loss) maternal chromatin in G1 zygotes. Data shown are based on n(WT, maternal) = 12, n(CTCF knockdown, maternal) = 12, n(CTCF knockdown + MCM loss, maternal) = 10 nuclei, from 4 independent experiments using 4-6 females for each genotype. Heat maps were normalized to an equal number of *cis* contacts. **c**, Insulation scores at TAD borders for maternal nuclei. **d**, Average contact probability $P_c(s)$ curves for WT and CTCF knockdown conditions, shown separately for maternal and paternal nuclei. **e**, $P_c(s)$ curves for individual samples (maternal and paternal pronuclei) with average $P_c(s)$ (same as in **d**) in bold overlaid.

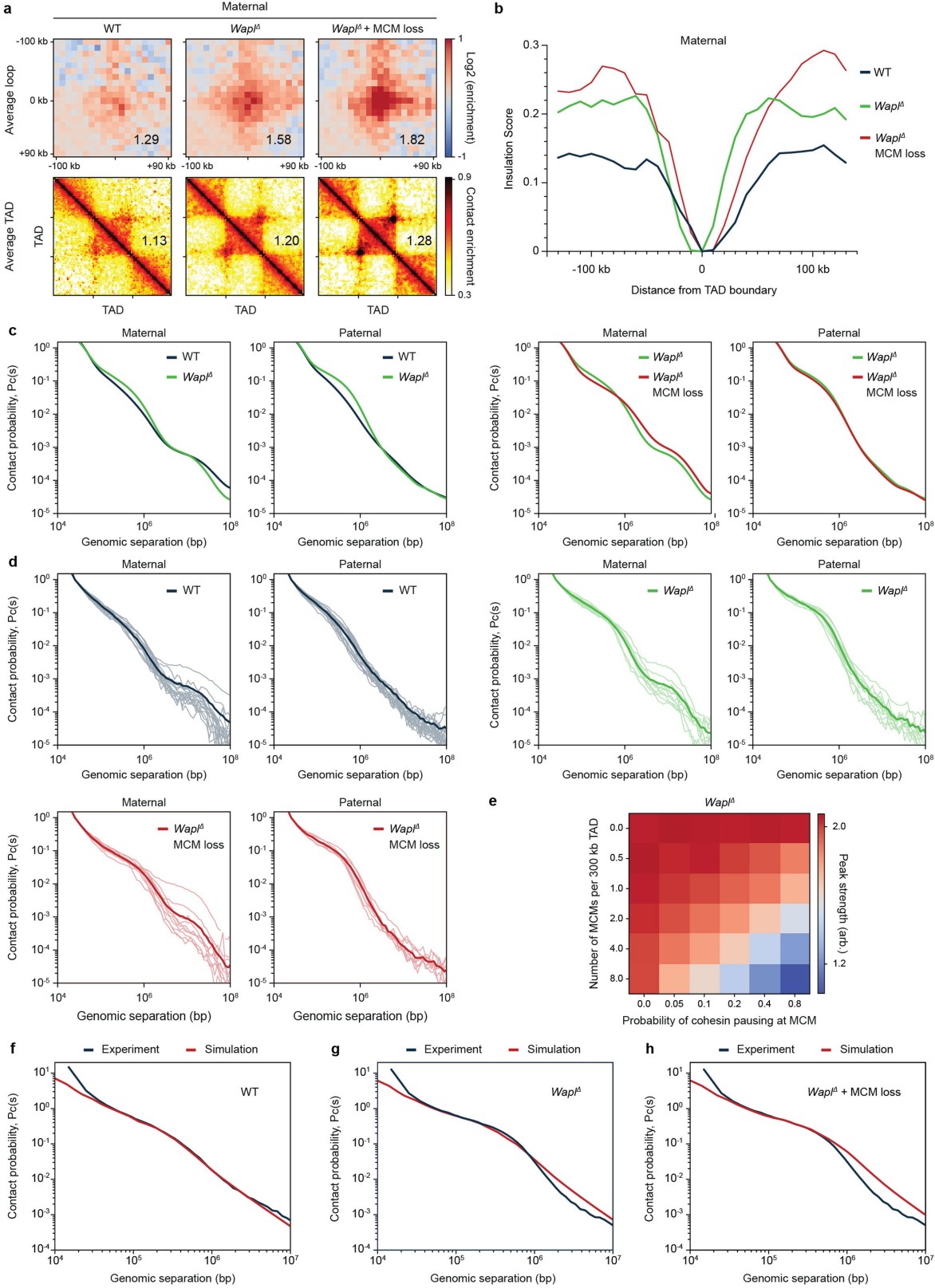

**Extended Data Fig. 6 | MCM restricts loop and TAD formation in maternal chromatin independently of Wapl-mediated cohesin release. a**, Aggregate peak and TAD analysis for control (WT), *Wapl^Δ* and *Wapl^Δ* + MCM loss for maternal chromatin in G1 zygotes. Data shown are based on n(WT, maternal) = 17, n(*Wapl^Δ*, maternal) = 10, n(*Wapl^Δ* + MCM loss, maternal) = 10, from 4 independent experiments using 4-6 females for each genotype. Control samples are WT (this study) pooled with *Wapl^fl* samples (published in[27]). **b**, Insulation scores at TAD borders for maternal nuclei. **c**, Average contact probability $P_c(s)$ curves for control (WT), *Wapl^Δ* and *Wapl^Δ* + MCM loss, shown separately for maternal and paternal nuclei. **d**, $P_c(s)$ curves for individual samples (maternal and paternal pronuclei) with average $P_c(s)$ (same as in panel c) in bold overlaid. **e**, Matrix of peak strengths, generated by polymer simulations, showing a linear trade-off between the MCM density and its ability to pause cohesins in *Wapl^Δ*. **f–g**, Simulated contact probability decay curve $P_c(s)$ for WT (**f**), *Wapl^Δ* (**g**) and *Wapl^Δ* + MCM loss (**h**). The simulated $P_c(s)$ curve is well matched with the experimental data.

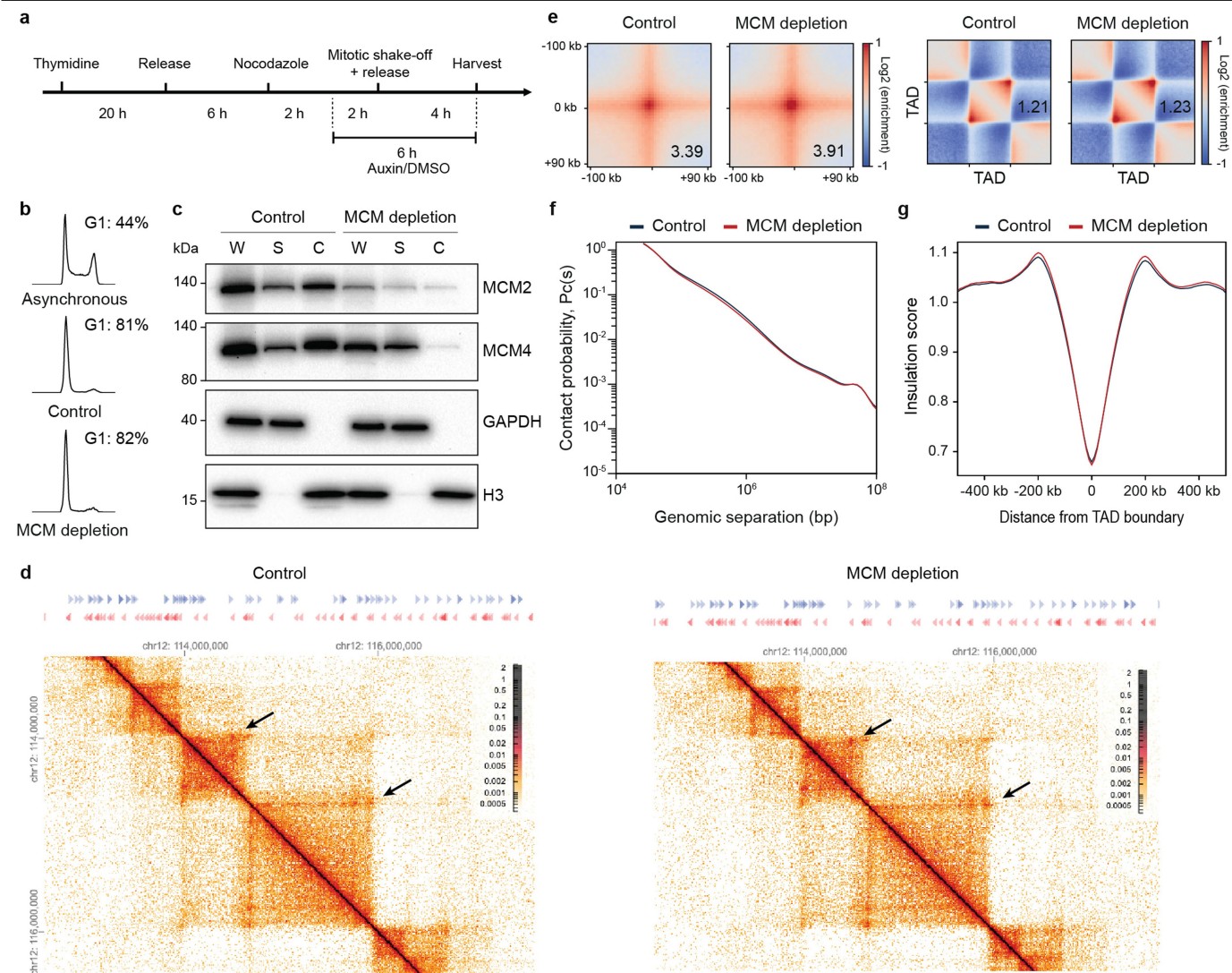

**Extended Data Fig. 7 | Moderate increase in aggregate peak strength after acute depletion of MCM in HCT116 cells using Hi-C. a**, Schematic for G1 synchronization of HCT116 MCM2-mAID cells. **b**, Cell cycle profiles of asynchronous and G1-synchronized (Control/MCM depletion) HCT116 MCM2-mAID cells. **c**, Immunoblotting analysis of whole-cell lysate (W), supernatant (S) and chromatin (C) fraction for MCM2, MCM4, GAPDH and H3 from G1-synchronized HCT116 MCM2-mAID cells treated with DMSO (Control) or auxin (MCM depletion). GAPDH and H3 are used as loading controls for supernatant and chromatin fraction, respectively. Uncropped blots are displayed in Supplementary Fig. 1. This experiment was repeated

independently three more times with similar results. **d**, Hi-C contact matrices for control and MCM depletion conditions for the region 112,5-117,6 Mb on chromosome 12 at 10 kb resolution. Increased corner peaks are denoted with an arrow. CTCF sites are depicted above the contact matrices. **e**, Average of total contact frequency for loops and TADs in aggregate peak and TAD analysis for control and MCM-depleted cells. **f**, Contact probability $P_c(s)$ curves for control and MCM depletion conditions. **g**, Insulation scores at TAD borders for control and MCM-depleted cells. Read statistics for Hi-C replicates can be found in Supplementary Table 1.

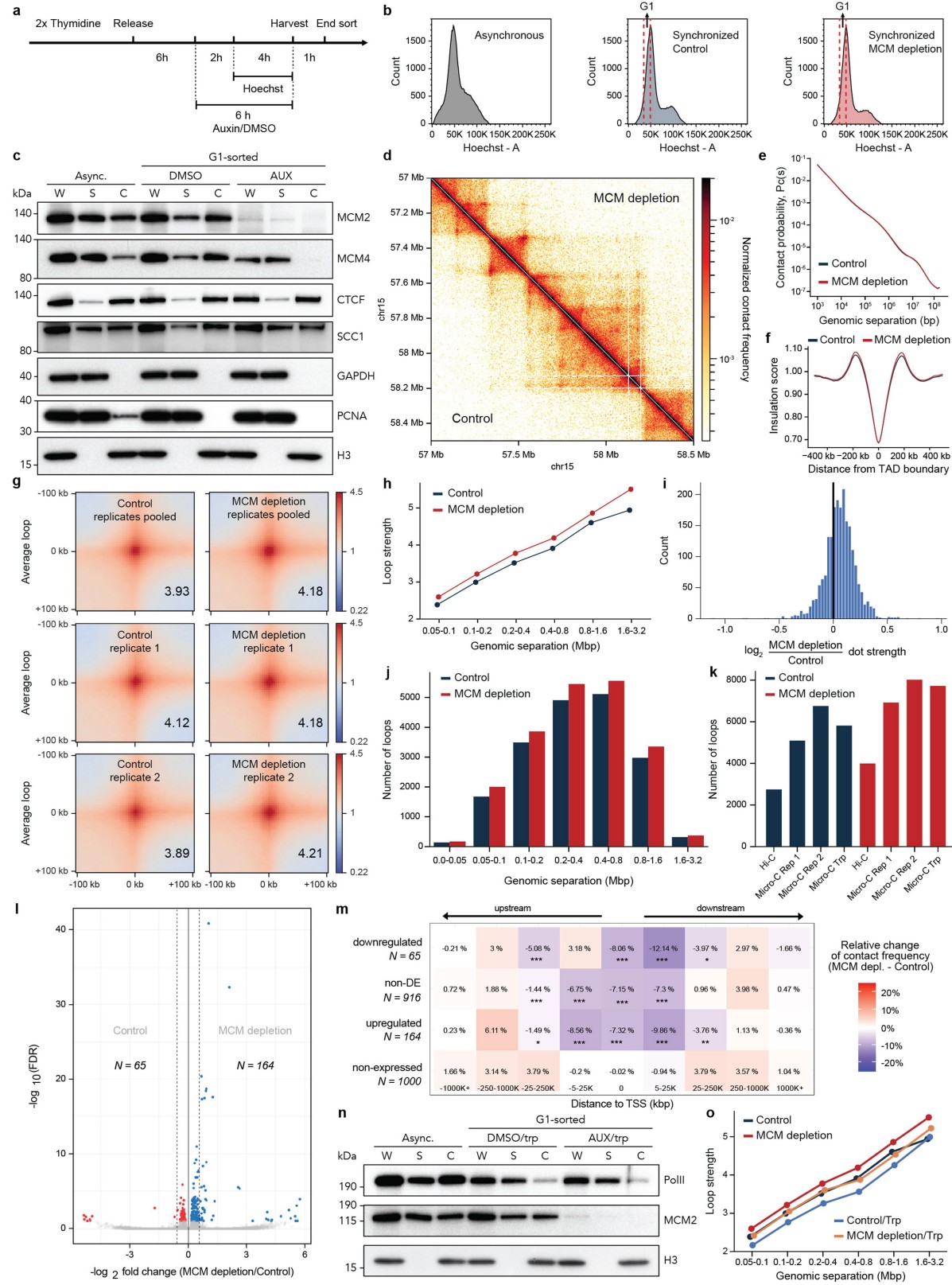

**Extended Data Fig. 8** | See next page for caption.

**Extended Data Fig. 8 | Micro-C reveals a genome-wide increase in peak strength and in de novo loop number after acute depletion of MCM2 in HCT116 cells. a**, Schematic for G1 synchronization of HCT116 MCM2-mAID cells before G1 FACs sorting. **b**, FACS-analysis of synchronized HCT116 MCM2-mAID cells. Only cells in the left part of the G1 peak were sorted and collected for Micro-C and RNA-seq to avoid contamination of S-phase cells (red dashed box). **c**, Immunoblotting analysis of whole-cell lysate (W), supernatant (S) and chromatin fraction (C) for MCM2, MCM4, CTCF, SCC1, PCNA, GAPDH and H3 from asynchronous and G1-sorted HCT116 MCM2-mAID cells treated with DMSO (Control) or auxin (MCM depletion). GAPDH and H3 are used as loading controls for the supernatant and chromatin fraction, respectively. Uncropped blots are displayed in Supplementary Fig. 1. This experiment was repeated independently one more time with similar results. **d**, Micro-C contact matrices for the region 57 - 58.5 Mb on chromosome 15 at 10 kb resolution in control vs MCM-depleted cells. **e**, $P_c(s)$ curves for control and MCM depletion conditions. **f**, Insulation scores at TAD borders for control and MCM-depleted cells. **g**, Average of the total contact frequency of loops in an aggregate peak analysis for two independent Micro-C replicates (middle and lower panel) and pooled dataset (upper panel) in control and MCM-depleted cells. **h**, Peak strengths for control and MCM-depleted cells over a range of genomic distances. **i**, Histogram showing the distribution of $\log_2$ ratio of peak strengths in MCM-depleted and control cells within 1 Mb bins across the whole genome, normalized to global and local background of interactions. Higher values indicate increase of peak strength after MCM depletion. Mean of the distribution is highly significantly different from 0 (one sample $t$-test), $P = 1.87 \times 10^{-70}$. **j**, de novo called loops using Mustache over a range of genomic distances in control and MCM-depleted cells. **k**, Number of de novo loops

(called with Mustache) in independent Hi-C and Micro-C experiments. All replicates were downsampled to 300 million total contacts. **l**, Volcano plot showing statistical significance $-\log_{10}$ (FDR) versus fold change ($\log_2$ fold change) for RNA-seq data between MCM2-mAID expressing HCT116 cells treated with DMSO (Control) or auxin (MCM depletion). Numbers indicate the number of transcripts significantly up- (right) or downregulated (left) after MCM depletion at FDR = 0.1. Dashed vertical lines indicate $-0.585$ and $+0.585$ $\log_2$ fold change in expression (1.5-fold decrease and increase in expression), respectively. RNA-seq libraries were generated in triplicate (independent replicates). **m**, Correlation between gene expression changes and relative change in chromatin contact frequencies around the transcriptional start sites (TSSs) after MCM loss. All mean contact frequency changes were tested against the non-DE TSS control using the non-parametric Kruskal-Wallis test followed by pairwise Wilcoxon (Mann-Whitney $U$) test. **n**, Immunoblotting analysis of whole-cell lysate (W), supernatant (S) and chromatin fraction (C) from asynchronous and G1-sorted HCT116 MCM2-mAID cells treated with triptolide (chromatin-bound RNA PolII degradation) and either DMSO (Control) or auxin (MCM depletion). H3 is used as loading control for the chromatin fraction. Uncropped blots are displayed in Supplementary Fig. 1. This experiment was repeated independently one more time with similar results. **o**, Peak strengths for control, MCM-depleted, triptolide-treated (DMSO/trp) and triptolide-treated/MCM-depleted (MCM depletion/trp) cells over a range of genomic distances. Notes: Micro-C datasets (control/MCM depletion) are pooled replicates from two biologically independent Micro-C experiments, unless otherwise stated. Read statistics for Hi-C and Micro-C replicates can be found in Supplementary Table 1.

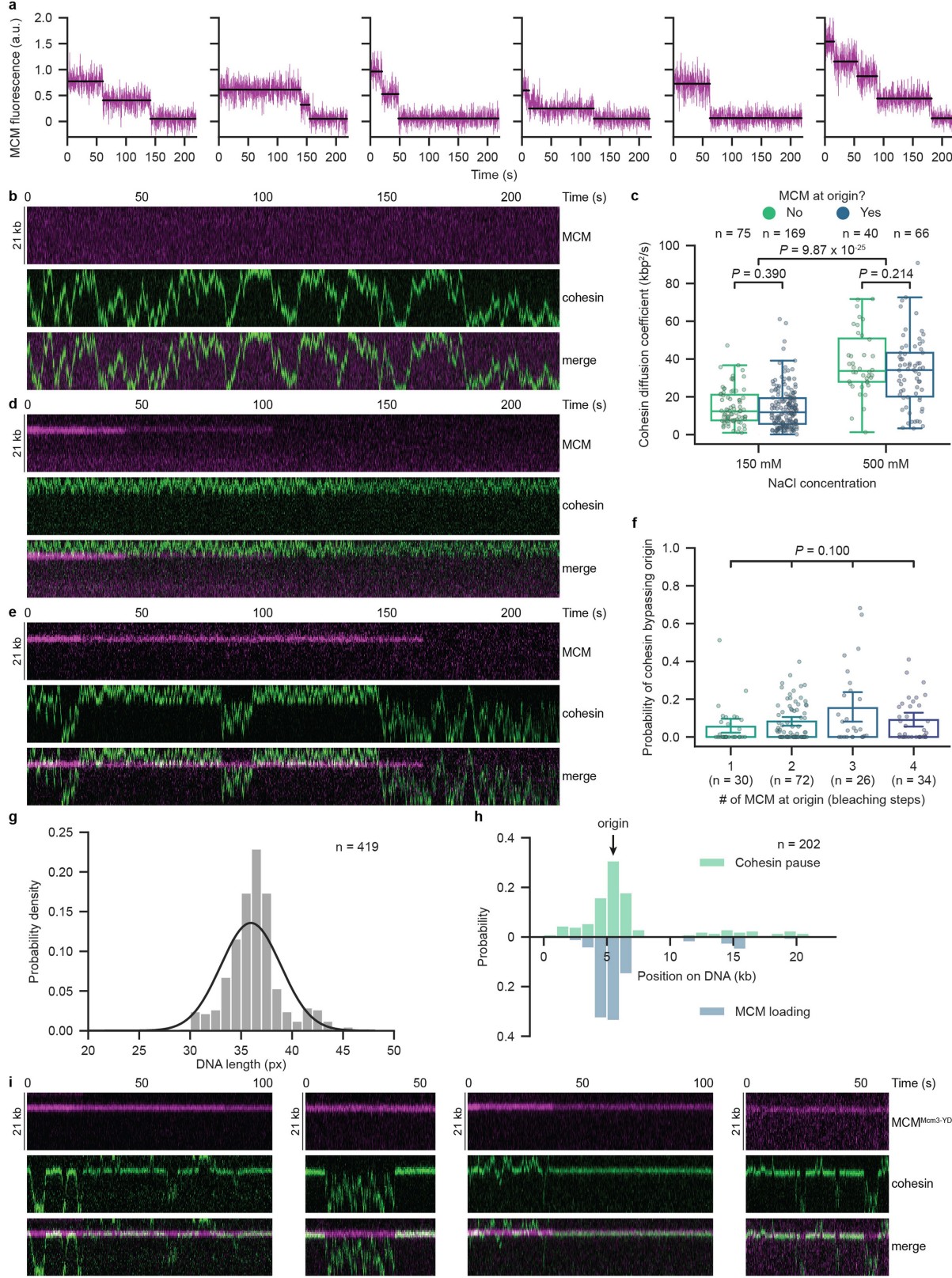

**Extended Data Fig. 9** | See next page for caption.

**Extended Data Fig. 9 | Translocating cohesin can bypass MCM with reduced efficiency. a**, MCM loads as double-hexamers at origins. Time traces of origin-bound MCM fluorescence intensity (purple) show two-step bleaching but less frequent also one and multi-step (black, fits by kinetic change point analysis) **b**, Representative kymograph of translocating cohesin in the absence of MCM at the origin. **c**, MCM does not alter observed cohesin translocation velocity. Box plots of cohesin diffusion coefficients in the absence (green) or presence (blue) of MCM at 150 or 500 mM NaCl. The centre line displays median, box edges show quartiles 1–3, and whiskers span quartiles 1–3 ± 1.5 × interquartile range. *P* values were determined by two-sided Mann-Whitney *U* test. **d**, **e**, Representative kymographs of translocating cohesin in the presence of MCM at the origin. MCM is a strong barrier for cohesin translocation (**d**) with MCM passage observed infrequently (**e**) during a 220 s interval. **f**, Multiple loaded MCMs do not increase the barrier strength for cohesin translocation. Photobleaching analysis confirm loaded MCM double-hexamers as main species. Data are depicted as mean within a 95 % confidence interval (generated by bootstrapping). *P* values were determined by Kruskal-Wallis test. **g**, Length distribution of doubly tethered DNA in pixels (px). The line represents a Gaussian fit. **h**, Cohesin translocation pauses at origins bound by MCM. Distribution of cohesin pause (green) and corresponding MCM (blue) positions on DNA. **i**, Representative kymographs of translocating cohesin showing frequent pausing after encountering MCM$^{Mcm3-YDF}$ at the origin. All data displayed (except in **e** and where specified in **c**) were imaged under physiological salt conditions (0.15 M NaCl).

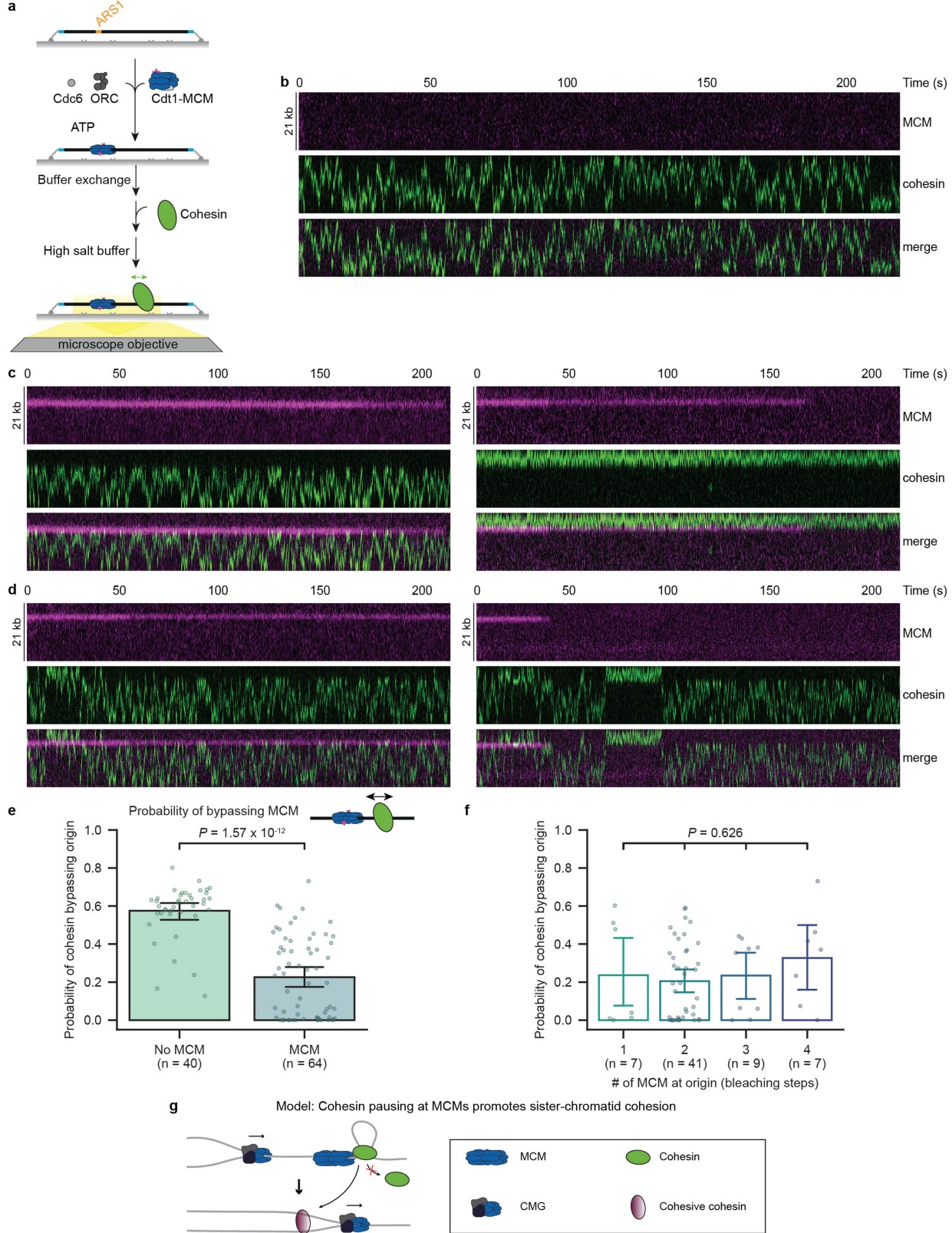

**Extended Data Fig. 10** | See next page for caption.

**Extended Data Fig. 10 | MCM is a barrier for cohesin translocation at a high salt concentration. a**, Schematic principle of a single-molecule cohesin translocation assay on licensed DNA. MCM is loaded onto DNA in the presence of the licensing factors ORC and Cdc6, followed by cohesin as described in Fig. 4a. Subsequently, cohesin translocation is visualized at high salt concentration (0.5 M NaCl) in the absence of free protein and buffer flow. **b–d**, Representative kymographs of translocating cohesin on DNA in the absence (**b**) or presence of MCM at the origin (**c**, **d**). Origin-bound MCM is a strong barrier to cohesin translocation (**c**) with passage events occurring infrequently (**d**) during a 220 s observation interval. **e**, MCM is a barrier for cohesin translocation at high salt concentration. Probability of translocating cohesin bypassing the origin in the absence or presence of MCM calculated from 40 or 64 molecules with 7802 or 9829 visualized encounters, respectively. **f**, Multiple loaded MCMs do not increase the barrier strength for cohesin translocation. Data in **e**, **f** are depicted as mean within a 95% confidence interval (generated by bootstrapping). *P values* were determined by Kruskal-Wallis followed by Dunn's post-hoc test. All data displayed were imaged in the presence of 0.5 M NaCl. **g**, Model showing that pausing of extruding cohesin at MCMs could promote sister-chromatid cohesion.

# Reporting Summary

Nature Research wishes to improve the reproducibility of the work that we publish. This form provides structure for consistency and transparency in reporting. For further information on Nature Research policies, see our Editorial Policies and the Editorial Policy Checklist.

## Statistics

For all statistical analyses, confirm that the following items are present in the figure legend, table legend, main text, or Methods section.

| n/a | Confirmed | |
|---|---|---|
| ☐ | ☒ | The exact sample size (*n*) for each experimental group/condition, given as a discrete number and unit of measurement |
| ☐ | ☒ | A statement on whether measurements were taken from distinct samples or whether the same sample was measured repeatedly |
| ☐ | ☒ | The statistical test(s) used AND whether they are one- or two-sided<br>*Only common tests should be described solely by name; describe more complex techniques in the Methods section.* |
| ☒ | ☐ | A description of all covariates tested |
| ☐ | ☒ | A description of any assumptions or corrections, such as tests of normality and adjustment for multiple comparisons |
| ☐ | ☒ | A full description of the statistical parameters including central tendency (e.g. means) or other basic estimates (e.g. regression coefficient) AND variation (e.g. standard deviation) or associated estimates of uncertainty (e.g. confidence intervals) |
| ☐ | ☒ | For null hypothesis testing, the test statistic (e.g. $F$, $t$, $r$) with confidence intervals, effect sizes, degrees of freedom and $P$ value noted<br>*Give P values as exact values whenever suitable.* |
| ☒ | ☐ | For Bayesian analysis, information on the choice of priors and Markov chain Monte Carlo settings |
| ☒ | ☐ | For hierarchical and complex designs, identification of the appropriate level for tests and full reporting of outcomes |
| ☒ | ☐ | Estimates of effect sizes (e.g. Cohen's *d*, Pearson's *r*), indicating how they were calculated |

*Our web collection on statistics for biologists contains articles on many of the points above.*

## Software and code

Policy information about availability of computer code

| Data collection | - Images: LSM780 and LSM880 microscope using a plan-apochromat 63x/1.4 oil immersion objective operated by ZEN acquisition software ZEN Black (2.8) and ZEN Blue (3.0-3.3) (ZEISS)<br>- Sequencing data: HiSeq 2500 v4 system in PE125 mode, NextSeq system using high-output lane in PE75 mode, NovaSeq 6000 system in PE100 mode<br>- Western blotting: ChemiDoc imaging system operated by Image Lab software (Bio-Rad)<br>- Single-molecule assay: RM21 micromirror TIRF microscope (Mad City Labs) with an Apo N TIRF 60 x oil-immersion TIRF objective (NA 1.49, Olympus) |
|---|---|
| Data analysis | numpy (Version 1.19.5)<br>scipy (Version 1.4.1)<br>pandas (Version 1.3.4)<br>matplotlib (Version 3.1.3)<br>seaborn (Version 0.11.2)<br>pingouin (Version 0.3.12)<br>scikit_posthocs (Version 0.6.7)<br>pairtools (https://pairtools.readthedocs.io/en/latest/)<br>distiller (https://github.com/open2c/distiller-nf)<br>cooltools (https://github.com/open2c/cooltools)<br>chromosight (https://github.com/koszullab/chromosight)<br>coolpup.py (https://github.com/open2c/coolpuppy)<br>Mustache (https://github.com/ay-lab/mustache)<br>HICcompare (https://github.com/dozmorovlab/HiCcompare)<br>ImageJ/FIJI (Version 1.53c)<br>SVI Huygens Professional (Version 20.04) |

R (Version 3.6.3)
RStudio (Version 1.3.959)
ggplot2 (Version 3.3.3)
Kallisto (Version 0.46.1)
ShinyGO (http://bioinformatics.sdstate.edu/go/)
Molecule Archive Suite (Mars) plugin (https://github.com/duderstadt-lab/)
ImageLab (Version 5.2.1)
Prism (Version 8.3.1)
Microsoft Excel for Mac (Version 16.26)

For manuscripts utilizing custom algorithms or software that are central to the research but not yet described in published literature, software must be made available to editors and reviewers. We strongly encourage code deposition in a community repository (e.g. GitHub). See the Nature Research guidelines for submitting code & software for further information.

## Data

Policy information about availability of data

All manuscripts must include a data availability statement. This statement should provide the following information, where applicable:
- Accession codes, unique identifiers, or web links for publicly available datasets
- A list of figures that have associated raw data
- A description of any restrictions on data availability

All sequencing data in support of the findings of this study have been deposited in the Gene Expression Omnibus (GEO) under the series accession numbers GSE196497 (snHi-C and RNA-Seq) and GSE155971 (Hi-C and Micro-C). The single-molecule video datasets supporting the findings in this study are deposited at Zenodo with following DOI´s: 10.5281/zenodo.5911107 (high salt experiments), 10.5281/zenodo.5911211 (physiological salt experiments) and 10.5281/zenodo.5911285 (ydf experiments). All data are also available from the authors upon request.

# Field-specific reporting

Please select the one below that is the best fit for your research. If you are not sure, read the appropriate sections before making your selection.

☒ Life sciences        ☐ Behavioural & social sciences        ☐ Ecological, evolutionary & environmental sciences

For a reference copy of the document with all sections, see nature.com/documents/nr-reporting-summary-flat.pdf

# Life sciences study design

All studies must disclose on these points even when the disclosure is negative.

| | |
|---|---|
| Sample size | No statistical methods were used to estimate sample size. Sample size was determined based on previous studies in the field to generate reproducible results. |
| Data exclusions | Exclusion criteria for snHi-C data:<br>- Unfertilized cells, polyspermic cells and zygotes in wrong cell cycle phase were excluded from the snHi-C procedure.<br>- snHi-C samples with < 100k total contacts were excluded from all analyses. |
| Replication | snHi-C: Number of mice and pronuclei is indicated in the figure legends. In general, 3-4 independent experiments were performed using 3-6 females for each genotype.<br>RNA-Seq: RNA-Seq for G1 and G2 zygotes was performed in duplicate and RNA-Seq for tissue culture cells was performed in triplicate.<br>Hi-C/micro-C: Hi-C/micro-C experiments with the MCM2-mAID cell line were performed using 1 Hi-C replicate and 2 biological independent micro-C replicates for the condition DMSO vs AUX and 1 micro-C replicate for the condition DMSO/trp vs AUX/trp.<br>Single-molecule studies: Number of observations analyzed is indicated in the figure or figure legends. All single-molecule observations were derived from at least three independent experiments.<br><br>All attempts for replication were successful. |
| Randomization | Randomization was not relevant to this study. Samples were organized into identifiable groups based on the experimental conditions (e.g. wild type vs mutant mouse strain,  +/- gemininL26A expression in G1 zygotes, auxin vs dmso treatment in HCT116 MCM2-mAID cell line, cohesin pausing at MCM or MCM-Mcm3-YDF). Samples within control or experimental groups were randomly assigned. |
| Blinding | Blinding was not relevant due to the design of this work. The researchers need to verify the experimental conditions (e.g. wild type vs mutant mouse strain, experimental time points, auxin vs dmso treatment in HCT116 MCM2-mAID cell line). Observer bias was prevented because the samples were measured using DNA sequencing or automated microscopy and the analysis was performed using automated python and R scripts. |

# Reporting for specific materials, systems and methods

We require information from authors about some types of materials, experimental systems and methods used in many studies. Here, indicate whether each material, system or method listed is relevant to your study. If you are not sure if a list item applies to your research, read the appropriate section before selecting a response.

## Materials & experimental systems

| n/a | Involved in the study |
|-----|------------------------|
| ☐ | ☒ Antibodies |
| ☐ | ☒ Eukaryotic cell lines |
| ☒ | ☐ Palaeontology and archaeology |
| ☐ | ☒ Animals and other organisms |
| ☒ | ☐ Human research participants |
| ☒ | ☐ Clinical data |
| ☒ | ☐ Dual use research of concern |

## Methods

| n/a | Involved in the study |
|-----|------------------------|
| ☒ | ☐ ChIP-seq |
| ☐ | ☒ Flow cytometry |
| ☒ | ☐ MRI-based neuroimaging |

# Antibodies

| | |
|---|---|
| Antibodies used | For immunofluorescence in oocytes and zygotes:<br>Primary antibodies<br>- anti-MCM2 (1:500, BD transduction Laboratories, #610701)<br>- anti-CTCF (1:250, Peters laboratory A992)<br>- anti-MYC (1:500, Millipore, #05-724)<br>Secondary antibodies<br>- Goat anti-mouse Alexa Fluor 488 (1:500, Invitrogen, #A11029)<br>- Donkey anti-rabbit Alexa Fluor 568 (1:500, Invitrogen, #A10042)<br>- Goat anti-mouse Alexa Fluor 647 (1:500, Invitrogen, #A-21235)<br><br>For Western blotting:<br>Primary antibodies<br>- anti-MCM2 (1:5000, BD transduction Laboratories, #610701)<br>- anti-MCM4 (1:5000, Abcam, #ab4459)<br>- anti-H3 (1:2000, Cell Signaling, #9715S)<br>- anti-GAPDH (1:2500, Millipore, #MAB374)<br>- anti-CTCF (1:1000, Peters laboratory A992)<br>- anti-PCNA (1:500, Santa Cruz, #PC10)<br>- anti-SCC1 (1:1000, Millipore, #05-908)<br>- anti-Pol II 8WG16 (1:500, Santa Cruz, #sc-56767)<br>Secondary antibodies:<br>- Goat Anti-Mouse Immunoglobulins/HRP (1:500, Dako, #P0447)<br>- Goat Anti-Rabbit Immunoglobulins/HRP (1:500, Dako, #P0448) |
| Validation | - anti-MCM2: https://www.bdbiosciences.com/us/reagents/research/antibodies-buffers/cell-biology-reagents/cell-biology-antibodies/purified-mouse-anti-bm28-46bm28/p/610701<br>- anti-CTCF: Wutz, G. et al. Topologically associating domains and chromatin loops depend on cohesin and are regulated by CTCF, WAPL, and PDS5 proteins. The EMBO journal e201798004 (2017)<br>- anti-MCM4: https://www.abcam.com/mcm4-antibody-ab4459.html<br>- anti-MYC: https://www.merckmillipore.com/BE/fr/product/Anti-Myc-Tag-Antibody-clone-4A6,MM_NF-05-724?ReferrerURL=https%3A%2F%2Fwww.google.com%2F<br>- anti-H3: https://www.cellsignal.com/products/primary-antibodies/histone-h3-antibody/9715?Ntk=Products&Ntt=9715<br>- anti-GAPDH: https://www.merckmillipore.com/AT/de/product/Anti-Glyceraldehyde-3-Phosphate-Dehydrogenase-Antibody-clone-6C5,MM_NF-MAB374?ReferrerURL=https%3A%2F%2Fwww.google.com%2F<br>- anti-SCC1: https://www.merckmillipore.com/BE/fr/product/Anti-RAD21-Antibody,MM_NF-05-908?ReferrerURL=https%3A%2F%2Fwww.google.com%2F&bd=1<br>- anti PCNA: https://www.scbt.com/p/pcna-antibody-pc10<br>- anti-Pol II 8WG16: https://www.scbt.com/p/pol-ii-antibody-f-12 gclid=EAIaIQobChMIw-3uuK6g9AIVjLh3Ch0lKg_6EAAYASAAEgKMiPD_BwE<br>- Alexa Goat anti-mouse 488: https://www.thermofisher.com/antibody/product/Goat-anti-Mouse-IgG-H-L-Highly-Cross-Adsorbed-Secondary-Antibody-Polyclonal/A-11029<br>- Alexa Donkey anti-rabbit 568: https://www.thermofisher.com/antibody/product/Donkey-anti-Rabbit-IgG-H-L-Highly-Cross-Adsorbed-Secondary-Antibody-Polyclonal/A10042<br>- Alexa Goat anti-mouse 647: https://www.thermofisher.com/antibody/product/Goat-anti-Mouse-IgG-H-L-Cross-Adsorbed-Secondary-Antibody-Polyclonal/A-21235<br>- Goat Anti-Mouse Immunoglobulins/HRP: https://www.agilent.com/en/product/immunohistochemistry/antibodies-controls/secondary-antibodies/goat-anti-mouse-immunoglobulins-hrp-(affinity-isolated)-153239<br>- Goat Anti-Rabbit Immunoglobulins/HRP: https://www.agilent.com/en/product/immunohistochemistry/antibodies-controls/secondary-antibodies/goat-anti-rabbit-immunoglobulins-hrp-(affinity-isolated)-153244 |

# Eukaryotic cell lines

Policy information about cell lines

| | |
|---|---|
| Cell line source(s) | HCT116 MCM2-mAID-NeoR (biallelic), CMV-AtAFB2<br>Clone 16 |

| Authentication | Supplied by Toyoaki Natsume
Described in: Natsume, T. et al. Acute inactivation of the replicative helicase in human cells triggers MCM8-9-dependent DNA synthesis. Genes & development 31, 816–829 (2017).

HR-mediated C-terminal tagging of MCM2 in the HCT116 MCM2-mAID-NeoR cell line was examined by western blotting. No growth defects were observed in this clone. |
| --- | --- |
| Mycoplasma contamination | Cell line tested negative for mycoplasma test. |
| Commonly misidentified lines
(See ICLAC register) | No commonly misidentified cell line was used. |

# Animals and other organisms

Policy information about studies involving animals; ARRIVE guidelines recommended for reporting animal research

| Laboratory animals | Mice were housed in individually ventilated cages under a 14h light/10h dark cycle at ambient temperature of 22 °C ± 1 °C and humidity of 55% ± 5% with continuous access to food and water. Animals were housed grouped (maximum 4 males per cage and maximum 5 females per cage). All mice were bred in the IMBA animal facility. Wildtype, Scc1fl/fl and Scc1Myc/+ mice were bred on a mixed background (B6, 129, Sv). Waplfl/fl and Zp3-dsCTCF mice were bred on a primarily C57BL/6J background. Zp3-dsCTCF mice were maintained by breeding Zp3-dsCTCF males to C57BL/6J females. Experimental Scc1fl/fl and Waplfl/fl mice were obtained by mating of homozygous floxed females to homozygous floxed males carrying Tg(Zp3Cre). Experimental Scc1Myc/+ mice were obtained by intercrossing heterozygous Scc1Myc/+ mice. Experimental Zp3-dsCTCF mice were maintained by breeding Zp3-dsCTCF males to C57BL/6J females. Experimental females/males were between 2 to 5 months old. |
| --- | --- |
| Wild animals | This study did not involve wild animals. |
| Field-collected samples | This study did not involve samples collected from the field. |
| Ethics oversight | The mice used in this work were bred and maintained in agreement with the authorizing committee according to the Austrian Animal Welfare law and the guidelines of the International Guiding Principles for Biomedical Research Involving Animals (CIOMS, the Council for International Organizations of Medical Sciences). |

Note that full information on the approval of the study protocol must also be provided in the manuscript.

# Flow Cytometry

## Plots

Confirm that:

☒ The axis labels state the marker and fluorochrome used (e.g. CD4-FITC).

☒ The axis scales are clearly visible. Include numbers along axes only for bottom left plot of group (a 'group' is an analysis of identical markers).

☒ All plots are contour plots with outliers or pseudocolor plots.

☒ A numerical value for number of cells or percentage (with statistics) is provided.

## Methodology

| Sample preparation | For G1 FACS-sorting, cells were synchronized with a double-thymidine arrest–release followed by release into fresh medium for 12 h. 4 h before sorting Hoechst 33342 (Sigma) was added to the medium at a concentration of 0.2 µg/ml. After harvesting, the cells were resuspended in FACS-buffer (PBS + 2% FCS) and immediately sorted. |
| --- | --- |
| Instrument | BD FACSAria™ III Cell Sorter |
| Software | The BD FACSAria™ III Cell Sorter controlled by FACSDiva software. Flow cytometry data was analyzed using FlowJo V10. |
| Cell population abundance | 1 million HCT116 MCM2-mAID cells were sorted for each sample. |
| Gating strategy | SSC-A vs FSC-A gating was used to filter out debris followed by FSC-H vs FSC-A gating to exclude doublets. The cells stained with Hoechst 33342 were selected using HOECHST-W vs HOECHST-A gating and the G1 population could be selected with Count vs HOECHST-A gating. To avoid S-phase cell contamination, only cells in the left part of the G1 peak were collected (red dashed box in Extended Data Fig. 8b). |

☒ Tick this box to confirm that a figure exemplifying the gating strategy is provided in the Supplementary Information.

