## [Peer Review File · Nature]

Manuscript Title: MCM complexes are barriers that restrict cohesin-mediated loop extrusion

Editorial Notes:

Redactions – unpublished data

Reviewer Comments & Author Rebuttals

Reviewer Reports on the Initial Version:

Referees' comments:

Referee #1 (Remarks to the Author):

The mammalian genome is organized into loops and TADs which are mainly facilitated by cohesin and extrusion blocking factors, in particular, CTCF. However, whether other factors can function as similar barriers for cohesin is unclear. In this manuscript, Dequeker et al. found that the MCM complexes, the key factors that mediate DNA replication, can also function as cohesin barriers. By elegant designs, the authors showed that blocking the loading of MCM complex to chromatin in mouse zygotes leads to increased loop, TAD strengths and TAD insulation. Such increase does not affect the process of loop extrusion, but depends on the presence of CTCF and is further enhanced by the deficiency of Wapl. In somatic cells, MCM loss shows similar but mild effect. Finally, single molecule analyses and loop extrusion simulation further validated the MCM barrier for cohesin.

Generally speaking, this is a well-designed project and the findings greatly shed light on the nature of cohesin translocation on chromatin. The genetic experiments and single-molecule experiments are impressive. The polymer simulation is a plus, although it is perhaps beyond my expertise to evaluate the results. At this moment, I do have some questions regarding the data quality and data interpretation (see below). Whether this interesting finding is also true elsewhere seems unclear to me, given the relatively weak changes upon the loss of MCM in somatic cells. Finally, whether the MCM-mediated cohesin barriers are functional or are simply a passive process deserves further (perhaps future) studies. Please see below for my specific comments:

1. I cannot seem to find the replicate information in the paper. How reproducible the results are, considering the variations in mouse embryos and single cell HiC? How many cells were investigated?
2. Again, due to the lack of replicates, it is difficult to assess some of the authors' conclusions. For example, the authors concluded that CTCF loss shows no gross changes in $p(s)$ curves (Fig. S5). Yet, to me, the two curves show apparent differences.
3. Despite the detailed explanation of the authors, it is still puzzling that MCM mutants do not affect loop sizes. This is a bit counter-intuitive as one would imagine that the additional stops of cohesin by

MCM would decrease loop sizes, as clearly illustrated in Fig. 4E. The authors concluded that if the separation between random barriers is large (to the size of existing loops) or the barriers are sufficiently permeable, then the loop sizes won't be affected. At least in the latter case, presumably cohesin won't have a problem reaching CTCF sites, which means the CTCF boundary should not be much affected?

4. A related question is that CTCF has recently been shown to be required for the formation of long chromatin loops by protecting cohesin-Stag1 from Wapl (Wutz et al., *Elife*, 2020). The data seem quite convincing there. Yet, the authors believed that there are no apparent changes in the $p(s)$ curve upon CTCF depletion (Fig. S5). If this is true, it remains unclear if such analysis can truly prove that the loop size is undisturbed. The same question would also apply to the MCM loss.
5. Fig. 1C, as MCM loss alone leads to stronger TADs and TAD boundary, it is surprising that the loss of MCM leads to more boundary insulation in the absence of CTCF. The difference seems a bit weak. Again, are these results reproducible? If so, how to explain the discrepancy? The authors attributed the results to the possible boundary function of MCM, but it is not quite convincing unless MCM can also be localized to CTCF sites.
6. Mouse 1-cell embryos in many ways are quite unique (highly permissive chromatin, relaxed chromatin architecture, etc.). It would make the finding more significant if MCM can also function as cohesin barriers in other cell types. At this moment, the effects of MCM loss for TAD and TAD boundary insulation in somatic cells seem too weak to be conclusive. Firstly, can the authors provide a replicate to confirm that such changes can be consistently observed?
7. One interesting question is that if the barrier function of MCM is simply due to its physical properties such as the bulkiness (so that other similar macromolecules may also have similar functions) or via specific protein-protein interactions. For example, does MCM interact with cohesin *in vitro*?
8. The single molecular experiment results are impressive. However, the components of the MCM complex are from *Saccharomyces cerevisiae*. Can the authors discuss the possible differences between yeast and mammalian MCM complex?
9. Hongtao Yu's group has shown that MCM is required for cohesin loading at the S phase (Zheng et al., *Elife*, 2018). Does this mean that MCM can both facilitate cohesin loading while also functions as cohesin barriers?

Minor comments:

10. Can the authors further exclude the possibility that geminin mutant-mediated MCM loss does not affect the protein levels of NIPBL, WAPL, and CTCF, etc.?
11. Fig. 2E-F. I may have missed but it seems the maternal result is missing?
12. In Fig. S7C, it would be helpful to also check the expression of cohesin/ctcf/wapl etc.

Referee #2 (Remarks to the Author):

In this very interesting paper, the authors use a combination of *in vivo* Hi-C approaches and *in vitro* single molecule experiments to support the hypothesis that MCM complexes loaded on chromatin block loop extrusion by cohesin. Using injection of RNA encoding a non-degradable Geminin protein to block MCM loading during the oocyte to zygote transition, they found that G1 cells with MCM depletion show an increase in peaks and TADs relative to the control. They use some elegant genetics to show that the effect is not mediated through Wapl and they show similar effects are seen in G1 HCT116 cells after MCM2 degradation using an auxin degron. They use a previously described single molecule loop extrusion assay to show that MCM loaded by ORC, Cdc6 and Cdt1 *in vitro* acts as a boundary for loop extrusion. Finally, via computer simulation, they estimate that loop

extrusion is blocked at 10-40% of loaded MCM. The work described here represents an extraordinary technical achievement. Nonetheless, I have some concerns that I feel need to be addressed.

1. The effects are quite subtle. I assume they are statistically robust (though I saw no statement to this effect); however, the real question is “Is this relatively subtle effect functionally meaningful?” While the effect on TADs will be interesting to workers in the field, it would be of more general interest if the alterations had some demonstrable physiological effect. The most likely such effect would be on G1 transcription. Since the zygotes are not transcriptionally active, they aren’t a great model for this, but the HCT116 system is transcriptionally active. Does MCM depletion affect the transcriptome in this situation in a cohesin-dependent way?

2. The interpretation of the results is absolutely dependent upon the quality of the G1 arrest: If even a few cells enter S phase, this will affect the results +/- MCM loading since there will be replication in one case but not the other, and replication might be expected to affect chromosome structure. I could find no assurance of the quality of synchrony for the zygotes. The very subtle differences in the HCT116 cells could be due to the ~15% non-G1 cells.

3. Can the authors rule out the possibility that more CTCF binds chromatin in the absence of MCM? This could explain at least some of their results.

4. The in vitro experiments use a mixed system: yeast MCM and human cohesin. This is far from ideal. The inability of cohesin to pass MCM could be due to non-conserved protein interactions, or to differences in the stability of the loaded MCM complex between yeast and humans. It would be far better if this experiment could be repeated with either yeast cohesin or (preferably) human MCM. The Bleichert lab have recently reconstituted MCM loading with human proteins in vitro (Schmidt and Bleichert, Nature Comms, 2020).

5. As I understand it, the conclusion that MCM single hexamers can block loop extrusion is based on a very small number (7) molecules. Is this robust? Can you exclude the possibility that these represent double hexamers in which one fluorophore bleached? If not, this discussion should be removed or the experiment repeated with bigger numbers and better controls for bleaching.

6. You begin by citing the diameter of MCM and FtsK both being 12-13nm. The diameter of a nucleosome core particle is ~11nm which begs the question, “Can a nucleosome block loop extrusion?” As I understand it, cohesin can compact nucleosomal DNA but can it pass a nucleosome? This seems a rather important control.

7. Related to this, how special is the ability of MCM to block cohesin? Is this likely to be a general property of DNA binding proteins or is there something special about MCM. If so, what is special?

Referee #3 (Remarks to the Author):

In this paper, Dequeker et al. investigated the possible scenario that DNA-bound proteins serve as barriers to cohesin-mediated loop extrusion. Using single-molecule, Hi-C, and simulation methods, the authors provided evidence that the replicative helicase MCM is a barrier that restricts cohesin translocation and loop extrusion in G1 phase. This is certainly an important and hotly pursued topic. The combination of different techniques is a strength of the paper. The implication that dormant MCM complexes could play a role in genome topology is also intriguing. The manuscript is in general well written. However, I have concerns with the relevance of the single-molecule results, on which my critiques are focused.

The major concern I have, which was also noted by the authors, is that in all of the single-molecule experiments, cohesin was in a passive translocation mode, not in an active loop-extruding mode. Therefore, whether MCM can restrict active loop extrusion remains to be established in vitro. The authors briefly mentioned that MCM loading and loop extrusion require mutually incompatible conditions, but did not provide a detailed explanation.

Specific comments on single-molecule results:

- (1) 500 mM NaCl was used to promote cohesin sliding. It would be more convincing to visualize cohesin behavior under physiological salt.
- (2) How did the authors conclude that only MCM double-hexamers, but not ORC/Cdc6/Cdt1-containing intermediates, were formed on DNA? In Fig 3D, the origin bypassing probability without MCM for the majority of cohesins is lower than 0.6. Is it because ORC remained bound at the origin, or is it a DNA sequence effect?
- (3) MCM double-hexamers have been shown to be pushed by DNA translocases and slide on DNA (e.g. PMC4680849). Was MCM always stationary in the current assay?
- (4) Depending on the distance between the two anchored DNA ends, the MCM and cohesin localization precision is expected to differ between molecules that were tight versus slack. However, the same thresholds were used to classify encounter and passing for all molecules. What is the distribution of observed DNA lengths? How was the 1.5-kb threshold chosen?

Comments on Hi-C and simulation results (I should admit that I am not an expert on either of these areas):

- (1) Most of the Hi-C results are presented in a descriptive manner and lack quantification (error bars, statistical tests, etc.). The effect of MCM loss in somatic cells is particularly difficult to judge by eye (Fig S7).
- (2) The simulated and experimental data were claimed to be well matched (e.g. Fig 4D, Fig S14B-D). What are the criteria? They certainly do not perfectly overlap.

Reviewer #4 (Remarks to the Author):

In this study, Dequeker et al. use a combination of mouse genetics, Hi-C, in vitro experiments and modeling to study the interplay between minichromosome maintenance (MCM) complexes and loop extrusion by cohesin in mammalian genomes. The experimental and computational results in the manuscript suggest, but do not fully support, that MCM complexes form semi-permeable barriers to loop extrusion and thus modulate contact probabilities between CTCF sites, at least in the fertilized oocyte genome.

Provided some major concerns are addressed (see below), the study will be of interest for the nuclear structure field and in particular for the community of researchers interested in modeling loop extrusion. However, the relevance of the finding that replication origins might obstruct loop extrusion through MCM binding (which is by the way not formally proven in the manuscript) seems rather limited, notably by the fact that such effect does not seem to be observable in differentiated cell types.

Main points:

1. The central result in the paper is the increase in Hi-C enrichment at CTCF loops following inhibition of MCM loading. This effect is however rather small (~20%) and detected after averaging over very small numbers of pronuclei (with WT and mutant experiments involving different numbers of pronuclei). This raises the question whether the effect could be dominated by some outlier single-cell Hi-C preparations within either the wild-type or the mutant samples. To exclude this possibility, it would be necessary to verify that the results hold true when different (smaller) numbers and combinations of single-cell Hi-C samples are retained in the analysis (e.g. 6, 8, 10).

2. Based on single-cell RNA-seq performed in two WT and two 'MCM-loss' 10-cell samples, the authors conclude that there are no major differences in gene expression following loss of MCM. However, WT replicates do not cluster together in PCA (Figure S3B) and their reciprocal distance in the two first PCA components is of the same order of their distances to MCM-loss samples. It is thus unclear what can be concluded from these experiments.

3. At page 4, the authors claim that MCM depletion further decreases insulation scores compared to depleting CTCF alone. However it is impossible to assess whether the minimal differences shown in Figure 3C are to any extent significant. At the very least, insulation analysis should be performed upon averaging over different numbers and combinations of single pronuclei, as well as using different genomic windowing.

4. Why are the data for maternal and paternal pronuclei merged in Figure 2B (CTCF-MCM depletion) but not in Figure 2C (WAPL-MCM depletion)? Does it relate to the fact that upon WAPL depletion, only the scaling of contact probabilities in the maternal genome changes (Fig. S6D)? And why is it so?

5. The claim that in HCT116 cells the effects of MCM depletion are consistent with those observed in the zygotes but less pronounced is an overstatement and should be toned down. There seems to be absolutely no effect in this cell line judging from Fig. S7.

6. As also acknowledged by the authors, there is a clear disconnect between the three parts of the manuscript (in vivo, in vitro, simulations). The in vitro experiments show that MCM can impair cohesin translocation, but do not necessarily relate to the in vivo part because they provide no evidence that MCM complexes can also interfere with loop extrusion. The simulations instead can reproduce the in vivo data, but they only do so if MCM permeability for loop extruding cohesin is much larger than measured in vitro for passive translocation (80% vs. 20%). Taken together, the three parts suggest that MCM-mediated interference of loop extrusion *could* be the cause of the observed phenotype but constitute no formal proof. In vitro experiments with loop extruding cohesin would provide at least part of the missing link and strengthen the causality claims.

Minor point:

It is unclear what the numbers inside meta-loop and meta-TAD boundaries refer to. I presume it is some sort of mean enrichment in a region of the plot, but the Methods section is not clear in this regard.

Author Rebuttals to Initial Comments:

We would like to thank all reviewers for the positive comments and helpful suggestions, which were incorporated to generate a stronger manuscript. To briefly summarize the main improvements:

- 1) To address the concern that our roadblock assay uses yeast MCM and human cohesin, we developed a DNA loading assay for a chimaeric yeast MCM complex that contains an engineered cohesin-interacting YDF motif from human MCM3 (please note that we could not simply use human MCM because a specific DNA loading assay has not been developed yet for human MCM). These experiments revealed that “humanized” MCMs are not only barriers but also induce cohesin pausing. This reflects a molecular docking event, which indicates that human MCM is not simply a passive physical “roadblock” for cohesin but an active barrier (new Fig. 4).
- 2) To address the function of MCMs in altering chromatin organization, we tested whether acute MCM degradation in G1 phase HCT116 cells affects gene expression. We found that at least 229 genes are differentially regulated, which is similar in magnitude to the effect seen after acute CTCF loss (Nora et al., 2017). Therefore, MCMs affect gene expression to some extent, and this might be relevant for human pathologies in the MCM loading pathway.
- 3) To address the concern that the MCM barrier effect is specific to early embryos, we used a second method to determine whether MCMs reduce CTCF-anchored loops in HCT116 cells. Micro-C experiments confirmed that the effects on anchored loops and TADs are reproducible and robust (new Extended Data Fig. 14). We conclude that MCMs also impede loop extrusion in somatic cells.
- 4) To address the concern that the roadblock assay was performed under high salt conditions, we performed extensive optimizations to find conditions under which MCM loading and cohesin translocation can both occur. This enabled us to repeat the experiments under physiological salt conditions (new Fig 4) and found that the barrier effect was even strengthened under these conditions, further supporting our conclusions.

Please find our responses to individual comments below.

Referee #1 (Remarks to the Author):

The mammalian genome is organized into loops and TADs which are mainly facilitated by cohesin and extrusion blocking factors, in particular, CTCF. However, whether other factors can function as similar barriers for cohesin is unclear. In this manuscript, Dequeker et al. found that the MCM complexes, the key factors that mediate DNA replication, can also function as cohesin barriers. By elegant designs, the authors showed that blocking the loading of MCM complex to chromatin in mouse zygotes leads to increased loop, TAD strengths and TAD insulation. Such increase does not affect the process of loop extrusion, but depends on the presence of CTCF and is further enhanced by the deficiency of Wapl. In somatic cells, MCM loss shows similar but mild effect. Finally, single molecule analyses and loop extrusion simulation further validated the MCM barrier for cohesin.

Generally speaking, this is a well-designed project and the findings greatly shed light on the nature of cohesin translocation on chromatin. The genetic experiments and single-molecule experiments are impressive. The polymer simulation is a plus, although it is perhaps beyond my expertise to evaluate the results. At this moment, I do have some questions regarding the data quality and data interpretation (see below). Whether this interesting finding is also true elsewhere seems unclear to me, given the relatively weak changes upon the loss of MCM in somatic cells. Finally, whether the MCM-mediated cohesin barriers are functional or are simply a passive process deserves further (perhaps future) studies. Please see below for my specific comments:

1. I cannot seem to find the replicate information in the paper. How reproducible the results are, considering the variations in mouse embryos and single cell HiC? How many cells were investigated?

We thank the reviewer for the positive comments. Due to the single-nucleus nature of the experiments, each nucleus represents an independent replicate. We have now provided information about the number of nuclei analyzed more explicitly in the figure legends, indicating how many nuclei were included for which analysis. In general, each aggregate analysis is based on at least 3 million contacts from at least 8 nuclei, which is comparable to previous work (Flyamer et al., Nature, 2017; Gassler et al., EMBO J, 2017). Specifically, the following number of nuclei were analyzed:

- 13 maternal and 16 paternal WT control; 16 maternal and 15 paternal MCM loss
- 17 maternal and 20 paternal control; 11 maternal and 11 paternal Wapl delta; 10 maternal and 9 paternal Wapl delta/MCM loss
- 13 maternal and 11 paternal control; 12 maternal and 8 paternal CTCF knockdown; 10 maternal and 8 paternal CTCF knockdown/MCM loss

2. Again, due to the lack of replicates, it is difficult to assess some of the authors' conclusions. For example, the authors concluded that CTCF loss shows no gross changes in $p(s)$ curves (Fig. S5). Yet, to me, the two curves show apparent differences.

We thank the reviewer for pointing this out. We have now amended this statement to reflect that there are no gross changes in $P_c(s)$ curves up to 1 Mb, which is the distance range of dynamically extruded loops (Schwarzer et al., Nature, 2017; Haarhuis et al., Cell, 2017; Gassler et al., EMBO J, 2017; Wutz et al., EMBO J, 2017), but that there is an increase in long-range chromatin contacts, which is more pronounced in paternal chromatin. The CTCF-independent increase in long-range interactions is consistent with previous findings in tissue culture cells (Wutz et al., eLife, 2020).

3. Despite the detailed explanation of the authors, it is still puzzling that MCM mutants do not affect loop sizes. This is a bit counter-intuitive as one would imagine that the additional stops of cohesin by MCM would decrease loop sizes, as clearly illustrated in Fig. 4E. The authors concluded that if the separation between random barriers is large (to the size of existing loops) or the barriers are sufficiently permeable, then the loop sizes won't be affected. At least in the latter case, presumably cohesin won't have a problem reaching CTCF sites, which means the CTCF boundary should not be much affected?

The reviewer is correct in that MCM depletion does affect loop sizes, exactly as one would expect, but not to the extent that this becomes obvious in $P_c(s)$ curves. The modest change in loop size (less than 10-15% change, new Extended Data Fig. 10, 11) is sufficiently small to prevent gross changes in the $P_c(s)$ curves but sufficiently large to affect the relative positioning of the cohesin loops such that the “corner peak” frequency is more visibly affected. We have modified the text accordingly and softened our language to say that the effect on loop sizes is “small”, e.g. “Together, these characteristic effects on chromatin organization are consistent with the notion that MCMs impede loop extrusion in a specific way, changing cohesin loop positioning, affecting largely CTCF-mediated peaks while having a small effect on overall sizes of extruded loops.”

We explain the intuition in the following way: the $P_c(s)$ curve is a genome-wide metric, which depends critically on the cohesin loop size; cohesin loop sizes are affected by CTCFs, MCM barriers and, importantly, other cohesins. In general, extrusion barriers like CTCFs and MCMs can have small effects on the average loop size, while having larger and visible effects on Hi-C features, e.g. CTCF depletion does not change the average loop size, but has a profound effect on “dots” and TADs. As we now explained in new Fig. Fig 3e and in the main text (see below), random barriers reduce the frequency of CTCF-CTCF bridging contacts by introducing gaps between CTCF-anchored loops: even a small gap would prevent a CTCF-CTCF contact while having little effect on the average loop length.

In our simulations, the effective density of CTCFs, cohesins and MCMs are at approximately 1/300 kb. Since other cohesins and CTCFs account for 2/3 of barriers to any cohesin's extrusion, the removal of MCMs might be expected to have at most a ~33% effect on the overall loop size. On the other hand, as far as “corner peaks”, only cohesins and MCMs are barriers to “corner peak” formation (since CTCFs are by construction what delimits the corner peak). Removal of the MCM barriers therefore has a greater effect on corner peak formation (i.e. since 50% of barriers to CTCFs disappear), making it more likely that a cohesin arrives at a CTCF.

To include more intuition, we have modified the results section: “These models provide a rationale for the seemingly contradictory ability of MCM to reduce the strength of CTCF-mediated features without strongly affecting sizes of extruded loops. CTCF-mediated peaks of contact probability emerge if one or more cohesins extrude loops in a region between neighbouring CTCF barriers and are able to bridge between these barriers, i.e. extrude all chromatin between CTCF, even transiently. Such peaks, however, become weaker or disappear if two CTCFs cannot be bridged when an unextruded gap is present between extruded loops. If one or more randomly located barriers are present between a pair of CTCFs, extrusion is likely to leave some unextruded gaps that result in weaker CTCF-CTCF peaks (Fig. 3e). For example, if separation between such random barriers is large (i.e. comparable to average TAD size of ~300 kb) or the barriers are sufficiently permeable to have an effective density of 1 per TAD, then the loop sizes change by <15%. This is a previously unrecognized and rather surprising effect of random barriers on different features of chromosome organization.”

4. A related question is that CTCF has recently been shown to be required for the formation of long chromatin loops by protecting cohesin-Stag1 from Wapl (Wutz et al., Elife, 2020). The data seem quite convincing there. Yet, the authors believed that there are no apparent changes in the $p(s)$ curve upon CTCF depletion (Fig. S5). If this is true, it remains unclear if such analysis can truly prove that the loop size is undisturbed. The same question would also apply to the MCM loss.

Please see our response to point 2 above. We have refined the statement to reflect that there are no gross changes in $P_c(s)$ curves up to 1 Mb but long-range interactions are increased upon CTCF loss. The similar findings in $P_c(s)$ curves between snHi-C and bulk Hi-C data (Wutz et al., eLife 2020) provide evidence that the analysis of loop sizes is appropriate.

The effect of CTCF on loop sizes can be two-fold: (i) as the reviewer has pointed out, CTCF-cohesin complex has an extended residence time on chromatin and thus lead to larger extruded loops; (ii) by stopping loop extrusion CTCF reduces the sizes of loops. As evident from several CTCF depletion studies, the collective effect of CTCF on the average loop size is minimal, possibly due to mutual compensation of one effect by the other.

5. Fig. 1C, as MCM loss alone leads to stronger TADs and TAD boundary, it is surprising that the loss of MCM leads to more boundary insulation in the absence of CTCF. The difference seems a bit weak. Again, are these results reproducible? If so, how to explain the discrepancy? The authors attributed the results to the possible boundary function of MCM, but it is not quite convincing unless MCM can also be localized to CTCF sites.

During the revision we have increased the number of nuclei in which CTCF and MCM are co-depleted from 8 to 18. The larger data set shows that in the absence of CTCF, there is little difference in TAD boundary insulation with or without MCMs. We therefore conclude that CTCF is instructive for boundary formation and that MCMs are additional, randomly positioned barriers.

We have preliminary data (see below) showing that MCMs are enriched at CTCF sites in tissue culture cells. It is therefore conceivable that MCMs reinforce boundaries at those positions. However, there is no evidence that MCMs localize to CTCF sites in the absence of CTCF binding.

***FIGURE REDACTED**

6. Mouse 1-cell embryos in many ways are quite unique (highly permissive chromatin, relaxed chromatin architecture, etc.). It would make the finding more significant if MCM can also function as cohesin barriers in other cell types. At this moment, the effects of MCM loss for TAD and TAD boundary insulation in somatic cells seem too weak to be conclusive. Firstly, can the authors provide a replicate to confirm that such changes can be consistently observed?

The reviewer is correct that the somatic cell data shown in the original version of our manuscript revealed only small effects of MCM depletion on genome architecture. This raises the question whether these results are robust. To address this, besides Hi-C, we used another method to test the effect of MCM depletion on corner peaks. Specifically, we performed duplicate micro-C experiments (new Extended Data Fig. 14) that confirm the earlier findings. CTCF corner peaks on contact maps are visibly enriched (new Extended Data Fig. 14d). and aggregate loops are stronger over a range of sizes in MCM-depleted cells (new Extended Data Fig. 14e, f). Moreover, the micro-C data revealed that consistently more loops are called *de novo* in MCM-depleted cells (new Extended Data Fig. 14g), providing further support that MCMs impede formation of CTCF-anchored loops in somatic cells. We conclude that MCMs function as cohesin barriers also in other cell types.

7. One interesting question is that if the barrier function of MCM is simply due to its physical properties such as the bulkiness (so that other similar macromolecules may also have similar functions) or via specific protein-protein interactions. For example, does MCM interact with cohesin in vitro?

This is a very interesting point that we investigated further during the revision. We think that bulkiness is not sufficient to explain MCM's barrier function. Consistent with this, a recent preprint from Cees Dekker's lab has shown that nucleosomes and large nanoparticles do not

effectively block loop extrusion *in vitro* (Pradhan et al., biorxiv preprint, 2021, <https://doi.org/10.1101/2021.07.15.452501>). Instead, we suspect that both the mode of DNA association and protein-protein interactions between MCM and cohesin are important for the barrier function of MCM.

We propose that the topological engagement of DNA by MCMs might promote its barrier function. This hypothesis is based on the observation that cohesive cohesin complexes, which are thought to entrap DNA inside their ring structure, restrict loop extrusion by extruding cohesin complexes in mouse oocytes, where these functions are carried out by distinct Rec8- and Scc1-cohesin complexes, respectively (Silva et al., J Cell Biol, 2020).

In addition, human MCM3 contains a disordered region harbouring a YDF motif and a peptide of this region can bind to cohesin *in vitro* (Li et al., Nature, 2020). However, yeast MCM3 does not contain this motif. Since there is no established human MCM loading assay, we modified our yeast assay by generating a chimaeric MCM complex in which the disordered region of human MCM3, containing the YDF motif, has been engineered into yeast MCM3. Remarkably, we found that human cohesin encountering this “humanized” version of yeast MCM is still blocked but now also pauses for extended periods of time. These results suggest that human MCM complexes will not only function as passive barriers for human cohesin but can actively bind cohesin. We suspect that this binding event promotes the barrier function of MCM and might even affect other functions of cohesin, for example could help to recruit cohesin.

8. The single molecular experiment results are impressive. However, the components of the MCM complex are from Saccharomyces cerevisiae. Can the authors discuss the possible differences between yeast and mammalian MCM complex?

The core architecture of the MCM complex is highly conserved from yeast to mammals, however, there is variability in the length and sequence of both the N- and C-terminal regions. MCM subunits from yeast to mammals all contain conserved AAA+ and N-terminal collar domains, which assemble into the ring that encircles and translocates along DNA. Additional phosphorylation sites and motifs contained in the disordered terminal regions allow for species specific adaptations. With this in mind, we examined the differences between the yeast and human proteins and noticed the presence of a YDF motif in mammalian but not in yeast MCM3. The same motif, found in the N-terminus of CTCF, was shown to bind SA2-SCC1 cohesin *in vitro* and to be required for the formation of CTCF-anchored loops (Li et al., Nature, 2020).

As explained above, we investigated whether the presence of this motif alters the outcome of cohesin-MCM collisions. For this purpose, we introduced a 19 amino acid region from human MCM3 containing the YDF motif into yeast MCM3 and performed additional cohesin translocation assays. We found that introduction of this disordered region altered the outcome of cohesin-MCM collisions *in vitro*. The presence of the motif lead to long pausing events during encounters between the MCMs and cohesin, suggesting a direct protein-protein interaction. These longer pauses were not observed for the yeast MCM lacking the motif. These new results as well as an accompanying discussion have been incorporated into the revised manuscript on pages 6-7 and in Figure 4 and Extended Data Figure 15.

9. Hongtao Yu's group has shown that MCM is required for cohesin loading at the S phase (Zheng et al., *Elife*, 2018). Does this mean that MCM can both facilitate cohesin loading while also functions as cohesin barriers?

Zheng et al., showed that MCM is required for cohesin loading during S phase but not in telophase/G1 phase (Zheng et al., *Elife*, 2018). In addition, MCM is not sufficient for cohesin loading and also requires CDC7. We therefore think that it is not mutually exclusive for MCM to function as extruding cohesin barriers in G1 phase, when CDC7-DDK is inactive, and to promote cohesin recruitment during S phase, when the kinase is active. Although speculative at this point, cohesin pausing at MCM-YDF complexes might in fact be part of cohesin recruitment or protection against release from chromatin in S phase.

Minor comments:

10. Can the authors further exclude the possibility that geminin mutant-mediated MCM loss does not affect the protein levels of NIPBL, WAPL, and CTCF, etc.?

We followed the reviewer's suggestion and examined the abundance of these proteins by immunofluorescent staining of zygotes (pre-extraction of soluble proteins) since Western blotting would require at least 100 zygotes and 8-10 female mice. We detected no gross changes in abundance for chromatin-associated SCC1 and CTCF (new Extended Data Fig. 2e, f). Unfortunately, the background signal of NIPBL and WAPL immunofluorescent staining was too high for a reliable quantification.

11. Fig. 2E-F. I may have missed but it seems the maternal result is missing?

Please find the maternal result in Extended Data Fig. 7

12. In Fig. S7C, it would be helpful to also check the expression of cohesin/ctcf/wapl etc.

In the G1-sorted cells, we have blotted for these additional proteins (new Extended Data Fig. 14c)

Referee #2 (Remarks to the Author):

In this very interesting paper, the authors use a combination of in vivo Hi-C approaches and in vitro single molecule experiments to support the hypothesis that MCM complexes loaded on chromatin block loop extrusion by cohesin. Using injection of RNA encoding a non-degradable Geminin protein to block MCM loading during the oocyte to zygote transition, they found that G1 cells with MCM depletion show an increase in peaks and TADs relative to the control. They use some elegant genetics to show that the effect is not mediated through Wapl and they show similar effects are seen in G1 HCT116 cells after MCM2 degradation using an auxin degron. They use a previously described single molecule loop extrusion assay to show that MCM loaded by ORC, Cdc6 and Cdt1 in vitro acts as a boundary for loop extrusion. Finally, via computer simulation, they estimate that loop extrusion is blocked at 10-40% of loaded MCM. The work described here represents an extraordinary

technical achievement. Nonetheless, I have some concerns that I feel need to be addressed.

1. The effects are quite subtle. I assume they are statistically robust (though I saw no statement to this effect); however, the real question is “Is this relatively subtle effect functionally meaningful?” While the effect on TADs will be interesting to workers in the field, it would be of more general interest if the alterations had some demonstrable physiological effect. The most likely such effect would be on G1 transcription. Since the zygotes are not transcriptionally active, they aren’t a great model for this, but the HCT116 system is transcriptionally active. Does MCM depletion affect the transcriptome in this situation in a cohesin-dependent way?

We thank the reviewer for the positive comments and helpful suggestions.

We have performed the experiments to address the reviewer’s question and performed RNA-seq from FACS-sorted G1 phase cells depleted of MCMs or not (Extended Data Fig. 14j). We found at least 229 differentially regulated genes in G1 phase cells, which is roughly comparable in magnitude to gene expression changes upon acute auxin-mediated degradation of CTCF (Nora et al., Cell, 2017). This suggests that MCMs are also affecting transcription in G1 phase.

We speculate that MCM’s barrier function could have multiple effects: it could modify genome architecture and thereby affect gene expression, and it could be part of the mechanism that recruits cohesin to sites of cohesion establishment.

2. The interpretation of the results is absolutely dependent upon the quality of the G1 arrest: If even a few cells enter S phase, this will affect the results +/- MCM loading since there will be replication in one case but not the other, and replication might be expected to affect chromosome structure. I could find no assurance of the quality of synchrony for the zygotes. The very subtle differences in the HCT116 cells could be due to the ~15% non-G1 cells.

Zygotes are naturally synchronized by the timing of *in vitro* fertilization. S phase starts between 8-10 hours after fertilization and we fix cells between 6-6.5 hours after fertilization in G1 phase. To demonstrate that this time point does not include contamination by S phase cells, zygotes were cultured in EdU and fixed at 6.5 h after fertilization. We found that 0/15 cells had incorporated EdU and had therefore not yet entered S phase; zygotes fixed at 12 h showed EdU incorporation (new Extended Data Fig. 2c). Although a small sample size, there are also morphological changes associated with cell cycle progression including an enlargement of the paternal pronucleus and visible pronuclear asymmetry. Both the molecular markers and the visual inspection of pronuclei makes us confident that there are no S phase contaminating nuclei in our preparations. Consistent with this, the scaling plots of individual nuclei all show a similar slope, indicating that there are no outliers that would skew the results.

Based on the reviewer’s concern that the HCT116 cells could be contaminated with non-G1 cells, we increased the stringency of our protocol to obtain a pure G1 population by combining synchronization followed by FACS, which excludes contamination of non-G1 phase cells. The

pure G1 population was used for the micro-C and RNA-seq experiments (new Extended Data Fig. 14b).

3. Can the authors rule out the possibility that more CTCF binds chromatin in the absence of MCM? This could explain at least some of their results.

Although we have no reason to think that the levels of CTCF on chromatin are affected by MCMs, we cannot formally rule out this possibility and have therefore discussed it as an alternative interpretation. In zygotes, we found no evidence for a gross change in CTCF abundance according to immunofluorescent staining of cells in which soluble proteins had been extracted, but a calibrated ChIP would provide a more definitive answer (new Extended Data Fig. 2e). Moreover, in somatic cells, the chromatin-bound fraction of CTCF didn't grossly change either (new Extended Data Fig. 14c). Nevertheless, a model in which the sole effect is mediated by MCMs occupying CTCF sites cannot explain that the results of the single-molecule assays in which cohesin blocks or pauses at MCMs.

4. The in vitro experiments use a mixed system: yeast MCM and human cohesin. This is far from ideal. The inability of cohesin to pass MCM could be due to non-conserved protein interactions, or to differences in the stability of the loaded MCM complex between yeast and humans. It would be far better if this experiment could be repeated with either yeast cohesin or (preferably) human MCM. The Bleichert lab have recently reconstituted MCM loading with human proteins in vitro (Schmidt and Bleichert, Nature Comms, 2020).

We agree that this is a limitation of our assay. Unfortunately, the assay published by the Bleichert lab reconstitutes MCM loading with *Drosophila* proteins (Schmidt & Bleichert, Nature Comm., 2020). Since no functional MCM loading assay has thus far been established for the human proteins, we modified the yeast loading reaction to recruit a chimaeric MCM complex in which yeast MCM3 contains a disordered region with a YDF motif from human MCM3. Although this is not a full substitution of a human MCM complex, the addition of the YDF region had a strong effect by converting blocking to pausing of cohesin at MCM (new Fig 4). This is possibly an important step forward in examining the human cohesin – human MCM interaction *in vitro*.

5. As I understand it, the conclusion that MCM single hexamers can block loop extrusion is based on a very small number (7) molecules. Is this robust? Can you exclude the possibility that these represent double hexamers in which one fluorophore bleached? If not, this discussion should be removed or the experiment repeated with bigger numbers and better controls for bleaching.

We have removed this statement since we cannot easily distinguish double hexamers with one fluorophore from single hexamers.

6. You begin by citing the diameter of MCM and FtsK both being 12-13nm. The diameter of a nucleosome core particle is ~11nm which begs the question, "Can a nucleosome block loop extrusion?" As I understand it, cohesin can compact nucleosomal DNA but can it pass a nucleosome? This seems a rather important control.

At the onset of this work, we considered that the diameter of chromatin-bound proteins would indeed matter. Recently, a preprint by Cees Dekker's lab has shown that nucleosomes and larger nanoparticles do not constitute roadblocks to loop extrusion *in vitro* (Pradhan et al., biorxiv preprint, 2021, <https://doi.org/10.1101/2021.07.15.452501>). This is consistent with our more recent thinking that molecular features other than size are important for determining which proteins constitute barriers to loop extrusion. We consider that the topological binding to DNA and/or the presence of a cohesin interaction motif are relevant for loop extrusion barriers.

7. Related to this, how special is the ability of MCM to block cohesin? Is this likely to be a general property of DNA binding proteins or is there something special about MCM. If so, what is special?

This is indeed the key question. Our experiments with chimaeric MCM provided further insights into the molecular requirements of a cohesin barrier. The presence of the cohesin interaction motif YDF in a disordered region of MCM3 affects the outcome of cohesin-MCM collisions. MCMs without this motif, as is the case for yeast, still function as barriers, which could be due to the topological binding of DNA and potentially interfering with cohesin's ability to "swing" or step over MCM and make DNA contacts on the other side. We hypothesize that YDF or other cohesin interaction motif-containing proteins will also function as loop extrusion barriers.

Referee #3 (Remarks to the Author):

In this paper, Dequeker et al. investigated the possible scenario that DNA-bound proteins serve as barriers to cohesin-mediated loop extrusion. Using single-molecule, Hi-C, and simulation methods, the authors provided evidence that the replicative helicase MCM is a barrier that restricts cohesin translocation and loop extrusion in G1 phase. This is certainly an important and hotly pursued topic. The combination of different techniques is a strength of the paper. The implication that dormant MCM complexes could play a role in genome topology is also intriguing. The manuscript is in general well written. However, I have concerns with the relevance of the single-molecule results, on which my critiques are focused.

The major concern I have, which was also noted by the authors, is that in all of the single-molecule experiments, cohesin was in a passive translocation mode, not in an active loop-extruding mode. Therefore, whether MCM can restrict active loop extrusion remains to be established *in vitro*. The authors briefly mentioned that MCM loading and loop extrusion require mutually incompatible conditions, but did not provide a detailed explanation.

Specific comments on single-molecule results:

(1) 500 mM NaCl was used to promote cohesin sliding. It would be more convincing to visualize cohesin behavior under physiological salt.

We thank the reviewer for the positive comments and helpful suggestions. We have followed the reviewer's suggestion and optimized the salt conditions to promote both MCM loading and cohesin sliding and were able to repeat the experiments under physiological salt conditions. These are shown in the new Figure 4. We found that MCMs function are even slightly stronger barriers under physiological salt conditions than in high salt, which might reflect the disruption of protein-protein interactions at higher salt concentrations.

(2) How did the authors conclude that only MCM double-hexamers, but not ORC/Cdc6/Cdt1-containing intermediates, were formed on DNA?

A high salt wash was performed during all experiments after MCM loading, which removed ORC/Cdc6/Cdt1 from all DNA molecules, now also highlighted in the main text on page 7. The complete removal of all ORC/Cdc6/Cdt1-containing intermediates by high salt washes was confirmed both in bulk MCM loading assay conducted on beads as well as under single-molecule conditions in Scherr et al., 2021 <http://dx.doi.org/10.2139/ssrn.3775178>. The figure reporting the results of the bulk loading assay from that manuscript is shown below for convenience. The relevant result can be found in panel b, which clearly shows that, for both wild type and (via ybbR tag) fluorescently labeled MCMs, Cdc6 and ORC (with Orc6 as a read-out) are removed by high salt washes when the loading reaction is conducted in the presence of ATP as performed for all cohesin experiments.

The figure also displays the results for ATP_γS where the loading process stops at an intermediate state called the OCCM containing ORC and one MCM ring. These complexes exhibit lower stability and are easily removed by a high salt wash; therefore, we did not evaluate the ATP_γS condition in the current study.

Fig. 1: Origin licensing intermediates are removed when challenged with high salt.

a, Schematic of a bulk helicase loading assay. ARS1-containing 5 kb DNA bound to a magnetic bead was incubated with licensing factors ORC, Cdc6 and Cdt1-MCM and subsequently washed with a low or high salt (HS) buffer. **b, c**, Bulk helicase loading assay as described in **(a)** in the presence of ATP or ATPγS. **b**, Eluted protein was analyzed by Western Blot using protein specific primary antibodies for Orc6, Cdc6 and Mcm4. **c**, Integration of Mcm4 signals showed no difference in loading efficiency for Cdt1-MCM containing wildtype (wt) or ybbR-LD655 Mcm6 (n = 4 and n = 2 for ATP and ATPγS condition, respectively). Error bars represent SEM. **d**, Schematic of the single-molecule helicase loading assay. ARS1-containing 21 kb DNA was incubated with licensing factors ORC, Cdc6 and Cdt1-MCM and imaged on flow-stretched DNA after removal of excess protein. **e**, Number of MCM foci on ARS1-DNA challenged with low or high salt (HS) after helicase loading in the presence of ATP or ATPγS. Error bars represent the estimated SEM by bootstrapping. Figure adapted from Scherr et al., 2021.

In Fig 3D, the origin bypassing probability without MCM for the majority of cohesins is lower than 0.6. Is it because ORC remained bound at the origin, or is it a DNA sequence effect?

We thank the reviewer for this comment. As explained above (2), the performed high salt wash prior to imaging cohesin translocation ensured removal of ORC or any other licensing factors or loading intermediates from DNA. Thus, we do not anticipate residual ORC at the origin. In fact, passive cohesin translocation is a random walk based / Brownian movement. Therefore, cohesin is expected to spontaneously change the direction of translocation, also within our chosen threshold. Thus, the bypassing probability in the absence of MCM is expected not to be 1 but rather around 0.5. In line with this assumption, we also determined the bypassing without MCM at different DNA regions (sequences) and found similar probabilities. Therefore, we also exclude a DNA sequence effect.

(3) MCM double-hexamers have been shown to be pushed by DNA translocases and slide on DNA (e.g. PMC4680849). Was MCM always stationary in the current assay?

We appreciate the reviewer raising this point. In fact, we have recently investigated MCM sliding and displacement during encounters with RNA polymerase (Scherr et al., 2021 <http://dx.doi.org/10.2139/ssrn.3775178>). We observed only ~ 1% of MCMs sliding at 500 mM NaCl. Therefore, observing a sliding MCM and cohesin on the same DNA molecule is very unlikely and would only be possible at higher salt concentrations. All DNA molecules analyzed and reported contained stationary MCMs.

(4) Depending on the distance between the two anchored DNA ends, the MCM and cohesin localization precision is expected to differ between molecules that were tight versus slack. However, the same thresholds were used to classify encounter and passing for all molecules. What is the distribution of observed DNA lengths? How was the 1.5-kb threshold chosen?

This is a very good point. We have corrected for these small differences in localization precision in the revised manuscript by individually measuring the lengths of all DNA molecules and calculating positions accordingly. The distribution of observed DNA lengths can be seen in Extended Data Fig. 15g. The 1.5-kb threshold for encounters was determined by calculating the expected ARS-encounter frequency based on the observed diffusion coefficient. The threshold accounts for the limited temporal and spatial resolution and ensure encounters are nonetheless detected.

We have added an extensive description of the length correction procedure and threshold determination in the methods section of the revised manuscript.

Comments on Hi-C and simulation results (I should admit that I am not an expert on either of these areas):

(1) Most of the Hi-C results are presented in a descriptive manner and lack quantification (error bars, statistical tests, etc.). The effect of MCM loss in somatic cells is particularly difficult to judge by eye (Fig S7).

We have plotted the loop strength of the individual samples for the WT vs MCM loss conditions and calculated the weighted mean based on number of cis-contacts 1kb+ for each sample (new Extended Data Fig. 3a).

To improve the resolution, we performed micro-C on HCT116 cells and found that aggregate peaks and TADs are increased over a range of sizes and more loops can be detected *de novo* (new Extended Data Fig. 14).

(2) The simulated and experimental data were claimed to be well matched (e.g. Fig 4D, Fig S14B-D). What are the criteria? They certainly do not perfectly overlap.

The criteria for comparing the experimental data and the simulated data were two-fold. First, we computed from snHi-C the corner peak strength above background. Secondly, we computed the $P_c(s)$ curves from simulations and experiments genome wide, aiming to achieve the best agreement in the 30 kb-1 M range. The choice of the range is motivated by following considerations: (i) that the effect of cohesin on $P_c(s)$ typically only extends up to ~1 Mb under normal conditions; (ii) below 30 kb, Hi-C data has been shown to contain artifacts and can vary significantly between different protocols.

The criteria then for evaluating the goodness of simulation, were 1) to obtain quantitative values for the corner peak strengths as close as possible to the experiments, reproducing the correct relative order of the loop strength between various conditions, i.e. in paternal nuclei, the corner peak strength from weakest to highest was: dCTCF, WT, dWapl, dMCM, dMCM/Wapl). We directly scored the goodness of the simulation by minimizing the absolute error between the simulated and experimental corner peak strengths. 2) Simultaneously, we evaluated the absolute values and shapes of the $P_c(s)$ curves between 30 kb-1 Mb. The goodness of $P_c(s)$ fit was evaluated by visual agreement. Therefore, we used a combined approach to evaluate the match between experiments and simulations, where the dot strength and $P_c(s)$ curves were evaluated together.

To clarify these points, we have added a new section to the Methods describing the above points called “Comparing simulated and experimental data”.

Referee #4 (Remarks to the Author)

In this study, Dequeker et al. use a combination of mouse genetics, Hi-C, in vitro experiments and modeling to study the interplay between minichromosome maintenance (MCM) complexes and loop extrusion by cohesin in mammalian genomes. The experimental and computational results in the manuscript suggest, but do not fully support, that MCM complexes form semi-permeable barriers to loop extrusion and thus modulate contact probabilities between CTCF sites, at least in the fertilized oocyte genome.

Provided some major concerns are addressed (see below), the study will be of interest for the nuclear structure field and in particular for the community of researchers interested in modeling loop extrusion. However, the relevance of the finding that replication origins might obstruct loop extrusion through MCM binding (which is by the way not formally proven in the manuscript) seems rather limited, notably by the fact that such effect does not seem to be observable in differentiated cell types.

Main points:

1. The central result in the paper is the increase in Hi-C enrichment at CTCF loops following inhibition of MCM loading. This effect is however rather small (~20%) and detected after averaging over very small numbers of pronuclei (with WT and mutant experiments involving different numbers of pronuclei). This raises the question whether the effect could be dominated by some outlier single-cell Hi-C preparations within either the wild-type or the mutant samples. To exclude this possibility, it would be necessary to verify that the results

hold true when different (smaller) numbers and combinations of single-cell Hi-C samples are retained in the analysis (e.g. 6, 8, 10).

We thank the reviewer for the helpful comments. Based on our previous snHi-C studies (Flyamer et al., *Nature*, 2017; Gassler et al., *EMBO J*, 2017; Silva et al., *J Cell Biol*, 2020; Chatzidaki et al., *Curr Biol*, 2021), we determined that the down-sampling of *cis* contacts in addition to having similar number of nuclei is a good measure to compare different conditions. All data presented in this manuscript has been down-sampled to the smallest number of *cis* contacts common to all conditions and the results have remained the same. To exclude that the observed effect is dominated by an outlier, i.e. cells with high number of *cis*-contacts, we plotted the loop strength of the individual samples and calculated the weighted mean based on number of *cis*-contacts 1kb+ for each sample (new Extended Data Fig. 3a). These results show that the increase in loop strength upon MCM loss is statistically significant. To further exclude skewing by any outliers, we performed combinations of single-cell Hi-C samples as suggested above and included these as new Extended Data Fig. 3c, showing the same results as before.

The measure of aggregate loop strength is the most valuable when compared across conditions. We therefore find it informative to compare the magnitude of effect by MCM loss to that of Wapl deletion because the latter is known to increase the strength of cohesin-mediated Hi-C features and cause chromatin compaction into “vermicelli” chromosomes in several cell types including oocytes and zygotes (Tedeschi et al., *Nature*, 2013; Gassler et al., *EMBO J*, 2017; Silva et al., *J Cell Biol*, 2020,). Notably, Wapl deletion has a smaller effect on aggregate loop strengths than does the loss of MCMs for paternal chromatin. We therefore consider MCM loss to have a strong effect on CTCF-anchored loops.

2. Based on single-cell RNA-seq performed in two WT and two ‘MCM-loss’ 10-cell samples, the authors conclude that there are no major differences in gene expression following loss of MCM. However, WT replicates do not cluster together in PCA (Figure S3B) and their reciprocal distance in the two first PCA components is of the same order of their distances to MCM-loss samples. It is thus unclear what can be concluded from these experiments.

We agree that there is substantial variability between the low input RNA-seq samples. To obtain better insights into the clustering, we compared PCA plots of G1 control, G2 control and G1 MCM loss samples and find that the MCM loss samples preferentially cluster together (Extended Data Fig. 4). Nevertheless, we have toned down this section as the main point was to reiterate that these cells are transcriptionally inactive and that therefore effects on 3D genome architecture are not via gene expression changes.

3. At page 4, the authors claim that MCM depletion further decreases insulation scores compared to depleting CTCF alone. However, it is impossible to assess whether the minimal differences shown in Figure 3C are to any extent significant. At the very least, insulation analysis should be performed upon averaging over different numbers and combinations of single pronuclei, as well as using different genomic windowing.

During the revision we have increased the number of nuclei in which CTCF and MCM are co-depleted from 8 to 18. The larger data set shows that there is little difference in TAD boundary insulation with or without MCMs. We therefore conclude that CTCF is instructive for boundary formation and that MCMs are additional, randomly positioned barriers. We have amended the section accordingly.

Please also see the response to reviewer 1 (point 4) on this point.

4. Why are the data for maternal and paternal pronuclei merged in Figure 2B (CTCF-MCM depletion) but not in Figure 2C (WAPL-MCM depletion)? Does it relate to the fact that upon WAPL depletion, only the scaling of contact probabilities in the maternal genome changes (Fig. S6D)? And why is it so?

The reviewer is correct in that we merged maternal and paternal snHi-C data for CTCF depletion because there are no parental-specific effects, unlike for Wapl. However, for consistency, we have revised this and show all paternal effects in Figure 2 as these represent *de novo* MCM loading during G1 phase and the maternal effects as Extended Data Fig. 6 and 7 in the revised manuscript.

The reviewer raises an interesting point as to why WAPL depletion affects the scaling of contact probabilities of the maternal genome more than the paternal genome. We speculate that Wapl depletion, which is occurring in the oocyte, leads to retention of Scc1-cohesin throughout the meiotic divisions. Indeed, Scc1 becomes detectable on condensed meiosis I chromosomes of oocytes without Wapl (Silva et al., J Cell Biol, 2020). Therefore, more cohesin might be carried over on maternal chromatin into zygotes, possibly leading to more compaction. In contrast, Wapl depletion will only affect *de novo* loaded cohesin on paternal chromatin in G1 phase zygotes and might therefore have a less extensive effect on compaction.

5. The claim that in HCT116 cells the effects of MCM depletion are consistent with those observed in the zygotes but less pronounced is an overstatement and should be toned down. There seems to be absolutely no effect in this cell line judging from Fig. S7.

We understand that the smaller effect on CTCF-anchored loops upon MCM degradation in bulk Hi-C data from HCT116 cells raises the concern that this might be noise. We therefore used another method, micro-C, to determine whether this effect is reproducible. Duplicate micro-C data confirmed an increase in aggregate loop strength and a visible enrichment of corner peaks on heat maps in MCM-depleted cells (new Extended Data Fig. 14). In addition, we can reliably detect a larger number of *de novo* called loops over a range of distances in MCM-depleted cells, providing further support that extruding cohesin reaches CTCF sites more frequently in the absence of MCM barriers. Although these effects are smaller than what is observed in zygotes, the Hi-C and replicate micro-C data support a barrier function for MCMs in HCT116 cells.

6. As also acknowledged by the authors, there is a clear disconnect between the three parts of the manuscript (*in vivo*, *in vitro*, simulations). The *in vitro* experiments show that MCM can impair cohesin translocation, but do not necessarily relate to the *in vivo* part because they provide no evidence that MCM complexes can also interfere with loop extrusion. The simulations instead can reproduce the *in vivo* data, but they only do so if MCM permeability for loop extruding cohesin is much larger than measured *in vitro* for passive translocation (80% vs. 20%). Taken together, the three parts suggest that MCM-mediated interference of loop extrusion *could* be the cause of the observed phenotype but constitute no formal proof. *In vitro* experiments with loop extruding cohesin would provide at least part of the missing link and strengthen the causality claims.

All three approaches are providing parts of the answer of how MCMs affect cohesin-mediated loop extrusion. The most parsimonious interpretation of the *in vivo* data from zygotes and HCT116 cells is that MCMs function as loop extrusion barriers. This is supported by simulation data. The *in vitro* assay provides evidence that MCMs can be obstacles that block or even pause cohesin. Together, these are pieces of a puzzle that fit a model in which MCMs function as loop extrusion barriers.

We agree that the ideal experiment would be to directly examine the collision of loop extruding cohesin with MCMs *in vitro*, as we pointed out in the manuscript, but the conditions under which extrusion and MCM loading/retention occur are very different and challenging to combine. The establishment of such an assay is beyond the scope of this manuscript.

Indeed, we would like to point out that examining barriers to loop extrusion *in vitro* is far from a trivial task. CTCF has been studied as a barrier for more than five years in the genome architecture field and there is still no report of whether CTCF is capable of stopping loop extrusion *in vitro*. This powerfully illustrates the challenge of such an assay.

Nevertheless, we have made strong headway in understanding the cohesin-MCM interaction through the chimaeric MCM experiments and the YDF motif.

Minor point:

It is unclear what the numbers inside meta-loop and meta-TAD boundaries refer to. I presume it is some sort of mean enrichment in a region of the plot, but the Methods section is not clear in this regard.

The numbers in aggregate loop and TAD panels indicate the signal over background ratio. In case of average loops, this is calculated by taking the average value of the central 3x3 square of the normalized pileup as shown in the panel below (adapted from Gassler et al., EMBO J, 2017). The loop strength is the average value within the middle black box divided by the average of the combined top-left and bottom-right green boxes.

Reviewer Reports on the First Revision:

Referees' comments:

Referee #1 (Remarks to the Author):

In this revised version, the authors have clearly improved the manuscript by adding new data, providing replicate information, or revising previous statements. Many of my previous concerns have been carefully addressed and I appreciate the authors' efforts in conducting these experiments.

At this moment, the weakness of the finding is still the relatively subtle impacts of chromatin organization upon the loss of MCM in somatic cells. Although I appreciate the endeavor of the authors to use micro-C to detect the structural changes upon MCM loss in somatic cells, I feel the changes are still rather weak based on the data presented in the revision. It is perfectly possible that weak effects may have profound functions. Yet, it is unclear if this is the case here. Regarding the MCM's barrier functions in this study, the authors identified 229 differential genes upon MCM loss in HCT116 cells. It remains unknown if these changes are related to the alteration of chromatin organization, or other functions of MCM. Mechanistically, the chimeric MCM provides a tantalizing hint that MCM may actively stall cohesin. Yet, future experiments may be needed to confirm this result given the human-yeast chimeric nature of MCM.

Referee #2 (Remarks to the Author):

The authors have done a tremendous job of addressing my concerns with the original submission. Although I still harbor some reservations about the hybrid system biochemistry, I don't think there is anything else they could do about this until a human MCM loading system is developed. I am fully supportive of publication.

Referee #3 (Remarks to the Author):

The authors addressed most of the technical concerns that I had. However, my main conceptual concern, i.e. whether MCMs represent effective barriers against loop-extruding cohesin, remains to be addressed. The authors acknowledged this shortcoming of their in vitro experiments and explained why it is challenging, if not impossible, to find a condition simultaneously compatible with loop extrusion and MCM loading. I believe that it is still critical to establish this link, but I will leave it to the editors to decide whether this is an indispensable piece of the study.

One specific comment: the observation that cohesin bypassing probability does not further decrease when multiple MCMs are loaded at the origin (Extended Data Fig. 15f) is somewhat unexpected, which indicates that a certain fraction of cohesin molecules is not sensitive to MCM barrier at all. Could they actually represent the "functional" cohesins capable of loop extrusion? This seems to be an important scenario to consider.

Referee #4 (Remarks to the Author):

The authors have performed an impressive amount of additional analyses and experiments that address my major concerns.

I would only like to express some residual doubts on the effects of MCM depletion in HCT116 cells, which are not very convincing despite the newly provided microC experiments. Increases in loop strength and number remain truly marginal even when assessed using microC. To make a stronger statement the results should, in my opinion, be verified in (and showed for) two independent replicates. It is currently unclear if quantifications shown in Extended Data Fig. 14 e-i arise from the two pooled micro-C replicates or only one of the two.

Author Rebuttals to First Revision:

We would like to thank the Referees for the overall very positive comments and are grateful for the opportunity to respond to the remaining questions. In brief:

- Referee 3 raised a concern that the barrier strength does not change even when multiple MCMs are loaded and suggests that this might reflect different populations of cohesin complexes. We reason that the step size of SMC complexes could provide an explanation for the similar barrier strength of one and two double-hexamers of MCM complexes. As is explained below, our data do not indicate that there are different functional fractions of cohesin molecules.
- Referee 4 raised a concern about the reproducibility of chromatin structure changes following MCM depletion as analyzed by micro-C. We provide a separate analysis of each micro-C experiment to demonstrate that independent/biological replicates show an increase in loop strength and number. Furthermore, additional analysis showed that there is a statistically significant difference genome-wide upon MCM loss (new Extended Data Fig. 8). The same effect was also seen in two Hi-C experiments.

Please find the specific responses to the Referees' comments below.

Referee #1 (Remarks to the Author)

In this revised version, the authors have clearly improved the manuscript by adding new data, providing replicate information, or revising previous statements. Many of my previous concerns have been carefully addressed and I appreciate the authors' efforts in conducting these experiments.

At this moment, the weakness of the finding is still the relatively subtle impacts of chromatin organization upon the loss of MCM in somatic cells. Although I appreciate the endeavor of the authors to use micro-C to detect the structural changes upon MCM loss in somatic cells, I feel the changes are still rather weak based on the data presented in the revision. It is perfectly possible that weak effects may have profound functions. Yet, it is unclear if this is the case here. Regarding the MCM's barrier functions in this study, the authors identified 229 differential genes upon MCM loss in HCT116 cells. It remains unknown if these changes are related to the alteration of chromatin organization, or other functions of MCM. Mechanistically, the chimeric MCM provides a tantalizing hint that MCM may actively stall cohesin. Yet, future experiments may be needed to confirm this result given the human-yeast chimeric nature of MCM.

We thank the reviewer for the supportive comments. We performed further analysis on the correlation between gene expression changes and chromatin contact frequencies around the transcriptional start sites (TSSs) upon MCM loss. We found that MCM loss leads to fewer contacts near differentially regulated genes over genomic ranges of TADs. Contact frequencies changed for expressed but not for non-expressed genes. These findings imply that transcriptionally active genes are sensitive to MCM abundance. We attach an updated version of Extended Data Fig. 14 (new extended Data Fig. 8) and can include this in a revised version of the manuscript.

Referee #2 (Remarks to the Author)

The authors have done a tremendous job of addressing my concerns with the original submission. Although I still harbor some reservations about the hybrid system biochemistry, I don't think there is anything else they could do about this until a human MCM loading system is developed. I am fully supportive of publication.

We greatly appreciate the reviewer's support of this work.

Referee #3 (Remarks to the Author)

The authors addressed most of the technical concerns that I had. However, my main conceptual concern, i.e. whether MCMs represent effective barriers against loop-extruding cohesin, remains to be addressed. The authors acknowledged this shortcoming of their in vitro experiments and explained why it is challenging, if not impossible, to find a condition simultaneously compatible with loop extrusion and MCM loading. I believe that it is still critical to establish this link, but I will leave it to the editors to decide whether this is an indispensable piece of the study.

One specific comment: the observation that cohesin bypassing probability does not further decrease when multiple MCMs are loaded at the origin (Extended Data Fig. 15f) is somewhat unexpected, which indicates that a certain fraction of cohesin molecules is not sensitive to MCM barrier at all. Could they actually represent the "functional" cohesins capable of loop extrusion? This seems to be an important scenario to consider.

We also found it interesting that multiple MCMs did not significantly increase the barrier strength. However, this observation is not the result of a certain fraction of cohesin molecules being insensitive to MCM barriers. The distributions displayed in the original Extended Data Fig. 15f are the result of many individual cohesin molecules exhibiting all outcomes (stopping, turning around, and bypassing) over the observation time. If a fraction of cohesins were insensitive to the MCM barrier, then we would expect these molecules to always bypass MCM barriers. We do not observe this. Instead, all cohesin molecules reported in our study were equally functional for translocation and during encounters with MCMs. Therefore, our data do not indicate that we observed different fractions of cohesins.

Since our study was focused on the outcomes during encounters with single double-hexamers, we performed experiments under conditions in which no more than two loaded double-hexamers (4 MCMs) were observed. Whether 4 rather than 2 MCMs are a more significant barrier will depend on the bypassing mechanism. The extended size of SMC complexes and step sizes may determine the strength of barriers. The step size of cohesin has not been directly determined but the related condensin complex has been reported to extrude loops at a step size of 200-500 bp (Ryu et al., *bioRxiv* 2020). A single double-hexamer (2 MCMs) shields about 60 bp of DNA (Noguchi et al., *PNAS* 2017). Therefore, we speculate that more than 4 MCMs may be needed to substantially alter the barrier strength.

Referee #4 (Remarks to the Author)

The authors have performed an impressive amount of additional analyses and experiments that address my major concerns.

I would only like to express some residual doubts on the effects of MCM depletion in HCT116 cells, which are not very convincing despite the newly provided microC experiments. Increases in loop strength and number remain truly marginal even when assessed using microC. To make a stronger statement the results should, in my opinion, be verified in (and showed for) two independent replicates. It is currently unclear if quantifications shown in Extended Data Fig. 14 e-i arise from the two pooled micro-C replicates or only one of the two.

We thank the reviewer for pointing out this ambiguity in the data representation and agree that showing the analysis of independent replicates makes a stronger statement. We clarify that the results presented in the original Extended Data Fig. 14 e-i arise from two pooled replicates. We have also analysed the two biological/independent replicates downsampled to the same number of contacts separately and show the increases in loop strength and number for each replicate separately in an updated Extended Data Fig. 8. Importantly, further analysis revealed that MCM depletion results in a moderate but genome-wide and significant ($p=1.87\times 10^{-70}$) increase in peak strength (see below, panel a and Extended Data Fig. 8i). Since the *de novo* calling of loops is a powerful measure applicable to bulk data, we re-analyzed all data and included below a summary showing an increase in loop numbers and loop strengths upon MCM depletion for all Hi-C and micro-C experiments (panels b and c). The reproducibility of these results strengthens the conclusion that MCMs impede formation of anchored loops in somatic cells.

There are multiple reasons that convince us that MCMs cause biologically relevant differences in chromatin structure. While we agree that the effects of MCM depletion are smaller in somatic cells, they are reproducible between methods (Hi-C and micro-C) and replicates. They principally show the same phenomenon in somatic cells as they do in zygotes, where the effects of MCM loss are even stronger than those of Wapl loss, which can be considered a gold standard of chromatin structure changes (Tedeschi et al. *Nature* 2013; Haarhuis et al., *Cell* 2017; Gassler et al., *EMBO J* 2017; Wutz et al., *EMBO J* 2017).

Summary figure: MCM depletion results in an increase in loop number and strength in HCT116 cells

- a) Histogram showing the distribution of \log_2 ratio of peak strengths in MCM-depleted and control cells within 1 Mb bins across the whole genome, normalized to global and local background of interactions. Higher values indicate increase of peak strength upon MCM depletion. Mean of the distribution is highly significantly different from 0 (one sample T-test), $p=1.87 \times 10^{-70}$
- b) Number of *de novo* loops (called with *Mustache*) in independent Hi-C and micro-C experiments. Trp (triptolide) is a transcription inhibition experiment performed with and without MCM depletion. All experiments were performed in HCT116 MCM2-mAID cells, except for the Hi-C untagged experiment, where a parental HCT116 cell line treated with auxin was used as control. Different experiments were sequenced to different depths and all pair-wise comparisons of control vs. experimental data sets were downsampled to a similar number of total contacts.
- c) Average of the total contact frequency of loops in an aggregate peak analysis for control and MCM-depleted HCT116 cells in independent Hi-C and micro-C experiments.

Reviewer Reports on the Second Revision:

Referees' comments:

Referee #4 (Remarks to the Author):

The authors now convincingly use using MicroC replicates to support the statement that MCM depletion leads to small changes in contact probabilities in HCT116 cells. It will be important to clearly state in the manuscript that changes are very subtle, and that it is mostly their direction (rather than their magnitude) that mostly matches the trend observed in oocytes.